# Neural mechanisms of parasite-induced summiting behavior in 'zombie' *Drosophila*

**Carolyn Elya\*, Danylo Lavrentovich, Emily Lee[†], Cassandra Pasadyn[‡], Jasper Duval[§], Maya Basak[#], Valerie Saykina[¶], Benjamin de Bivort\***

Department of Organismic and Evolutionary Biology, Harvard University, Cambridge, United States

**\*For correspondence:**
cnelya@gmail.com (CE);
debivort@oeb.harvard.edu (BdB)

**Present address:** [†]New York Genome Center, New York, United States; [‡]The Ohio State University College of Medicine, Columbus, United States; [§]Northeastern University, Boston, United States; [#]Emory University, Atlanta, United States; [¶]University of Connecticut, Storrs, United States

**Competing interest:** The authors declare that no competing interests exist.

**Abstract** For at least two centuries, scientists have been enthralled by the "zombie" behaviors induced by mind-controlling parasites. Despite this interest, the mechanistic bases of these uncanny processes have remained mostly a mystery. Here, we leverage the *Entomophthora muscae-Drosophila melanogaster* "zombie fly" system to reveal the mechanistic underpinnings of summit disease, a manipulated behavior evoked by many fungal parasites. Using a high-throughput approach to measure summiting, we discovered that summiting behavior is characterized by a burst of locomotion and requires the host circadian and neurosecretory systems, specifically DN1p circadian neurons, pars intercerebralis to corpora allata projecting (PI-CA) neurons and corpora allata (CA), the latter being solely responsible for juvenile hormone (JH) synthesis and release. Using a machine learning classifier to identify summiting animals in real time, we observed that PI-CA neurons and CA appeared intact in summiting animals, despite invasion of adjacent regions of the "zombie fly" brain by *E. muscae* cells and extensive host tissue damage in the body cavity. The blood-brain barrier of flies late in their infection was significantly permeabilized, suggesting that factors in the hemolymph may have greater access to the central nervous system during summiting. Metabolomic analysis of hemolymph from summiting flies revealed differential abundance of several compounds compared to non-summiting flies. Transfusing the hemolymph of summiting flies into non-summiting recipients induced a burst of locomotion, demonstrating that factor(s) in the hemolymph likely cause summiting behavior. Altogether, our work reveals a neuro-mechanistic model for summiting wherein fungal cells perturb the fly's hemolymph, activating a neurohormonal pathway linking clock neurons to juvenile hormone production in the CA, ultimately inducing locomotor activity in their host.

## Editor's evaluation

The phenomenon of summit disease, where complex animal behaviours are controlled by single-celled parasites, captivates biologists and non-scientists alike. In this valuable study, the authors use a laboratory model (*Drosophila melanogaster* infected with Entomophthora muscae) for this disease to provide compelling evidence for the neuroanatomical and physiological underpinnings of summit disease. This is an excellent example of how seemingly intractable questions in behavioural ecology can be effectively addressed in laboratory settings using decades of work in creating 'models' for biology.

## Introduction

Many organisms infect animals and compel them to perform specific, often bizarre, behaviors that serve to promote their own fitness at the expense of their host. For example, 'zombie ant' fungi of genus *Ophiocordyceps* compel their host carpenter ants to aberrantly leave the nest, wander away from established foraging trails, scale nearby stems or twigs, and, in their dying moments, clamp onto vegetation to ultimately perish in elevated positions (*Hughes et al., 2011*; *Pontoppidan et al., 2009*). Days later, a fungal stalk emerges from the dead ant's pronotum, well poised to rain spores on the ants that forage below (*Evans and Samson, 1984*). But this is far from the only example: jewel wasps that subdue cockroaches (*Gal and Libersat, 2010*), protozoans that suppress a rodent's fear of cat odors (*Vyas et al., 2007*), and worms that drive crickets to leap to watery deaths are all examples of parasites hijacking host behavior (*Thomas et al., 2002*).

One of the most frequently encountered behavior manipulations in parasitized insects is summit disease (also referred to as tree-top disease or Wipfelkrankheit) (*Hofmann, 1891*). Summit disease is induced by diverse parasites, ranging from viruses to fungi to trematodes, and affects a broad range of insect species, including ants, beetles, crickets, caterpillars, and flies (*Goulson, 1997*; *Hughes et al., 2011*; *Krasnoff et al., 1995*; *Loos-Frank and Zimmermann, 1976*; *Pickford and Riegert, 1964*; *Steinkraus et al., 2017*). The most consistently reported symptom of summit disease is elevation prior to death (*Evans, 1989*; *Lovett et al., 2020*; *Roy et al., 2006*). This positioning advantages the parasite by either making the spent host more conspicuous, and therefore, likely to be consumed by the next host in its life cycle (e.g. *Dicrocoelium dendriticum*-infected ants; *Martín-Vega et al., 2018*), or by positioning the spent host for optimal dispersal of infectious propagules (e.g. *Mamestra brassicae* nuclear polyhedrosis virus; *Goulson, 1997*).

Some of the deepest mechanistic understanding of parasite-induced summiting comes from nucleopolyhedroviruses (NPVs). Disrupting the *ecdysteroid uridine 5'-diphosphate* (*egt*) gene in NPVs of the moths *Lymantria dispar* or *Spodoptera exigua* prevents summiting in infected larvae (*Han et al., 2015*; *Hoover et al., 2011*). This effect is thought to occur via *egt*'s inactivation of the hormone 20-hydroxyecdysone and the resulting disruption of molting (*O'Reilly and Miller, 1989*). However, *egt* has been found to be dispensable for driving summit disease in other NPV-insect systems (*Kokusho and Katsuma, 2021*), suggesting there are undiscovered viral mechanisms driving summiting in NPV-infected hosts. On the host side, evidence in NPV-infected *L. dispar* and *Helicoverpa armigera* point to changes in the host phototactic pathway underlying summiting behavior (*Bhattarai et al., 2018*; *Liu et al., 2022*). Outside of NPVs, work in *Ophiocordyceps* suggests that the parasitic fungus may use enterotoxins and small secreted proteins to mediate end-of-life 'zombie' behaviors (*Beckerson et al., 2022*; *de Bekker et al., 2015*; *Will et al., 2020*), potentially targeting host phototaxis (*Andriolli et al., 2019*), circadian rhythm, chemosensation, and locomotion (*de Bekker et al., 2015*; *Trinh et al., 2021*; *Will et al., 2020*).

*Entomophthora muscae* is a behavior-manipulating fungal pathogen that infects dipterans and elicits summit disease prior to host death (*Graham-Smith, 1916*; *MacLeod et al., 1976*). *E. muscae* infection begins when a fungal conidium (informally: spore) ejected from a dead host lands on a fly's cuticle. The spore penetrates the cuticle and enters the hemolymph where it begins to replicate, first using the fat body (a tissue analogous to the liver and used for storing excess nutrients) as a food source (*Brobyn and Wilding, 1983*). When nutrients are exhausted, *E. muscae* elicits a stereotyped trio of behaviors to position its dying host for the next round of spore dispersal. The fly (1) summits (*Graham-Smith, 1916*), (2) extends its proboscis, which glues the fly in place via sticky, exuded secretions (*Brobyn and Wilding, 1983*), and finally, (3) the fly's wings lift up and away from its dorsal abdomen, clearing the way for future spore dispersal (*Elya et al., 2018*; *Krasnoff et al., 1995*). Fungal structures (conidiophores) then emerge through the cuticle and forcefully eject infectious spores into the surrounding environment via a ballistic water cannon mechanism (*de Ruiter et al., 2019*). *E. muscae* kills flies at a specific time of day: flies die around sunset and exhibit their final bout of locomotion between 0–5 hr prior to lights off (*Elya et al., 2018*; *Krasnoff et al., 1995*). Time-of-day specificity is a common feature of fungal-induced summit disease: *Ophiocordyceps*-infected ants die around solar noon (*Hughes et al., 2011*), *Entomophaga grylli*-infected grasshoppers within a 4 hr window prior to sunset (*Roffey, 1968*), and *Erynia neoaphidis*- and *Entomophthora planchoniana*-infected aphids die most frequently around 8.5 and 14 hr after sunrise, respectively (*Milner et al., 1984*).

*E. muscae*-infected 'zombie flies' have been known to the scientific literature for the last 167 years (*Cohn, 1855*), yet the mechanistic basis of their behavior manipulation is still a mystery. It is challenging to culture *E. muscae* in the laboratory and typical host species, like houseflies, lack experimental access. A strain of *E. muscae* that infects fruit flies was recently isolated and used to establish a laboratory-based 'zombie fly' system in the tool-replete model organism *Drosophila melanogaster* (*Elya et al., 2018*), permitting investigation of the specific host mechanisms underlying manipulated behaviors.

The rich experimental toolkit of *D. melanogaster* has been used to decipher the mechanistic underpinnings of host-symbiont interactions ranging from mutualism to parasitism. For example, a mutant screen identified the Toll pathway as essential for *Drosophila*'s antiviral immune response (*Zambon et al., 2005*). Genetic access to specific neuronal populations allowed the identification of class IV neurons as mediating the larval escape response to oviposition by *Leptopilina boulardi* wasps (*Robertson et al., 2013*). It was recently shown that the gut bacterium *Lactobacillus brevis* alters fly octopaminergic pathways to drive an increase in locomotion (*Schretter et al., 2018*). Fruit flies have also been leveraged to investigate mechanisms of medically important parasites naturally vectored by other dipterans, including the protozoans *Plasmodium*, *Leishmania,* and *Trypanosoma* (*dos-Santos et al., 2015*; *Peltan et al., 2012*; *Tonk et al., 2019*).

Here, we describe our progress using the zombie fruit fly system to unravel the mechanistic basis of summiting behavior. We first show that the hallmark of summiting behavior is an increase in locomotion beginning ~2.5 hr before death. By combining the powerful fruit fly genetic tool kit with a custom high-throughput behavioral assay, we demonstrate that the fly circadian and neurosecretory systems—specifically DN1p clock neurons, pars intercerebralis projection neurons that innervate the corpora allata (PI-CA neurons), and the juvenile hormone-producing corpora allata—are essential components mediating summiting. Using a real-time machine learning classifier to identify the moment flies begin to summit, we were able to characterize the anatomy and physiology of summiting flies with temporal precision. We found that *E. muscae* specifically invades the brain region harboring DN1p axons and PI-CA dendrites. The hemolymph of summiting flies contains specific metabolites that, when transfused into recipient flies, induce summiting-like locomotion. Taken together, these experiments reveal that *E. muscae* uses hemolymph-borne factors, targets a specific neural circuit, and hijacks endogenous neurohormonal control of locomotion.

## Results

### A novel assay to measure summiting behavior

We first set out to develop an assay that would allow us to characterize the behavioral mechanisms of summit disease (*Figure 1A*). Given the variability in the day and exact time when flies die, and the unknown duration of summiting, our assay needed to accommodate continuous monitoring of flies over many hours. The assay also needed to allow flies to express behavior with respect to the direction of gravity. We also wanted to make sure our chambers provided enough space for flies to lift their wings without interference (*Figure 1B*). Each behavioral arena was 65 mm long along the main gravitational axis, 5 mm wide, and 3.2 mm deep, and housed a single fly (*Figure 1C*). The bottom of the chamber was plugged with food to sustain flies over long periods of observation (24–96 hr). Four rows of 32 arenas each were fabricated in laser-cut acrylic trays, allowing us to measure the behavior (position along the main gravitational axis, referred to as 'relative y position,' and overall speed) of 128 flies simultaneously. Trays and the imaging boxes that housed them were angled at 30° (*Kladt and Reiser, 2023*) to provide the gravitactic gradient (*Figure 1C*).

We first monitored *E. muscae*-exposed wild-type (Canton-S) flies. Experiments started no later than Zeitgeber time 20 (ZT20, i.e. 19 hr after the dark-to-light transition) on the day prior to their earliest possible death, until flies either succumbed to or survived their infection (ZT13 of day 4–7, depending on the experiment). After tracking, we manually assessed if each fly was alive or dead, and if the latter, whether it had sporulated. Henceforth, we will use the term 'zombies' as a shorthand for *E. muscae*-exposed flies that perform fungus-induced behaviors before dying and sporulating. Sporulated flies were retroactively declared 'zombies' and living flies 'survivors.' Dead flies without signs of sporulation were excluded from further analysis. The time of zombie deaths was manually determined by the time of the last movement (*Figure 1D*). As expected, wild-type flies killed by *E. muscae* tended to die in the

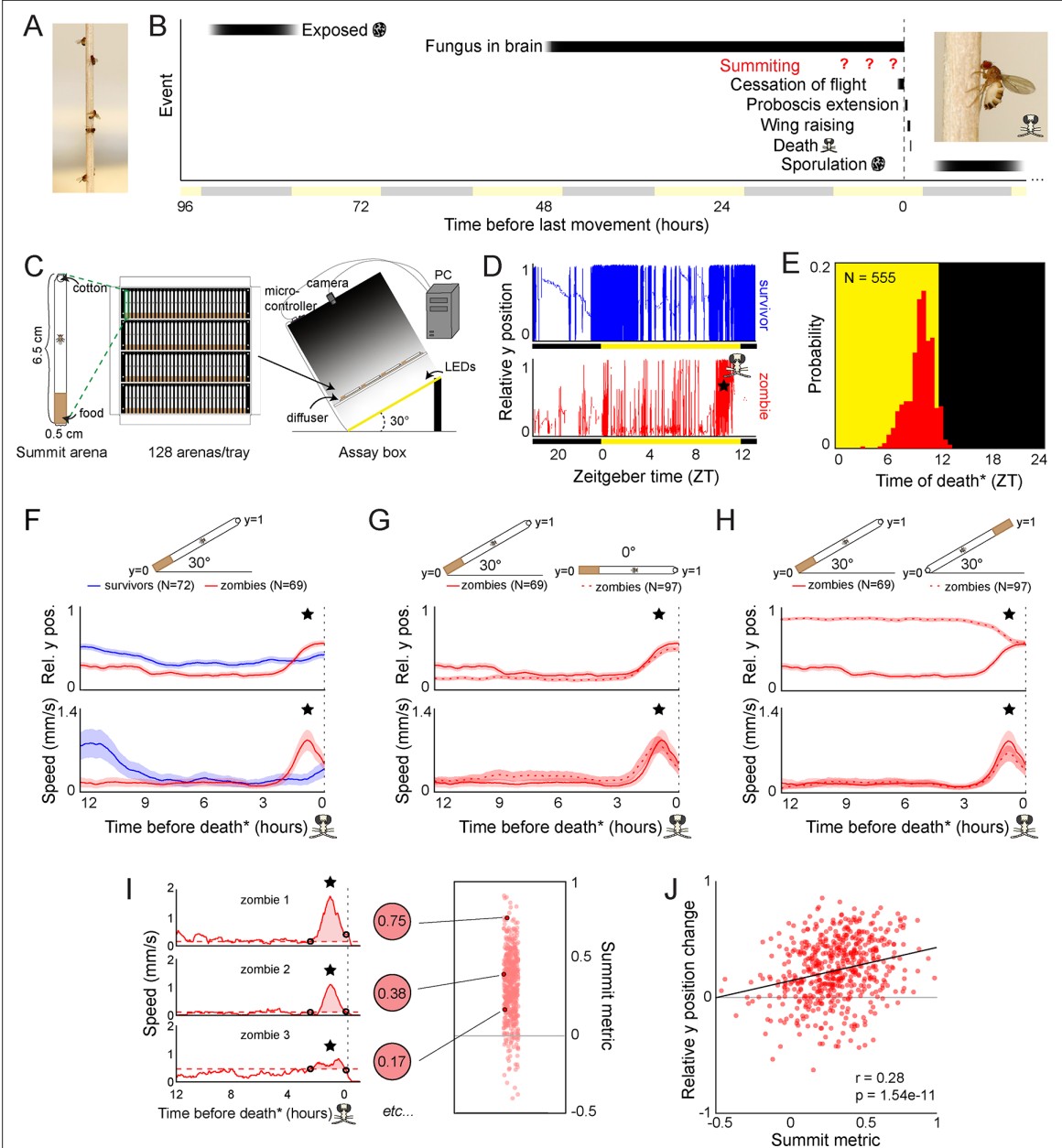

**Figure 1.** Behavioral signature of *E.muscae*-induced summiting in wild-type flies. (**A**) *E. muscae*-killed fruit flies that summited on a wooden dowel prior to death. (**B**) Timeline of events relative to an *E. musace*-infected fly's last movement (dashed line). See (***Elya et al., 2018***; ***Krasnoff et al., 1995***). (**C**) Summiting assay schematic. (**D**) Example y position data for a typical survivor fly (top) and zombie (bottom). X-axis is Zeitgeber time (ZT), hours since lights were turned on. The fly 'skull' indicates the manually-annotated time of zombie death (see Methods). Black and yellow bars indicate the state of visible illumination. (**E**) Distribution of time of death for Canton-S flies killed by *E. muscae*. Background color indicates the state of visible illumination. (**F**) Mean y position (middle) and mean speed (bottom) of survivor flies (blue) and zombie flies (red) housed in arenas angled at 30° with food at the bottom (schematic at top) during the 12 hr preceding the time of death. Here and in all other panels, shaded regions are +/− 1 standard error of the mean. Time of death for zombies was manually determined as the time of the last movement from the y position trace. Survivors did not die but were assigned fictive times of death from the distribution of zombie death times for comparability (see Methods). (**G**) As in (**F**), but comparing zombies in standard arenas (30° with respect to gravity, same data as (**F**); solid lines) to zombies in flat arenas (0°; dashed lines). (**H**) As in (**F**) and (**G**), but comparing zombies in standard arenas (food at the bottom, same data as (**F**); solid lines) to zombies in arenas with food at the top (dashed lines). (**I**) Speed versus time for three examples Canton-S zombies (left) and their corresponding summit metrics (middle) outlined in black (right) amidst all Canton-S summit metrics (N=555, right). Black circles denote the window of summiting behavior as determined from the mean behavior of Canton-S zombie flies. Dashed red line indicates the mean speed in the hour preceding summiting (baseline speed). Summit metric is calculated as the integral of speed

*Figure 1 continued on next page*

**Figure 1 continued**

minus baseline in the summing window (shaded region). (**J**) Relative y position change versus summit metric for Canton-S zombies (N=555). Points are individual flies. Linear regression line in black; Pearson's correlation r & p-value (upper left).

The online version of this article includes the following figure supplement(s) for figure 1:

**Figure supplement 1.** *E.muscae*-infected flies die at specific times of day in the absence of proximal lighting cues.

**Figure supplement 2.** Additional features of summing behavior in the custom behavior assay.

evening (mean death time = ZT9:50 *Figure 1E*), but there was variability in the timing of death. 90% of all deaths occurred between ZT7 and ZT12. E. muscae-exposed flies continued to die at specific times of the day even in complete darkness (*Figure 1—figure supplement 1*), suggesting that the timing of death is under circadian control.

## A burst of locomotion before death is a key signature of *E. muscae*-induced summiting

With our assay in its standard configuration (angled 30° with respect to gravity, food at the bottom), *E. muscae*-exposed survivors and zombies exhibited significantly different time-varying patterns in the mean vertical position and mean speed in the final 12 hr before death (*Figure 1F*; survivors were randomly assigned a fictive time of death to enable this comparison). Survivor flies typically resided close to the center of the summit arena throughout tracking. In contrast, the average position of the zombie fly was near the bottom of the arena until approximately 2.5 hr before death when the average elevation increased, ultimately surpassing that of survivors. The difference between zombies and survivors in average speed over time was even more striking. Zombies maintained a low average speed (0.18 mm/s) until ~2.5 hr before death when it increased substantially, peaking at 0.87 mm/s approximately one hour prior to death. In contrast, survivors exhibited high mean speed (~0.8 mm/s)~12 hr prior to the end of the experiment and a small increase in mean activity (0.22 m/s)~2 hr after the burst of zombie activity. These peaks of survivor activity correspond to the crepuscular peaks of activity expected in healthy flies.

Surprisingly, the average 'elevation' and speed trajectories of zombie flies did not change in the absence of a gravitactic gradient (i.e. when the arena was laid flat, and the food was designated as the 'bottom' of the arena) (*Figure 1G*). Flies resided near the food and exhibited low average speed (0.19 mm/s) until ~2.5 hr prior to death, when speed peaked at 0.8 mm/s and flies had a mean position near the middle of the chamber. These patterns were largely statistically indistinguishable from those of the 30° experiment. When the chamber was angled at 30°, but with food at the top, average y position trends were essentially flipped, with flies on average residing near the top of the chamber until 2.5 hr prior to death, at which point they moved downward (*Figure 1H*). Notably, speed trends were statistically indistinguishable in this new configuration: flies still exhibited low average speed (0.15 mm/s) until ~2.5 hr prior to death when they exhibited a marked increase in speed peaking at 0.66 mm/s ~1 hr prior to death.

The burst of speed prior to death in zombie flies was specific to how they died. Unexposed flies that were killed by starvation (*Figure 1—figure supplement 2A*) or desiccation (*Figure 1—figure supplement 2B*) did not exhibit a burst of speed prior to death. In both cases, flies maintained a high average speed at 12 hr before death (2.2 mm/s and 2.9 mm/s, respectively) with the average speed of starved flies gradually declining over ~5 hr before death. The mean speed of desiccated flies gradually increased from 12 to ~3 hr before death, peaking at 4.85 mm/s, then exhibited a steady decline until death. Unlike zombie flies, starved or desiccated unexposed flies did not die at a specific time of day (*Figure 1—figure supplement 2C*, S1D). These experiments suggest that an increase in speed ~2.5 hr before death and dying at specific times are signatures of *E. muscae* mortality.

Average zombie y position appeared to be dictated by the location of food in our assay. Zombie flies began to reside closer to the food than survivors starting ~24 hr prior to death in the food-at-the-top configuration (*Figure 1—figure supplement 2E*). This behavior was dependent on the nutritive content of the food. When given a choice between sugar-containing and sugarless agar in a 0° assay, zombie flies tended to reside near the sugar-containing media before moving away ~2.5 hr prior to death (*Figure 1—figure supplement 2F*). Providing food within the last 24 hr was necessary for the pre-death burst of locomotion: flies that were housed on sugarless media starting the day

prior to death failed to exhibit a pre-death burst of locomotion (*Figure 1—figure supplement 2G*) though still died with the expected circadian timing (*Figure 1—figure supplement 2H*). These results suggest that flies are likely starving by late infection (*Elya et al., 2018*) and need access to sustenance to exhibit a final burst of locomotion during summiting.

A burst of locomotion will move flies, on average, away from the closed end of an arena, a consequence of that boundary condition. We were curious about what would happen if flies were residing at food in the middle of an arena at the onset of summiting. We lengthened the arena and situated the food in the middle. As expected, in 0° arenas, zombie flies remained on average centered on the food prior to death (*Figure 1—figure supplement 2I*). However, in 30° arenas, zombie flies moved on average slightly upward at the end of life (*Figure 1—figure supplement 2I*). The distance that flies traveled during summiting did not differ between arenas angled at 0–30° (*Figure 1—figure supplement 2J and K*), indicating that the net upward motion of summiting in this condition could not be attributed to differences in activity.

Taken together, these experiments reveal a burst of speed in the final 2.5 hr before death as a key signature of *E. muscae*-induced summiting in our assay. We devised a simple metric, the summit metric (SM), to quantify the 'summity-ness' of individual flies. SM is calculated as the integral of baseline-corrected speed over the summiting window. Three example speed traces for Canton-S flies and their corresponding SM values are shown in *Figure 1I*. As expected, there was a weak, positive correlation across individual flies between SM and change in y-position over summiting (*Figure 1J*). Comparing SM values across over 400 male and female Canton-S flies, we observed that, on average, males are moderately more 'summity' (have 18% higher SM values) than females (*Figure 1—figure supplement 2L and M*). However, this difference is dwarfed by interindividual variation in summiting, and since *E. muscae* infects both males and females in the wild, we opted to use mixed-sex experimental groups in subsequent experiments.

## Summiting behavior requires host circadian and neurosecretory pathways

With the understanding that a burst of activity shortly before death is the signature of summiting in this assay, we performed a screen to identify circuit and genetic components mediating summiting in the host fly. We adopted a candidate approach, but cast a wide net for neurons and genes involved in neuromodulation or previously implicated in arousal and gravitaxis (*Figure 2A–C*, *Supplementary file 1*). To disrupt neurons, we drove the expression of tetanus toxin (TNT-E; a vesicle release blocker; *Keller et al., 2002*) using 103 different Gal4 drivers (*Supplementary file 1*). The effect size of each of these perturbations on summiting behavior was estimated relative to a common heterozygous control (*UAS-TNT-E/+*), and confidence intervals on each effect size were calculated by bootstrapping (*Figure 2B*). Similarly, we screened 101 lines targeting candidate genes, either by pan-neuronally reducing their expression via RNAi (i.e. driving CNS-wide expression of short hairpin RNAs targeting the desired gene) or testing mutant alleles (*Supplementary file 1*). Again, effect sizes were estimated by comparing each line's summiting metric to common control genotypes, for pan-neuronal RNAi, the heterozygous pan-neuronal driver (*R57C10-Gal4/+*); for mutants, wild-type (CantonS) control (*Figure 2C*). Genotype details and our rationales for including each line in the screen are given in *Supplementary files 1 and 2*. In both the circuit and genetic screens we observed a range of effects on summiting from extreme impairment of the behavior (effect size –1) to rare amplification of summiting (effect size >0). Most perturbations had effects that were not statistically distinguishable from zero.

Our manipulations targeted low-level biological elements (single genes and sparse neuronal expression patterns, as well as some broad expression patterns). To determine what higher-level systems might be *E. muscae*'s target, we looked for enrichment of large effect sizes in the genes (or circuit elements) involved in the same higher-level functions (or brain regions). We binned the behavioral data for each reagent type (i.e. neurons or genes) into quintiles according to effect size, looked at annotation frequencies across these bins, and noted annotations that occurred in a given quintile more frequently than expected by chance (*Figure 3D and E*). We found that neurons in the antennal mechanosensory and motor center (AMMC), subesophageal ganglion (SOG), circadian system, and pars intercerebralis (PI) were overrepresented in the quintile of most negative effect size (*Figure 3D*). Underscoring the potential importance of the PI, we observed that many of the neurons of large effect in the AMMC, SOG, and circadian system also innervated the PI (*Figure 3D* - pink overlay). In a similar

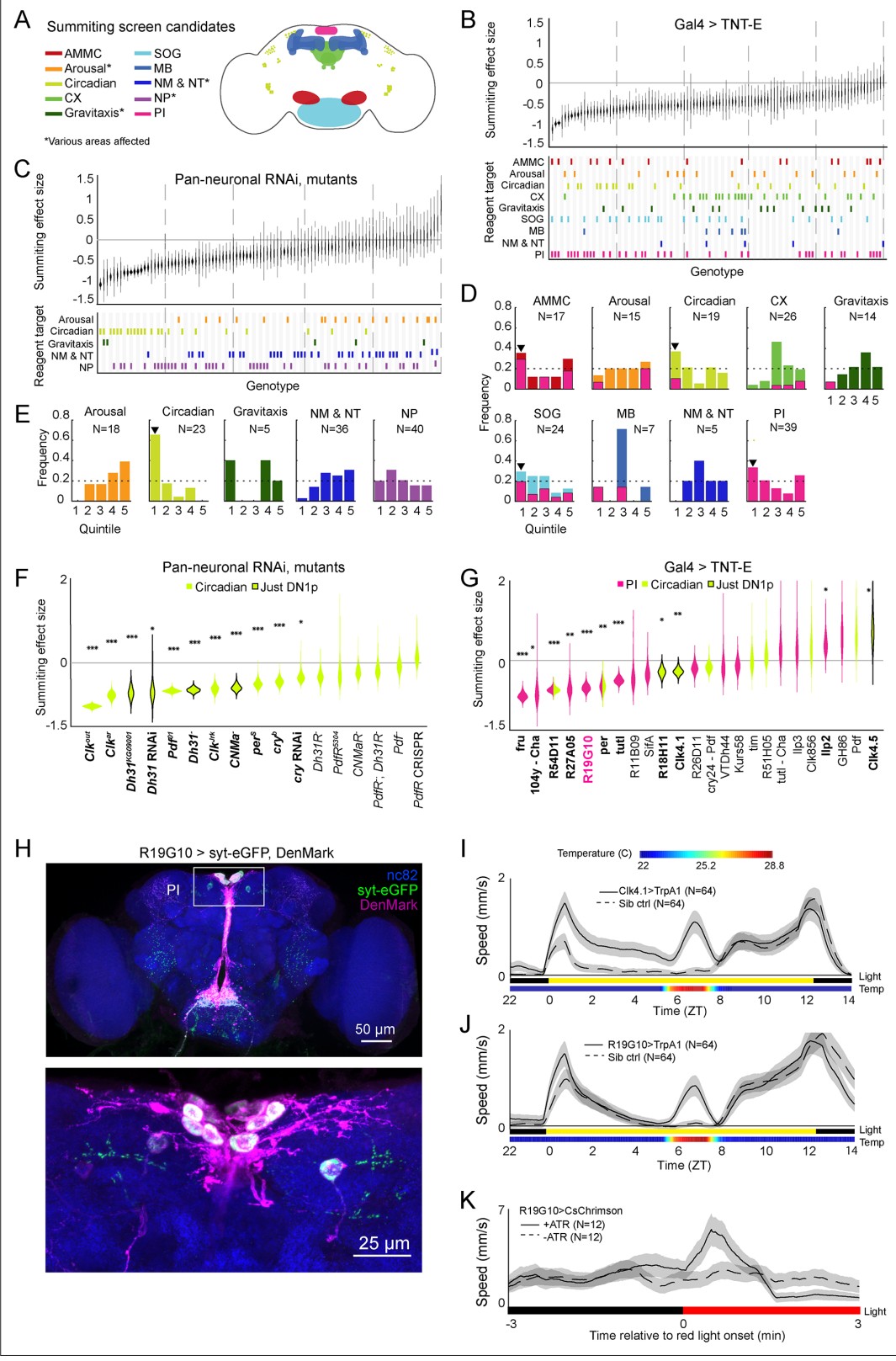

**Figure 2.** Identification of host circuits and genetic components involved in summiting behavior. (**A**) Regions and pathways targeted in the candidate screen. AMMC = antennal mechanosensory and motor center; CX = central complex; SOG = subesophageal ganglion; MB = mushroom body; NM & NT = neuromodulator or neurotransmitter; NP = neuropeptide; PI = pars intercerebralis. (**B** and **C**) Effects of neuronal disruption (B;

*Figure 2 continued*

12<N<111, median N=35) or gene knockdown or mutagenesis (C; 10<N<182, median N=46) on summiting. Above: Summiting effect size estimate distributions as estimated by bootstrapping. Experimental groups are ordered by mean effect (negative to positive). Below: gene function and brain region annotations associated with each screened reagent. See **Supplementary file 1** for genotype and annotation details. Solid gray line indicates an effect size of zero. Dashed vertical lines separate ranked data into quintiles. (**D** and **E**) Frequency of annotations by quintile for (**B**) and (**C**), respectively. The number of lines screened (**N**) is indicated for each annotation. Dashed line indicates the frequency of annotation expected from a null, uniform distribution. Black arrowheads highlight annotations that are overrepresented in the first quintile. For (**D**), pink overlays indicate the portion of line annotations that are co-annotated for expression in the PI. (**F** and **G**) Summiting effect size estimate distributions of disrupting specific circadian genes (F; 19<N<182, median N=62) or circadian and/or PI neurons (**G**; 11<N<111, median N=46) compared to genotype-matched controls. Lines are ordered by effect size. Pink indicates Gal4 expression in the PI, lime circadian Gal4 lines and genes, and black outlines expression only in DN1ps. Asterisks indicate statistically significant effects on summiting behavior by a two-tailed t-test (*=p<0.05; **=p<0.01; *** p<0.001). R19G10 is highlighted in pink to emphasize its subsequent use as the main PI reagent. See **Supplementary file 2** for genotypes and matched controls. (**H**) Maximum z-projections of brains showing pre- (synaptotagmin; syt-eGFP) and post- (DenMark) synaptic compartments of R19G10 neurons. Bruchpilot (nc-82) staining (blue) visualizes neuropil. Above: brain imaged from anterior. Below: another brain, imaged from the posterior. (**I** and **J**) Mean speed of unexposed flies vs time for Clk4.1>TrpA1 and R19G10>TrpA1 genotypes and sibling controls, respectively. Shaded regions are +/− 1 standard error of the mean. Bars along the x-axis indicate the state of visible illumination (above) and temperature (below). (**K**) Red light onset-triggered mean speed across flies of unexposed R19G10>CsChrimson flies versus time. All trans retinal (ATR) indicates control flies not fed CsChrimson cofactor. Shaded regions are +/− 1 standard error of the mean. Bar along the x-axis indicates lighting conditions (black: darkness, red: red-light illumination).

The online version of this article includes the following figure supplement(s) for figure 2:

**Figure supplement 1.** Additional experiments assessing summiting after clock neuron and R19G10 disruption.

**Figure supplement 2.** Additional experiments assessing the sufficiency of DN1p and R19G10 neuron activation for increased locomotion.

---

analysis for our genetic manipulations, we saw a clear enrichment for genes expressed in circadian cells (*Figure 2E*). Thus, our screen pointed conspicuously toward roles for the PI and the circadian network in summiting behavior.

With these high-level systems implicated as targets of fungal manipulation, we returned to a granular analysis to determine what specific circuit elements in circadian cells and the PI best recapitulated the high-level effects. We measured the summiting response of an individually tailored genetic control for each circadian gene and PI or circadian circuit element (rather than screen-wide controls), and recalculated the effect size of each perturbation (*Figure 2F and G*). With respect to the circadian experiments, eleven mutants (*Figure 2F*) and four Gal4 lines (*Figure 2G*) showed impaired summiting compared to matched genetic background and/or sibling controls. Three different mutants of Clock (Clk), a gene expressed in all clock cells, showed greatly reduced summiting behavior (62–104%, 3.4e-28<p<7e-8). The cryptochrome gene (cry) encodes a blue light sensor expressed by a subset of circadian neurons that synchronizes the molecular oscillator with environmental lighting cues (*Emery et al., 2000*; *Benito et al., 2008*; *Yoshii et al., 2008*). A cry mutant and a pan-neuronal RNAi knockdown of cry both showed reduced summiting (32%, p=0.018; 45%, p=0.00097, respectively).

We noticed that several of our hits affected a subtype of clock neurons, the group 1 posterior dorsal neurons (DN1ps). DN1ps are a heterogeneous population of neurons numbering approximately 15 cells per brain hemisphere (*Ma et al., 2021*). About half of DN1ps express cry (*Yoshii et al., 2008*). Silencing neurons with two drivers that label many, but not all, of the DN1ps (Clk4.1 and R18H11; *Zhang et al., 2010*; *Kunst et al., 2014*) via TNT-E expression reduced summiting by 24–25% (p=0.005, 0.019; *Figure 2G*, *Figure 2—figure supplement 1B and C*). However, silencing the entire population of DN1p neurons by driving the inward-rectifying potassium channel Kir2.1 (*Baines et al., 2001*) with a pan-DN1p driver had no apparent effect (*Figure 2—figure supplement 1D*) as did silencing neurons labeled by an additional driver previously reported to be expressed in DN1ps (R51H05; *Kunst et al., 2014*). Silencing a sparser population of DN1ps (Clk4.5) with TNT-E led to an increase in summiting (*Figure 2G*). Genetic disruption of two signaling molecules expressed by DN1ps, Diuretic Hormone 31 (*Dh31*) and the neuropeptide CNMamide (*CNMa*), reduced summiting by 59–72% (3e-16<p<0.025;

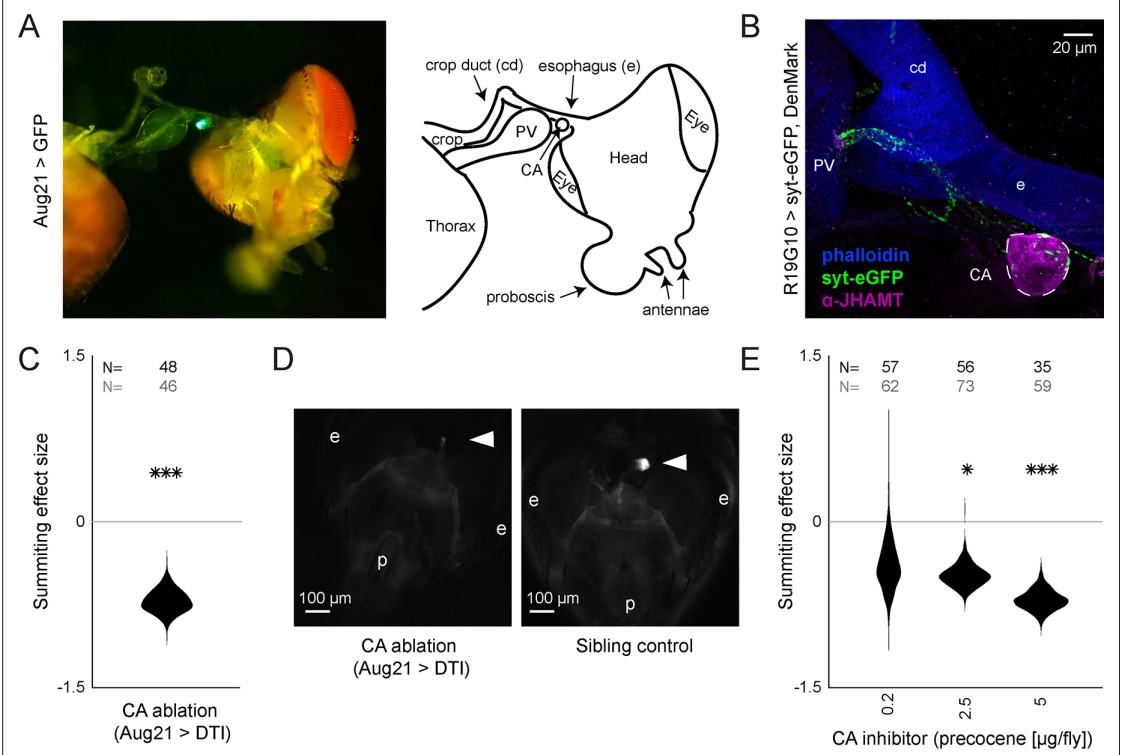

**Figure 3.** R19G10 (PI-CA) neurons project to the corpora allata, which are required for summiting behavior. (**A**) Left: Composite micrograph of dissected Aug21>GFP fly, showing GFP fluorescence in the corpora allata (CA) overlaid on bright field image. Right: Diagram of A with anatomical features labeled. PV = proventriculus. (**B**) Representative confocal micrograph of immunostained RC from an R19G10>syt-eGFP, DenMark fly. Synaptic terminals are visible as green puncta, including in the CA. Magenta is anti-JHAMT and marks the CA. Blue phalloidin counterstain marks actin. Labels as in A. (**C**) Summiting effect size estimate distribution of ablating the CA with diphtheria toxin (DTI). Effect size is calculated relative to effector-less sibling controls. (**D**) Representative micrographs of CA-ablated and effector-less, sibling, temperature-matched control flies (additional examples in *Figure 3—figure supplement 1D*). White arrows indicate the expected location of CA. e = eye, p = proboscis. (**E**) Summiting effect size estimate distributions of various concentrations of the CA-ablating drug precocene. Effect size is calculated relative to vehicle (acetone) control. For (**C** and **E**), effect sizes were estimated as in *Figure 2*; asterisks indicate statistically significant effects (*=p<0.05; **=p<0.01; ***p<0.001) by two-tailed t-test. Sample sizes of experimental and control experiments are given in black and gray, respectively.

The online version of this article includes the following figure supplement(s) for figure 3:

**Figure supplement 1.** Supporting data for juvenile hormone involvement in summiting.

**Figure supplement 2.** Additional experiments examining juvenile hormone involvement in summiting.

*Figure 2F*). However, flies mutant for the receptors that recognize these molecules (Dh31R and PdfR for Dh31; CNMaR for CNMa) did not show significantly impaired summiting (0.054<p<0.3), though Dh31R came close with a 33% impairment at p=0.054. Taken together, these results implicate DN1ps as mediating fungal manipulation while also revealing fine-scale complexity, as activity in some DN1ps, but not others, is required for full summiting.

DN1p activity is influenced by a class of pacemaker neurons called small ventrolateral neurons (sLNvs) (*Zhang et al., 2010*) that express the neuropeptide Pigment-dispersing factor (Pdf; *Helfrich-Förster and Homberg, 1993*; *Renn et al., 1999*). While one *Pdf* mutant (*Pdf*$^{01}$) exhibited a large, significant reduction in summiting (67%; p=1.8e-16; *Figure 2F*), we saw no effect with another mutant whose *Pdf* locus was completely replaced (*Pdf*). We also did not observe a significant decrease in summiting in Pdf receptor (PdfR) mutants (0.3<p<0.38). Disrupting sLNVs by expressing TNT-E, channel Kir2.1, or pro-apoptotic protein hid (*Grether et al., 1995*) also had no effect on summiting (*Figure 2—figure supplement 1D*, E). This suggests that the main population of clock neurons upstream of DN1ps is irrelevant for summiting.

DN1ps send some processes medially, with presynaptic sites occurring at or near the PI (*Reinhard et al., 2022b, Chatterjee et al., 2018*). We tested the effect on summiting of silencing neurons in the PI using 16 different Gal4 drivers. Of these, seven produced significant reductions in summiting

ranging from 44-79% (2.6e-9<p<0.02; *Figure 2G*). While some of these drivers were quite broad (such as *fru-Gal4*), others were quite sparse and specific to the PI, including *R19G10-Gal4* which is expressed in ~12 neurons (all but two of which are in the PI; *Figure 2H*). Silencing R19G10 neurons reduced summing by 60% (p=2.4e-8; *Figure 2G*, *Figure 2—figure supplement 1A*). Given the sparseness of this Gal4 driver and the large effect on summing of expressing TNT-E with it, we focused on its PI neurons as the likely target of manipulation in this neuropil.

We next tested whether the ectopic activation of DN1ps or R19G10 neurons could drive 'summing' in flies that had never been exposed to *E. muscae*. We expressed a thermosensitive cation channel TrpA1 (*Hamada et al., 2008*) using *Clk4.1-Gal4* (to target DN1ps) or *R19G10-Gal4* (to target the PI) in flies unexposed to *E. muscae*. We conducted a 20 hr summing assay with these flies, raising the temperature from 22–28°C, for 2 hr (ZT6-8) between the flies' daily circadian activity peaks that occur at the light-dark transitions (ZT0 and ZT12). Activating either DN1p or R19G10 neurons in this way led to a 28.7-fold or 9.7-fold increase in mean fly speed compared to sibling controls, respectively (*Figure 2I* and *Figure 2J*). This effect was significant across both males and females, though the effect was smaller in females for both experiments (*Figure 2—figure supplement 2A*, *Figure 2—figure supplement 1B, C, D*). As another test of the sufficiency of activating R19G10 neurons to induce summing-like behavior, we expressed the optogenetic reagent CsChrimson (*Klapoetke et al., 2014*) in these cells. We ran these flies in a modified summing assay with alternating periods of 3 min of darkness and red light. R19G10>CsChrimson flies fed all-trans retinal (ATR), the CsChrimson cofactor, exhibited a burst of mean speed for the first 60 s after light onset (*Figure 2K*, *Figure 2—figure supplement 2G*) and suppressed walking speed for the last 90 s of light stimulation, perhaps due to depolarization block (*Herman et al., 2014*). In contrast, the control fly speed remained roughly constant throughout. The higher mean walking speed reflects a higher portion of flies walking after light onset (*Figure 2—figure supplement 2E and F*). Thus, ectopically activating DN1Ps and R19G10 neurons appear to robustly induce a summing-like increase in activity in flies unexposed to the fungus.

## The corpora allata are post-synaptic to R19G10 (PI-CA) neurons and necessary for summing

In insects, pars intercerebralis neurons often project to the neurohemal organs of the retrocerebral complex (RC) (*Carrow et al., 1984*; *de Velasco et al., 2007*; *Hartenstein, 2006*; *Pipa, 1978*; *Rüegg et al., 1983*; *Siegmund and Korge, 2001*). We suspected this might be the case for R19G10 neurons. The RC in *Drosophila* consists of two pairs of fused neurohemal organs: the corpora cardiaca (CC) and the corpora allata (CA) (*Nässel, 2002*), the sole sites of adipokinetic hormone (Akh) (*Noyes et al., 1995*) and juvenile hormone (JH) synthesis, respectively (*Klowden, 2008*). Akh null mutants exhibited intact summing (*Figure 3—figure supplement 1A*), so we focused on potential R19G10 connections to the CA. We expressed the presynaptic marker synaptotagmin-GFP in R19G10 neurons and co-stained dissected brain-RC complexes for the CA-specific marker JH methyltransferase (JHMAT) (*Niwa et al., 2008*, *Figure 3—figure supplement 2A*). We observed R19G10 presynaptic terminals at the CA (*Figure 3B*), so we named R19G10 neurons 'PI-CA' neurons to reflect this connectivity (Following the convention of *Wolff and Rubin, 2018*, the letters before the dash indicate the postsynaptic compartment, the letters after the presynaptic compartment).

To test if the CA was required for summing, we turned to genetic ablation. First, we drove the expression of a Nuclear inhibitor of Protein Phosphatase type 1 (NiPP1) with a driver that targets the CA (Aug21; *Siegmund and Korge, 2001*). NiPP1 overexpression causes cell-autonomous lethality in a variety of cell types (*Parker et al., 2002*) and has been previously used to ablate the CA in adult flies (*Yamamoto et al., 2013*). Aug21 >NiPP1 animals showed reduced summing by 60% (p=2.7e$^{-5}$) (*Figure 3—figure supplement 1B*), but immunohistochemistry showed that the degree of CA ablation varied by the animal (*Figure 3—figure supplement 1C*). In a second ablation approach, we used a temperature-sensitive Gal80 (*McGuire et al., 2004*) to repress the expression of diphtheria toxin (DTI) driven by Aug21 until flies had reached wandering 3rd instar (*Bilen et al., 2013*). Tub-Gal80(ts), Aug21 >DTI flies housed at the restrictive temperature also showed reduced summing 72% (p=1.1e$^{-5}$, *Figure 3C*) and were confirmed by microscopy to have either greatly reduced or absent CA (*Figure 3D*, *Figure 3—figure supplement 1D*).

We used pharmacology as a complementary approach to confirm the role of the CA in summing. First, we blocked the production of JH by feeding flies fluvastatin, a compound that targets the JH

synthesis pathway by inhibiting 3-hydroxy-3-methylglutaryl coenzyme A (HMG-coA) (*Figure 3—figure supplement 2A*, *Debernard et al., 1994*). Flies fed with fluvastatin at 72 hr after exposure to the fungus showed severely reduced summiting (110% (p=3.1e-11) *Figure 3—figure supplement 2B*). However, these flies released very few spores compared to untreated zombies and died at atypical times (after sunset; *Figure 3—figure supplement 2C*). This observation led us to suspect that fluvastatin was impairing fungal growth. A series of experiments confirmed that feeding fluvastatin to flies well in advance of summiting (24 hr post-exposure) led to the premature death of infected flies (*Figure 3—figure supplement 2D*) and abolished the circadian timing of death (*Figure 3—figure supplement 2E*). Altogether, these data indicate that while fluvastatin disrupted summiting, that effect was likely due to disruption of fungal growth. We next turned to precocene (*Bowers, 1981*), a natural product that reduces JH titers per *Amsalem et al., 2014* by inducing CA necrosis (*Pratt et al., 1980*). Applying 2.5 or 5 µg of precocene to exposed flies led to a 47% and 70% reduction of summiting behavior (p=0.001 and 6e-6, respectively) (*Figure 3D*). Increased doses of precocene led to more off-target deaths in both exposed and control flies, suggesting that precocene toxicity is fungus-independent (*Figure 3—figure supplement 2F*). Precocene treatment did not alter the timing of death by *E. muscae* (*Figure 3—figure supplement 2G*).

We wondered if we could enhance summiting by dosing flies with the juvenile hormone analog (JHA) methoprene (*Cerf and Georghiou, 1972*). We topically applied methoprene at two different concentrations (2.5 and 5 µg). Surprisingly, these treatments led to a statistically non-significant reduction of summiting by 22.2 and 30.9% (p=0.13, 0.09, respectively; *Figure 3—figure supplement 2H*). We also tried to rescue the effects of precocene, either by co-application of methoprene (2.5 µg) or by feeding flies another JHA, pyriproxyfen (5 µg) (*Riddiford and Ashburner, 1991*). Neither of these treatments rescued the effects of precocene treatment (*Figure 3—figure supplement 2I*). Overall, these results indicate that CA function is necessary for summiting, but that supplementing flies with JHA is not sufficient to elicit this behavior. It could be that the acute release of JH is critical for driving summiting or that the CA produces a specific cocktail of juvenile hormones that are not well mimicked by our drug treatments.

## A real-time, automated classifier for summiting behavior

Having identified a neurohormonal circuit that is required in the fly host for summiting, we next sought to investigate how the fungus gains access to this target and manipulates it to induce summiting. We reasoned that there may be physiological and anatomical differences between summiting and non-summiting flies that reflect causal mechanisms on the fungal side. These correlates likely degrade by the time the fly dies, so real-time identification of summiting flies is needed. We developed an automated classifier to identify summiting flies and alert an experimenter real-time. Our ground-truth dataset for training the classifier was made from a dataset of ~20 hr recordings of speed and y-position from 1306 *E. muscae*-exposed Canton-S flies, 345 of which were zombies. Each of the zombie traces was manually annotated with the time of summiting onset and time of death. Based on these timepoints, every frame was labeled as 'pre-summiting,' 'during summiting,' or 'post-summiting.' Every frame from survivor flies was labeled as 'never summiting' to reflect that they would not summit for the period of observation (*Figure 4A*).

From each fly trajectory, we selected 200 random time points (for 261,200 total training data points) and from each generated a 61-element feature vector consisting of the current time, recent y-position and speed values, and past values of those measures log-spaced back to the start of the experiment (*Figure 4B*). Paired with each feature vector was the associated summiting label. We trained a random forest classifier with 75% of the data and validated performance with the remaining 25% (*Figure 4SA*). Of the variables in the feature vector, current time, initial y position, and initial and current speed were the most influential factors in classification (*Figure 4C*). The distributions of these variables by summiting labels made sense: summiting labels were most abundant in the evening, at low y positions prior to summiting, and at higher speeds during summiting versus pre-summiting (*Figure 4D*). The classifier had a middling recall (56%) but high precision (88%) on a novel test dataset collected separately from the training and validation data (*Figure 4E*).

We next focused on how to use the classifier to flag summiting flies for upcoming real-time experiments. A rule wherein a fly was flagged as summiting when its during-summiting class probability exceeded its never-summiting class probability for three consecutive classifications (spanning 8 min)

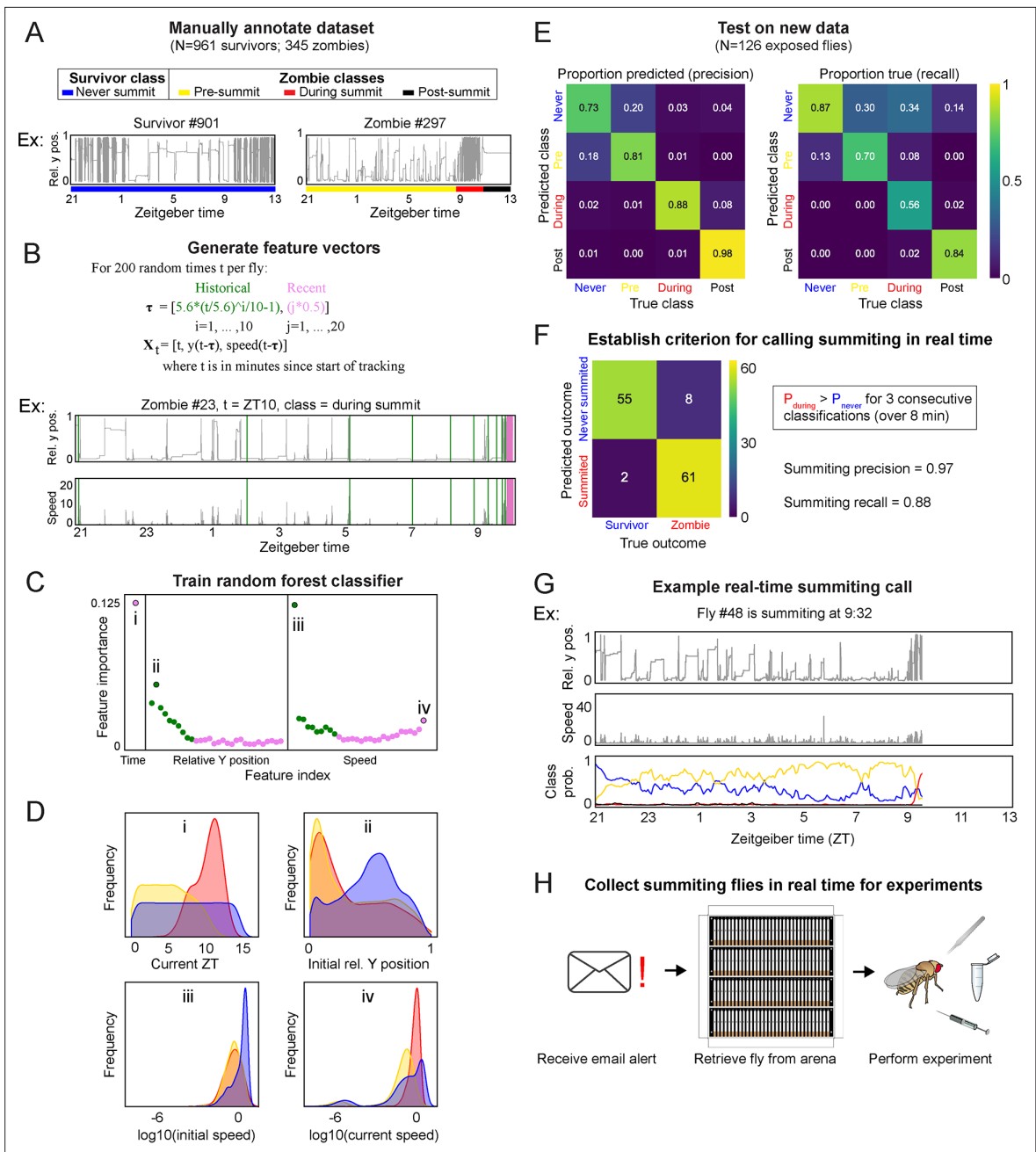

**Figure 4.** A random-forest classifier (RFC) for identifying summiting flies in real-time. (**A**) Top: classes learned by the classifier for zombies were pre-summiting=prior to the onset of summiting (yellow), during summiting = after the onset of summiting but before the time of death (red), and post-summiting=after the time of death (black). For survivors, there was one class, never-summiting (blue). Bottom: annotations of these classes on example y position trajectories from a survivor (left) and zombie (right). (**B**) Feature vectors ($X_t$) generated for 200 random time points (t) for each fly. Vertical green and pink lines in the example trajectory below indicate the historical (green) and recent (pink) values selected for the feature vector. (**C**) Feature importance for classification of the 61 input variables. Roman numerals correspond to plots in subsequent panels. (**D**) Distributions of important feature variables, visualized with kernel density estimation, across never summiting (blue), pre-summiting (yellow), and summiting (red) classes within the training dataset. (**E**) Confusion matrices for precision (left) and recall (right) performance of the classifier on the test dataset. (**F**) Confusion matrix for the survivor and zombie outcomes after implementing the real-time zombie-calling criterion. (**G**) Example real-time behavior and class probability trajectories for a zombie fly, ending on the frame when it was called as a zombie. (**H**) Summarized experimental workflow using the real-time classifier.

The online version of this article includes the following figure supplement(s) for figure 4:

**Figure supplement 1.** Development of a real-time random forest classifier for summiting behavior.

had high precision (97%) and recall (88%) (*Figure 4F*) in simulations of real-time experiments with ground truth labels (*Figure 4G*). Flies that never passed this threshold were flagged as 'survivors.' Finally, we configured our fly-tracking software to run the classifier concurrently and email the experimenter when a summiting fly was flagged. Thus, we had a convenient, high-accuracy tool for experiments requiring real-time identification of summiting flies (*Figure 4H*).

## During summiting, *E. muscae* cells are adjacent to the PI and the PI-CA pathway appears intact

Using the real-time classifier, we assessed the distribution of *E. muscae* cells within the brains of summiting flies. We imaged the brains of summiting flies expressing RFP-tagged histones in all cells, counterstained with Hoechst to label all nuclei (fly and fungi). We observed a consistent pattern of *E. muscae* occupancy in the brain, with a plurality of fungal cells (27–41%) in the superior medial protocerebrum (SMP), the region that contains the PI. Notably, there were very few fungal cells in the central complex, a premotor region (*Figure 5A–C*). Phalloidin staining suggested that each fungal cell sat in a 'hole' in the neuropil (*Figure 5A*). The dense occupancy of the SMP is established as early as 72 hr after exposure (*Figure 5—figure supplement 1A*).

To determine if the numerous *E. muscae* cells in the SMP were grossly disrupting PI-CA neurons, we imaged summiting animals expressing membrane-bound GFP in PI-CA neurons and compared them with uninfected controls. Despite the abundance of *E. muscae* cells in the SMP of summiting animals, the overall morphology of PI-CA neurons in summiting animals appeared normal (*Figure 5D*). There was no difference in the number of PI-CA cell bodies between summiting flies and unexposed controls (*Figure 5E*). In contrast, freshly killed cadavers had on average 60% fewer PI-adjacent cell bodies compared to summiting or non-summiting controls ($0.0055 < p < 0.0029$) (*Figure 5E*).

Fungal cells appear to displace host brain tissue, sitting in 'holes' visible in actin-binding phalloidin counterstains (*Figure 5A and D* bottom middle). Consistently, the distribution of holes across brain regions (*Figure 5F*) was indistinguishable from the distribution of fungal nuclei (*Figure 5C*). Occasionally, we observed holes within the axon bundle of PI-CA neurons (*Figure 5—figure supplement 1B*), but there was no indication of broken axons. Our interpretation is that during summiting, fungal cells displace neuropil without substantially consuming neural tissue or severing neural connections. This is consistent with the logic of zombie manipulation: *E. muscae* only consumes host tissues once they have served their purpose in aiding fungal dispersal.

While the brain is largely intact in summiting, this is not the case for organs in the abdomen, which are essentially obliterated in summiting flies (*Figure 5G–H*, *Figure 5—figure supplement 1E*). The state of the abdominal organs is striking considering that these flies walk apparently normally. *E. muscae* in the abdomen of summiting flies adopted a spherical morphology distinct from their irregular protoplastic form before summiting, even as the interstices of the abdomen are packed with fungal cells (*Figure 5G*). *E. muscae* cells in the brain of summiting flies retain the appearance of pre-summiting hemolymph-bound cells (*Figure 5G* insets). The CA resides in the thorax adjacent to the esophagus and proventriculus. We wondered if these tissues might be degraded like the abdominal organs in summiting flies. We used the classifier to collect summiting and non-summiting Aug21 >GFP animals and found that the CA was consistently present in summiting flies (as well as controls) (*Figure 5I*, *Figure 5—figure supplement 1F*). Overall, the preservation of the CA during summiting suggests that its function is needed to mediate summiting behavior.

## Evidence for the metabolic induction of summiting behavior

We wondered if *E. muscae*'s invasion of the brain disrupts the fly's blood-brain barrier (BBB). Like vertebrates, flies maintain a BBB that restricts the diffusion of compounds circulating in the hemolymph into nervous tissue (*Hindle and Bainton, 2014*). We assayed the integrity of the BBB of flies by injecting flies with Rhodamine B (RhoB), a fluorescent compound that is partially BBB-permeable (*Pinsonneault et al., 2011*). When RhoB enters the brain, it can be detected as fluorescence in the pseudopupil, the portion of eye ommatidia oriented toward the observer; high levels of RhoB can be observed as fluorescence across ommatidia ('bright eyes') (*Mayer et al., 2009*). We found that BBB permeability was higher in exposed flies versus controls at 98 hr after exposure (*Figure 6A*). The increased permeability was not restricted to flies with confirmed infection (59% bright eyes), but was broadly observed among flies that had encountered the fungus (85% bright eyes), compared to

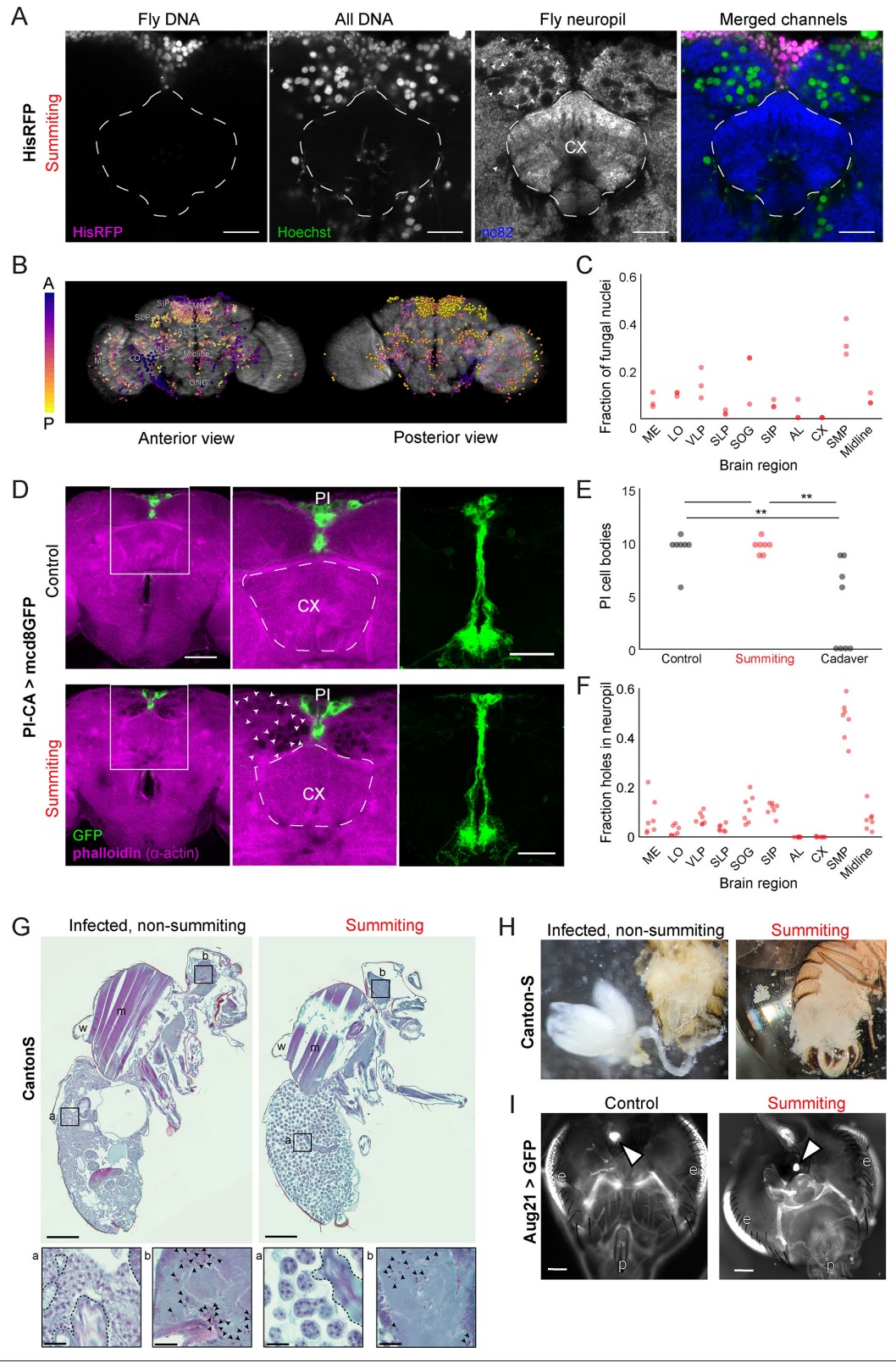

**Figure 5.** *E.muscae* densely occupies the superior medial protocerebrum (SMP) during summiting without apparent degradation of pars intercerebralis to corpora allata (PI-CA) neurons or corpora allata (CA). (**A**) Confocal micrographs of the superior medial protocerebrum (SMP) from summiting His-RFP fly. Non-fly nuclei (Hoechst+, HisRFP−) are large compared to fly neuronal nuclei (Hoechst+, HisRFP+) and sit in 'holes' in the neuropil visible in

*Figure 5 continued on next page*

*Figure 5 continued*

the nc82 counterstain channel. Scale bar is 20 microns. (**B**) Whole brain invasion pattern of *E. muscae* (same brain as A). Nuclei are colored according to depth from anterior (A) to posterior (P). (**C**) Distribution of fungal nuclei across brain regions (N=3). AL = antennal lobe, SIP = superior intermediate protocerebrum, SLP = superior lateral protocerebrum, CX = central complex, VLP = ventrolateral protocerebrum, SOG = subesophageal ganglion, LO = lobula, ME = medulla, midline = cells along the midline of the brain not in any other region. (**D**) Confocal micrographs of PI-CA neurons (green) and phalloidin counterstain (magenta) in control and summing flies. Left: sagittal planes of the central brain. Holes are apparent (in the phalloidin channel) in the SMP of the summing brain, marked by arrowheads in one hemisphere. Holes are absent in CX of summing brains and all control brain regions. Middle: Inset from the left. Right: Maximum z-projections of GFP channel from full brain z-stacks. PI-CA morphology appears the same in summing and control brains. Scale bars are 50 microns. (**E**) Counts of PI-CA cell bodies in control (unexposed), summing, or recently-killed (cadaver) PI-CA >mcd8 GFP flies (** indicates p<0.01 by a two-tailed t-test). (**F**) Distribution of 'holes' across brain regions. Abbreviations as in C. (**G**) Safranin and fast green stained sections of paraffin-embedded Canton-S flies. Left: Infected, non-summing fly (96 hr after exposure to fungus). Right: summing, *E. muscae*-infected fly. a=abdomen, b=brain, w=wing, m=muscle. Scale bars are 200 microns. Insets of the abdomen and brain are shown for each fly below (scale bars are 25 microns). Host tissues are outlined in dashed black; black arrowheads indicate fungal nuclei. (**H**) Micrographs of dissected abdomens of 96-hour post-exposure non-summing (left) and summing (right) female flies. Gut and reproductive organs are still present in the non-summing fly, but are absent in the summing fly. Clumps of spherical fungal cells are visible in the dissection saline of summing but not non-summing fly. (**I**) Fluorescence images of dissected Aug21 >GFP flies. White arrowheads indicate CA. p=proboscis, e=eyes. Scale bars are 100 microns. Additional examples are available in *Figure 5—figure supplement 1F*.

The online version of this article includes the following figure supplement(s) for figure 5:

**Figure supplement 1.** Supporting data for host morphology during *E.muscae* infection.

---

unexposed controls (10% bright eyes) (*Figure 6A*). The proportion of bright-eyed flies was lower at earlier time points following *E. muscae* exposure: 0% after 21 hr, 4.3% after 45 hr, 21.8% after 69 hr (*Figure 6—figure supplement 1*). Our data are consistent with BBB permeability-increasing with time since exposure.

We next used LC-MS metabolomics to compare the molecular composition of hemolymph in summing flies to that of exposed, non-summing flies. We performed this experiment twice: once staging animals by hand based on flightlessness, which occurs during mid to late summing (*Figure 1B*), and a second time using our automated classifier. For each experiment, we collected 1 µL samples of hemolymph bled from a pool of 20 mated females for each of three conditions: (1) healthy (unexposed flies), (2) exposed, non-summing, and (3) summing. Triplicate samples were analyzed when the classifier was employed (*Figure 6—figure supplement 2B*) and duplicate samples were analyzed in the manual experiment (*Figure 6—figure supplement 2C*). We found that 168 compounds were detected in both of these experiments (*Figure 6B*, *Figure 6—figure supplement 2A–C*), with nine compounds enriched and two compounds depleted in summing versus exposed, non-summing flies (*Figure 6—figure supplement 2A*; see *Supplementary file 3* for specific fold-changes and p-values). Many of the compounds could not be identified. These included three compounds that were uniquely detected in summing flies ($C_6H_8N_2O_3$, $C_{14}H_{16}N_6O_7$, and $C_{12}H_{19}N_2PS$) (*Figure 6B*). Three additional compounds (molecular weights 276.08, 179.08, and 429.15 Da) were significantly greater in summing versus exposed, non-summing flies (*Figure 6—figure supplement 2A*, *Supplementary file 3*). Similarly, one compound of molecular weight 451.27 Da was significantly depleted in summing flies (*Figure 6—figure supplement 2A*, *Supplementary file 3*).

Seventy-two compounds could be putatively identified. Cytosine was undetectable in the hemolymph of unexposed flies, but present in both exposed, non-summing, and summing exposed flies (*Figure 6B*, *Figure 6—figure supplement 2A*). Cytosine was significantly enriched in summing versus exposed, non-summing exposed flies (*Figure 6B*, *Figure 6—figure supplement 2A*, *Supplementary file 3*). Ergothioneine, an amino acid produced by some plants and microbes, including fungi (*Borodina et al., 2020*), was only detected in *E. muscae*-exposed animals (*Figure 6—figure supplement 2A*), but did not appear to vary between summing and exposed, non-summing flies (*Figure 6B*). A handful of putatively identified compounds were present in all samples, but had significantly higher abundance in summing flies versus exposed, non-summing flies. These included uridine, guanosine, and 5-methylcytosine (*Figure 6B*, *Figure 6—figure supplement 2A*, *Supplementary file 3*). Other

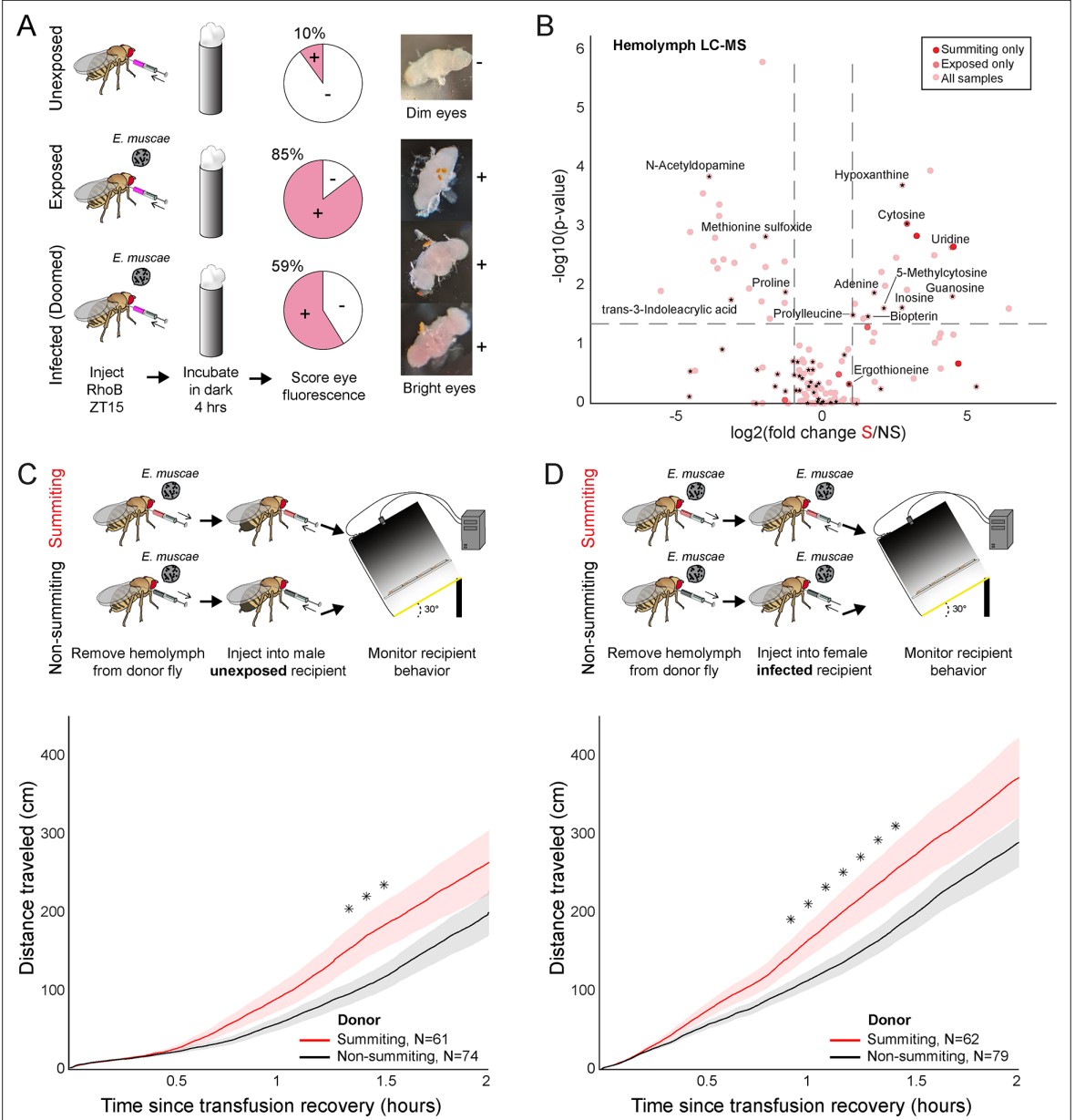

**Figure 6.** Hemolymph of summiting flies has a distinct metabolome and induces locomotion. (**A**) Blood-brain barrier (BBB) permeability of *E. muscae* exposed (96 hr) or unexposed flies assessed as the portion of flies with eye fluorescence after Rhodamine B (RhoB) injection (N=40–50 per group). Infected (doomed) flies are exposed flies with fungal growth visible by the eye through the abdominal cuticle, all of whom would go on to summit within 22 hr. Bright-eyed flies (+) had visible RhoB uptake. Representative brains from dim and bright-eyed flies are shown at right. (**B**) Volcano plot of hemolymph metabolites detected by LC-MS mass spectrometry in summiting (S) versus exposed, non-summiting (NS) flies. Putative identifications are given for selected compounds. See ***Supplementary file 3*** for compound abundances and statistical details. (**C** and **D**) Total distance traveled versus time for flies receiving a transfusion of hemolymph from summiting donors. Diagrams at the top indicate the hemolymph transfusion experiment configuration. Shaded areas indicate +/− 1 standard error. Asterisks indicate p-values <0.05 for two-tailed t-tests performed at each timepoint.

The online version of this article includes the following source data and figure supplement(s) for figure 6:

**Figure supplement 1.** Blood-brain permeability as a function of time since exposure.

**Figure supplement 2.** Metabolomics of summiting flies.

**Figure supplement 2—source data 1.** Compounds over- or under-abundant in the hemolymph of summiting flies from classifier-staged metabolomics experiment.

putatively identified compounds were more abundant in exposed, non-summing versus summing flies: N-acetyldopamine, methionine sulfoxide, and trans-3-Indoleacrylic acid (*Figure 6B*, *Figure 6— figure supplement 2B and C*). Overall, these data indicate that summing fly hemolymph is distinct from that of exposed, non-summing flies.

To determine if factor(s) in the hemolymph of summing flies could cause summing behavior, we transfused hemolymph from summing donors to non-summing recipients, and tracked their ensuing behavior. We performed this experiment using exposed female donors and naive (unexposed) male recipients. Males tend to be smaller than females, so this choice of sexes maximized the quantity of hemolymph we could extract while minimizing its dilution in recipients. We observed a modest (37%) but significant increase in the distance traveled between 80 and 90 min post-transfusion, in flies that received summing hemolymph compared to controls that received non-summing hemolymph ($0.033 < p < 0.039$; *Figure 6C*). We conducted a second version of this experiment, this time with fungus-exposed females as the recipients, and observed a similar increase in total distance traveled within the first 55–85 min after transfusion (44% increase, $0.024 < p < 0.048$; *Figure 6D*). It is apparent that the hemolymph carries factors that can induce a summing-like increase in locomotor activity.

## A neuro-mechanistic framework for summing behavior

Altogether, our experiments point to a series of mechanisms by which *E. muscae* induces zombie summing behavior (*Figure 7*). The fungus invades the brain as early as 48 hr prior to death (*Elya et al., 2018*), establishing extensive SMP occupancy by at least 24 hr before death. When summing behavior begins ~2.5 hr prior to death, the fungus has altered host hemolymph, likely via secretion of secondary metabolites. We hypothesize that these metabolites lead to the activation of PI-CA neurons, potentially via upstream DN1p clock neurons. In turn, we suspect that PI-CA activation stimulates the CA, leading to the release of JH. This hormone ultimately feeds back on the nervous system to generate the increase in locomotion at the heart of summing. This framework unites the observations from many experiments and provides several specific hypotheses that we aim to tackle in future work.

## Discussion

The discovery of dead, fungus-covered flies in elevated locales has fascinated the scientifically curious for at least the past 150 years (*Berisford and Tsao, 1974*; *Cohn, 1855*; *Gryganskyi et al., 2013*; *Mullens et al., 1987*). Until very recently the biological mechanisms determining how they got there have been purely a matter of guesswork. Here, we reported a multi-pronged approach to characterize summing behavior in zombified flies and make the first substantial progress towards understanding its mechanistic underpinnings using the *E. muscae-D. melanogaster* 'zombie fly' system.

## A new understanding of summit disease

By analyzing the behavior of hundreds of *E. muscae*-exposed wild-type Canton-S flies in a custom summing assay (*Figure 1C*), we discovered that a signature of summit disease is a burst of locomotor activity in the final ~2.5 hr of a zombie fly's life (*Figure 1F–H*). If the fly was previously in a low position, such as on the ground, or, in our assay, on the food, the net effect of increased activity will be upward motion. Perhaps it may be easier for parasites to evolve to manipulate neural mechanisms underlying activity in general, rather than the more specific circuits mediating negative gravitaxis. Notably, flies tend to die in higher positions when they begin summing in the middle of a long arena (as determined by the positioning of the food) (*Figure 1—figure supplement 2I*). This implies that *E. muscae* induces both increased activity and negative gravitaxis (to some degree), which interact with the geometry of the arena and the position of the fly prior to behavioral manipulation, to produce the summing phenotype. Enhanced locomotor activity (ELA) is emerging as a recurring theme in insect behavior manipulation, having now been reported as a result of parasitism by not only fungi (*Boyce et al., 2019*; *Trinh et al., 2021*) but also viruses (*Kamita et al., 2005*; *van Houte et al., 2012*). It remains to be seen if other known examples of ELA are driven by similar mechanisms as by *E. muscae* and whether ELA is a universal feature of parasite-induced summit disease (e.g. in *Entomophaga grylli*-infected grasshoppers and *Pandora formica*- (*Małagocka et al., 2017*) and *Dicrocoelium dendriticum*-infected ants; *Pickford and Riegert, 1964*; *Martín-Vega et al., 2018*).

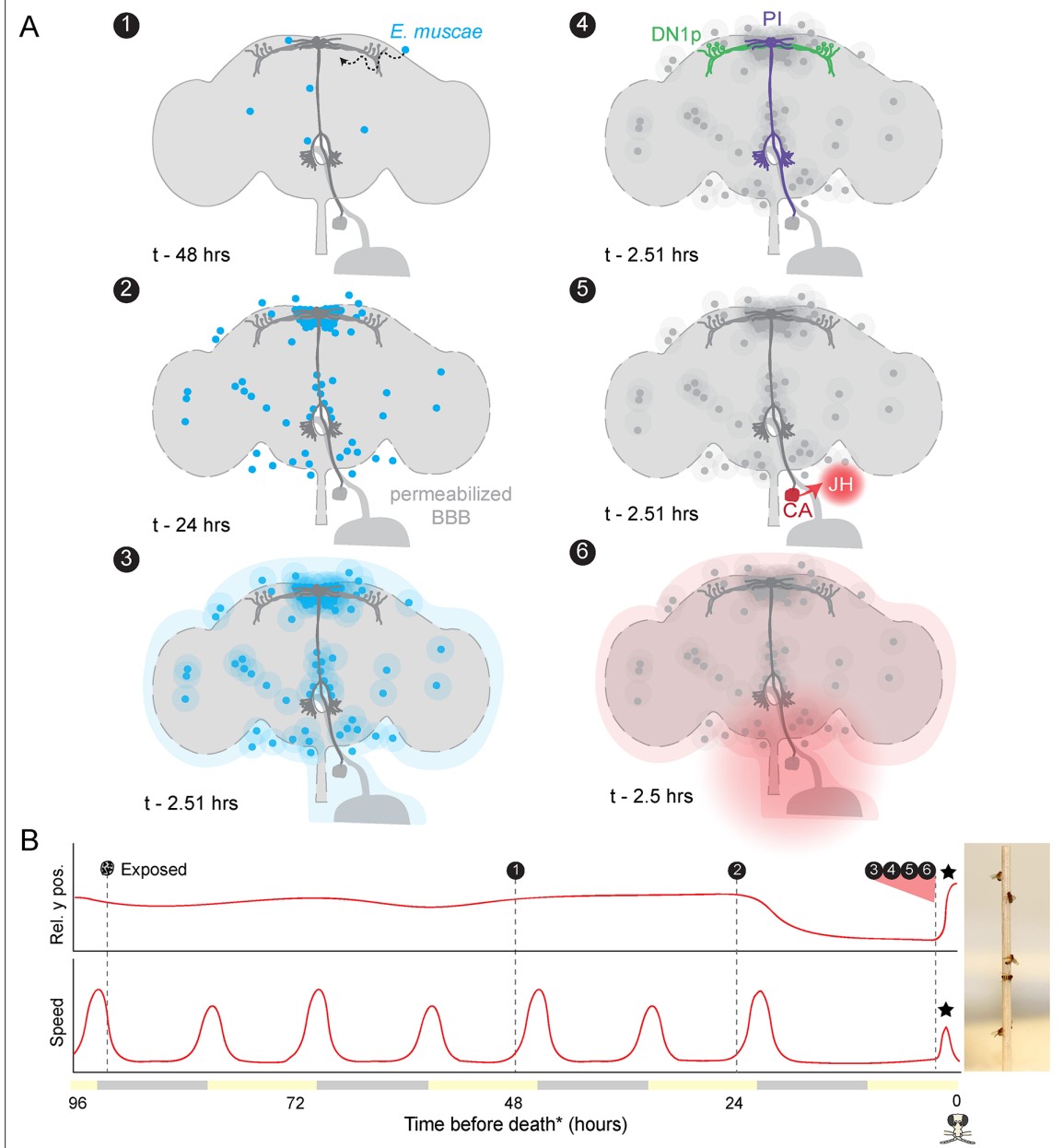

**Figure 7.** Proposed sequence of *E.muscae*-induced summiting mechanisms in zombie flies. (**A**) Events in the host brain leading to *E. muscae*-induced summiting. (1) *E. muscae* cells are present in the brain as soon as 48 hr prior to death (*Elya et al., 2018*). (2) By 24 hr prior to death, the fungus is present at a high density in the superior medial protocerebrum (SMP). This corresponds to the 'infected (doomed)' status of flies in *Figure 6*. (3) *E. muscae* alters the hemolymph (perhaps by secreting compounds, as depicted here) to trigger the onset of summiting behavior. (4) Hemolymph-borne factors alter the activity of the circadian network/DN1p and pars intercerebralis to corpora allata (PI-CA) neurons. (5) Juvenile hormone (JH) is released from the corpora allata (CA) following changes in PI-CA activity. (6) Increased JH levels drive an increase in locomotion. The dashed outline of the brain becomes more prominent between steps 1 and 3 to reflect an increase in blood-brain barrier (BBB) permeability over these timepoints. (**B**) Left: Timeline of events depicted in (**A**) overlaid on cartoon plot of average relative y position (above) and speed (below) for zombie flies. Summiting is indicated by a black star; death (time of the last movement) is indicated by a fly 'skull.' Right: Zombie flies summited on a wooden dowel.

## Host circadian and pars intercerebralis neurons mediate summiting

We leveraged our high throughput assay to screen for fly circuit elements mediating summiting and found evidence for the involvement of circadian and neurosecretory systems (*Figure 2A–E*). We identified two specific neuronal populations important for summiting: DN1p circadian neurons labeled by *Clk4.1-Gal4* (*Figure 2F*) and a small population of PI-CA neurons labeled by *R19G10-Gal4* (*Figure 2G*).

Silencing these neurons significantly reduced summiting and ectopically activating them induced a summiting-like burst of locomotor activity (*Figure 2I–K*). These neurons are likely part of the same circuit; the projection of DN1ps to the PI has been confirmed both anatomically (*Cavanaugh et al., 2014*) and functionally (*Barber et al., 2021*). Future work to visualize PI-CA and DN1p activity during summiting is needed to verify this assertion.

The pathway formed by these neurons is reminiscent of a previously characterized circadian-locomotor pathway. *Cavanaugh et al., 2014* showed that sLNv pacemaker neurons signal via DN1ps to a subset of PI neurons expressing the neuropeptide Dh44. Dh44-positive PI neurons project to a population of hugin-positive neurons in the subesophageal ganglion (SOG), some of which send descending processes to the VNC (*Cavanaugh et al., 2014*; *King et al., 2017*). Recently, neurons that express both hugin and Dh44 receptor 2 (putatively the hugin+ SOG neurons in *King et al., 2017*) were found to project to the CA (*Mizuno et al., 2021*). We did not observe a decrease in summiting by silencing or ablating sLNvs (*Figure 2—figure supplement 1D*) or by silencing Dh44+ PI neurons (*Figure 2—figure supplement 1F*). However, we did observe an effect of silencing hugin+ neurons (*Figure 2—figure supplement 1F*). While it remains to be seen if any PI-CA neurons express Dh44, it is likely there are multiple connections between the PI and neurosecretory organs, and these pathways collectively exert control over locomotion. In the future, defining the neuropeptide profiles of PI-CA neurons may provide insight into the parasite's proximate manipulation mechanism.

Silencing PI-CA neurons or mutating *Dh31* blocked summiting almost entirely, but silencing DN1p neurons had an effect that was roughly half as large (*Figure 2G*). This could reflect the heterogeneity of DN1p cells (*Ma et al., 2021*). Another possibility is that additional inputs to PI-CA also mediate summiting manipulation, perhaps the Lateral Posterior clock Neurons (LPNs), which were also recently discovered to express Dh31 (*Reinhard et al., 2022a*). The evolutionary logic of targeting the circadian network is elegant: strains of *E. muscae* have been reported to infect and manipulate a diverse collection of dipteran hosts (*Elya and De Fine Licht, 2021*). The proximate motor circuits controlling locomotor activity may vary from species to species, but all flies have a clock (*Helfrich-Förster et al., 2020*; *Sandrelli et al., 2008*) and the clock exerts a strong influence on locomotor behavior. Targeting the clock network and downstream neurosecretory neurons may represent a simple, conserved mechanism to appropriately activate motor programs across host species.

Our data indicate that the host circadian network is involved in mediating the increased locomotor activity that we now understand to define summiting. However, our data do not speak to how the timing of this behavior is determined in the zombie-fly system. That is, we have yet to address the mechanisms underlying the temporal gating of summiting and death. Our observation that *E. muscae*-infected fruit flies continue to die at specific times of day in the absence of proximal lighting cues (*Figure 1—figure supplement 1*) suggests that the timing of death is under circadian control and aligns with previous work in *E. muscae*-infected house flies (*Krasnoff et al., 1995*). Given that molecular clocks are prevalent across the tree of life, it is likely that two clocks (one on the fly, one in *E. muscae*) are present in this system. Additional work is needed to determine if the host clock is required for the timing of death under free-running conditions and to assess if *E. muscae* can keep time.

## PI-CA neurons induce summiting via their connection to the corpora allata

A defining feature of PI-CA neurons is their expression of presynaptic markers at the CA (*Figure 3B*), the conserved sites of JH synthesis and release within insects. JH has been implicated in a variety of physiological and behavioral phenomena within insects broadly (*Riddiford, 2020*; *Tsang et al., 2020*) and within fruit flies specifically (*Zhang et al., 2021*). Importantly, JH is known to have sexually dimorphic effects (*Belgacem and Martin, 2007*; *Wu et al., 2018*). While thermogenetic activation of DN1ps and PI-CA neurons induced both males and females to locomote (*Figure 2—figure supplement 2A–D*), the effect was 22.4- and sixfold stronger in males, respectively. This difference is consistent with previous work implicating JH and the PI in sexually dimorphic locomotion (*Belgacem and Martin, 2002*; *Gatti et al., 2000*) and supports our conclusion that the CA and JH are the major output of DN1p and PI-CA neurons with respect to summiting. Given the sexually dimorphic effects of JH and ectopic PI-CA activation, one might expect strong sexual dimorphism in zombie summiting, but this is not observed (*Figure 1—figure supplement 2M*). We propose that the apparent absence of sexual

dimorphism in summiting is a consequence of effective castration by the fungus. Histological data showed that summiting flies either have severely damaged gonads or lack them entirely (*Figure 5G1*), similar to other instances of parasitic castration (*Cooley et al., 2018*; *Ewen, 1966*; *Lafferty and Kuris, 2009*). As JHRs are present in gonads (*Abdou et al., 2011*; *Baumann et al., 2017*), it follows that in the absence of these sexually dimorphic tissues, JH-mediated behavioral differences between the sexes would be minimized.

We showed that summiting was reduced in *E. muscae*-infected flies with ablated CA (*Figure 3C*) or when treated with the JH synthesis inhibitor precocene (*Figure 3E*). However, we did not observe exacerbated summiting behavior in animals that had been treated with the juvenile hormone analog (JHA) methoprene (*Figure 3—figure supplement 2H*) or a restoration of summiting behavior when animals received JHAs in addition to precocene (*Figure 3—figure supplement 2I*). JH manipulations were not part of our initial screen, becoming a focus after the discovery of the role of the PI-CA neurons. We suspect that summiting is driven by an acute spike in JH starting ~2.5 hr before death, and our JHA experiments did not have this timing: methoprene was delivered in a single burst 20 hr prior to summiting and pyriproxyfen was administered chronically via the food. Second, we have strong reason to believe that whatever we applied to the fly was also making its way to the fungus (recall that healthy flies treated with both fluvastatin and methoprene were fine, but that this treatment was lethal for exposed flies *Figure 3—figure supplement 2D*). Thus, another possibility is that the fungus is metabolizing the JHAs before they have a behavioral effect. We did not detect JH in any of our metabolomic experiments, however, this was expected given that we used extraction and separation methods appropriate for polar, not hydrophobic, compounds. Future work leveraging targeted, high-sensitivity chemical detection of hydrophobic compounds is needed to verify that JH titers are indeed elevated during the transient summiting window.

The role of the CA in *E. muscae*-induced summiting is consistent with the growing list of examples of parasites exploiting host hormonal axes (*Adamo and Robinson, 2012*; *Beckage, 1997*; *Herbison, 2017*; *Tong et al., 2021*). The JH pathway, in particular, has been shown to be modulated by a variety of insect parasites, ranging from nematodes to baculoviruses (*Ahmed et al., 2022*; *Jiao et al., 2022*; *Nakai et al., 2016*; *Palli et al., 2000*; *Saito et al., 2015*; *Subrahmanyam and Ramakrishnan, 1980*; *Sun et al., 2019*; *Zhang et al., 2015*). While there is a clear consensus that JH is involved in a multitude of host physiological and behavioral processes, the extent of JH's activities in insects is still being uncovered. Our data reveal another role for JH in the fruit fly: mediating *E. muscae*-induced summiting behavior.

## Machine learning classification of summiting animals in real-time

Identifying the molecular and physiological correlates of summiting is challenging for several reasons: summiting behavior is subtle to a human observer, summiting lasts just a few hours within a specific circadian window, and flies' small size makes procuring sufficient material non-trivial. To make such experiments possible, we developed an automated classifier to identify flies as early into summiting behavior as possible (*Figure 4*). The random forest algorithm (*Breiman, 2001*; *Pedregosa et al., 2012*) at the heart of our classifier identified time of day (evening), previous position (low), previous speed (low), and current speed (high) as key features identifying summiting flies (*Figure 4C and D*). The classifier achieved excellent precision and good recall on a novel cohort of exposed flies. By interfacing the classifier with an email alert system, we created a robust, scalable pipeline for procuring summiting flies for a variety of downstream experiments (*Figures 5 and 6B–D*).

## Morphological correlates of summiting

Using our real-time classifier, we conducted a comparison of host morphology prior to and during summiting. Previous analyses of infection progression suggested that the fungus was not occupying the brain with any spatial specificity (*Elya et al., 2018*), but here we found otherwise. There is a clear pattern of fungal cells densely invading the SMP of summiting flies, a neuropil that harbors DN1p axons and PI-CA cell bodies and dendrites (*Figure 5B, C and F*). This concentration of fungal cells is apparent at least 72 hr after exposure to *E. muscae* (*Figure 5—figure supplement 1A*). Fungal cells are present in the brain as early as 48 hr after exposure (*Elya et al., 2018*), and the exact timing of when they accumulate in the SMP remains to be established. The distribution of *E. muscae* across neuropils, which is consistent across animals (*Figure 5C*), is interesting both for where fungal cells

are and are not found. Fungal cells are noticeably absent from the central complex, a pre-motor center (*Bender et al., 2010*; *Strausfeld, 1976*) that may be involved in coordinating walking during summiting. Though morphological examination suggested that fungal cells are displacing (*Figure 5—figure supplement 1B*), rather than consuming, nervous tissue, more work is needed to determine if neurons are damaged or dying as a result of adjacent fungal cells. In addition, it remains unclear what role, if any, the pattern of fungal brain occupancy plays in the mechanism of summiting or if the fungal cells in the brain play a distinct role in behavior manipulation compared to those in the body cavity. Additional work is needed to address these questions.

We observed extensive degradation of host abdominal tissues in summiting animals (*Figure 5G–H*, *Figure 5—figure supplement 1E*). We were stunned to find flies with obliterated guts and gonads walking apparently normally. Despite widespread destruction in the body, the CA and PI-CA neurons appear intact in summiting animals, which is consistent with an acute role in summiting. We speculate that the fungus might achieve preservation of these tissues by preferentially digesting the remaining host tissues from posterior to anterior. However, just because PI-CA neurons and the CA are present doesn't mean they are functioning normally or at all. Future work should assess the physiology of these cells throughout the course of *E. muscae* infection.

## Physiological correlates of summiting

We discovered that the permeability of the blood-brain barrier was increased in exposed flies, as determined by assaying RhoB retention in fly brains (*Figure 6A*, *Figure 6—figure supplement 1*). Our data suggest that BBB integrity degrades by the end of infection (*Figure 6—figure supplement 1*), rather than rapidly after fungal exposure (by 21 hr) or upon fungal invasion of the nervous system (around 45 hr). A variety of insults, including bacterial infection, can lead to increased BBB permeability in fruit flies (*Kim et al., 2021*). We speculate that the progressive reduction in BBB integrity may result from the growing burden of the infection as the flies become sicker and sicker. In addition, the permeability of the BBB fluctuates over the day in a clock-dependent manner (*Zhang et al., 2018*). If the host's circadian system is disrupted during infection, this could also be a source of compromised BBB integrity.

We found that the hemolymph metabolome of exposed, summiting flies differs from that of exposed, non-summiting flies and healthy controls (*Figure 6B*, *Figure 6—figure supplement 1*). Three compounds of putative chemical formulae $C_6H_8N_2O_3$, $C_{14}H_{16}N_6O_7$, and $C_{12}H_{19}N_2PS$ appeared unique to summiting flies but could not be identified further. These compounds are prime candidates for further studies. Seven other compounds were significantly more abundant in summiting versus non-summiting flies across our replicate experiments: three of these could not be identified (MW 276.08, 179.08, and 429.15 g/mol) and the other four were putatively identified as guanosine, uridine, cytosine, and 5-methylcytosine. Future collection of large quantities of summiting flies and fractionation approaches could be used to home in on compounds of interest and determine their chemical structure such that these compounds can be produced synthetically and assayed for behavioral effects (*Beckerson et al., 2022*). Cytosine is a pyrimidine nucleobase used in both DNA and RNA, a core molecular building block. It is intriguing that it was only detected in fungus-exposed fly hemolymph. High levels of cytosine have also been detected in the hemolymph of *Beauveria bassiana*-infected silkworms (*Xu et al., 2015*) and the serum of Sars-Cov2-infected humans (*Blasco et al., 2020*), with cytosine levels actually being predictive of infection status. Notably, a major derivative of cytosine, 5-methylcytosine, is also more abundant in summiting than non-summiting hemolymph. We hypothesize that elevated levels of cytosine could be a general indicator of infection, and its specific correlation with summiting warrants further investigation.

We detected ergothioneine in flies exposed to the fungus, either summiting or non-summiting. Ergothioneine has been hypothesized to play a role in host tissue preservation in *Ophiocordyceps* manipulated ants (*Loreto and Hughes, 2019*). Our data are consistent with ergothioneine being produced by *E. muscae*, but are not consistent with ergothioneine being produced only during summiting.

We saw that N-acetyldopamine (NADA), methionine sulfoxide, and trans-3-indoleacrylic acid were more abundant in non-summiting versus summiting flies. NADA is a product of dopamine (DA) breakdown (*Neckameyer and Leal, 2017*) and has been found to inhibit CA synthesis of JHs in *Manduca sexta* larvae (*Granger et al., 2000*). DA, on the other hand, has been detected in the CA of *Manduca*

*sexta* (*Krueger et al., 1990*) and studies in bees suggest a positive correlation between dopamine (DA), JH, and activity (*Akasaka et al., 2010*; *Mezawa et al., 2013*).

To test whether hemolymph-circulating factors in summiting animals can cause an increase in locomotion, we transfused hemolymph from classifier-flagged summiting flies into fungus-exposed and non-exposed recipients (*Figure 6C and D*). In both of these experiments, recipient flies exhibited a significant increase in locomotion over ~1.5 hr post-transfusion. The effect size was modest (40% increase in total distance traveled in that interval), but this was not surprising as (1) we could only extract and transfer very small quantities (*MacMillan and Hughson, 2014*) of hemolymph between animals and (2) this small quantity was diluted throughout the whole recipient fly's body. Overall, this experiment provides direct evidence that one or more factors in the hemolymph of summiting flies cause summiting. The identity of these factors and their precise timing and origin of production (fungal or fly) remain mysteries that we hope to address in future studies.

## A mechanistic framework for summiting behavior and beyond

Our experiments have revealed key mechanisms likely to underlie the summiting behavior of zombie flies. *E. muscae* cells perturb the activity of circadian and neurosecretory neurons, leading to the release of JH and a resultant increase in locomotion. This effect is at least partially mediated by summiting-specific factors circulating in the hemolymph. Of course, many questions remain. What compounds mediate the effect of transfused hemolymph? What cells are targeted by these compounds and by what molecular mechanisms? Do the fungal cells need physical access to the brain to induce a full summiting response? Is the proximity of fungal cells adjacent to DN1p axons and PI during summiting merely a coincidence? Future work should use spatially-resolved transcriptomic, metabolomic, and immunohistochemical approaches to answer these questions.

It is likely there are yet-to-be-discovered circuit elements mediating summiting. Silencing PI-CA neurons or ablating the CA severely attenuated summiting, but did not completely eliminate it. The dispersal and survival of *E. muscae* depend on a robust summiting response in the host (*Carruthers, 1981*), and the co-evolutionary relationship between these species likely extends back 200–400 million years (*Boomsma et al., 2014*; *Elya and De Fine Licht, 2021*). Such a robust strategy is unlikely to rely on a single perturbation that could be countered by simple evolutionary changes in the host. An increase in locomotion can be achieved in many ways and is the likely output of many different behavioral circuits (*Bidaye et al., 2020*; *Cavanaugh et al., 2014*; *Lee et al., 2021*), so it would be unsurprising to find that multiple host circuits are targeted, including others yet to be discovered. Nevertheless, our study has identified a host pathway that likely mediates the predominant effects of the zombie fly summiting manipulation. These discoveries were made possible by studying summiting in a genetic model organism using high throughput behavioral assays. These tools and more will be essential to answer the many exciting questions arising from this work.

# Materials and methods

**Key resources table**

| Reagent type (species) or resource | Designation | Source or reference | Identifiers | Additional information |
|---|---|---|---|---|
| antibody | anti-Chicken-AF488 (goat polyclonal) | Thermo Fisher | Cat#: A-11039, RRID:AB_2534096 | IF(1:800) |
| antibody | anti-dsRed (rabbit polyclonal) | Takara Bio | Cat#: 632496, RRID:AB_10013483 | IF:(250) |
| antibody | anti-GFP (chicken polyclonal) | Aves Labs | Cat#: GFP-1020, RRID:AB_10000240 | IF(1:4000) |
| antibody | anti-Guinea Pig-AF568 (goat polyclonal) | Thermo Fisher | Cat#: A-11075, RRID:AB_2534119 | IF(1:400) |
| antibody | anti-JHAMT (guinea pig polyclonal) | *Niwa et al., 2008* | | IF(1:1000) |
| antibody | anti-Mouse-Cy5 (goat polyclonal) | Millipore | Cat#: AP500S, RRID:AB_805361 | IF(1:400) |

*Continued on next page*

*Continued*

| Reagent type (species) or resource | Designation | Source or reference | Identifiers | Additional information |
|---|---|---|---|---|
| antibody | anti-nc82 (mouse monoclonal) | Iowa Developmental Studies Hybridoma Bank | Cat#: nc82, RRID:AB_2314866 | IF(1:40) |
| antibody | anti-Rabbit-AF568 (goat polyclonal) | Thermo Fisher | Cat# A-11011, RRID:AB_143157 | IF(1:250) |
| genetic reagent (*D. melanogaster*) | 104y-Gal4 | Bloomington *Drosophila* Stock Center | BDSC:81014 | |
| genetic reagent (*D. melanogaster*) | 104y-Gal4; Cha-Gal80 | Derived from BDSC:81014 & *Cha-Gal80* | | |
| genetic reagent (*D. melanogaster*) | acj6- | Bloomington *Drosophila* Stock Center | BDSC:30025 | |
| genetic reagent (*D. melanogaster*) | acj6-Gal4 | Bloomington *Drosophila* Stock Center | BDSC:30025 | |
| genetic reagent (*D. melanogaster*) | Akh- | Bloomington *Drosophila* Stock Center | BDSC:84448 | |
| genetic reagent (*D. melanogaster*) | AstC- | Bloomington *Drosophila* Stock Center | BDSC:84453 | |
| genetic reagent (*D. melanogaster*) | Bl/CyO; tub-Gal80(ts) | Kristin Scott (*McGuire et al., 2004*) | | |
| genetic reagent (*D. melanogaster*) | C(1)Dxyfv(X^X)/Y; Aug21-Gal4, UAS-GFP/CyO | Rochele Yamamoto (*Yamamoto et al., 2013*) | | |
| genetic reagent (*D. melanogaster*) | c17-Gal4 | Bloomington *Drosophila* Stock Center | BDSC:39690 | |
| genetic reagent (*D. melanogaster*) | c41-Gal4 | Bloomington *Drosophila* Stock Center | BDSC:30834 | |
| genetic reagent (*D. melanogaster*) | c708a-Gal4 | Bloomington *Drosophila* Stock Center | BDSC:50743 | |
| genetic reagent (*D. melanogaster*) | Canton-S | Liming Wang | | |
| genetic reagent (*D. melanogaster*) | CCha1- | Bloomington *Drosophila* Stock Center | BDSC:84458 | |
| genetic reagent (*D. melanogaster*) | CCKR-17D1- | Bloomington *Drosophila* Stock Center | BDSC:84462 | |
| genetic reagent (*D. melanogaster*) | CCLKR-17D3- | Bloomington *Drosophila* Stock Center | BDSC:84463 | |
| genetic reagent (*D. melanogaster*) | Cha-Gal80/TM3, Sb | Toshihiro Kitamoto (*Kitamoto, 2002*) | | |
| genetic reagent (*D. melanogaster*) | Clk4.1-Gal4 | Bloomington *Drosophila* Stock Center | BDSC:36316 | |
| genetic reagent (*D. melanogaster*) | Clk4.5-Gal4 | Bloomington *Drosophila* Stock Center | BDSC:37526 | |
| genetic reagent (*D. melanogaster*) | Clk856-Gal4/CyO; 911-QF, QUAS-FLP/TM6, Sb | David Cavanaugh (*Nettnin et al., 2021*) | | |
| genetic reagent (*D. melanogaster*) | Clk856-Gal4/CyO; MKRS/TM6B | Daniel Cavanaugh (*Gummadova et al., 2009*) | | |
| genetic reagent (*D. melanogaster*) | Clkar | Bloomington *Drosophila* Stock Center | BDSC:24513 | |
| genetic reagent (*D. melanogaster*) | ClkJrk | Bloomington *Drosophila* Stock Center | BDSC:24515 | |

*Continued on next page*

*Continued*

| Reagent type (species) or resource | Designation | Source or reference | Identifiers | Additional information |
|---|---|---|---|---|
| genetic reagent (*D. melanogaster*) | Clout | Bloomington *Drosophila* Stock Center | BDSC:56754 | |
| genetic reagent (*D. melanogaster*) | CNMa- | Bloomington *Drosophila* Stock Center | BDSC:84485 | |
| genetic reagent (*D. melanogaster*) | CNMaR- | Bloomington *Drosophila* Stock Center | BDSC:84486 | |
| genetic reagent (*D. melanogaster*) | cry-Gal4.Z16 | Bloomington *Drosophila* Stock Center | BDSC:24514 | |
| genetic reagent (*D. melanogaster*) | cry-Gal4.Z24 | Bloomington *Drosophila* Stock Center | BDSC:24774 | |
| genetic reagent (*D. melanogaster*) | cry02 | Bloomington *Drosophila* Stock Center | BDSC:86267 | |
| genetic reagent (*D. melanogaster*) | cryb | Bloomington *Drosophila* Stock Center | BDSC:80921 | |
| genetic reagent (*D. melanogaster*) | cyc01 | Bloomington *Drosophila* Stock Center | BDSC:80929 | |
| genetic reagent (*D. melanogaster*) | DAT- | Bloomington *Drosophila* Stock Center | BDSC:25547 | |
| genetic reagent (*D. melanogaster*) | Dh31- | Bloomington *Drosophila* Stock Center | BDSC:84490 | |
| genetic reagent (*D. melanogaster*) | Dh31KG09001 | Bloomington *Drosophila* Stock Center | BDSC:16474 | |
| genetic reagent (*D. melanogaster*) | DH31R- | Bloomington *Drosophila* Stock Center | BDSC:84491 | |
| genetic reagent (*D. melanogaster*) | disco1 | Bloomington *Drosophila* Stock Center | BDSC:5682 | |
| genetic reagent (*D. melanogaster*) | DNc01 | Janelia Research Center | JRC:SS04161 | |
| genetic reagent (*D. melanogaster*) | DNc02 | Janelia Research Center | JRC:SS02395 | |
| genetic reagent (*D. melanogaster*) | DNp01 | Janelia Research Center | JRC:SS00726 | |
| genetic reagent (*D. melanogaster*) | DNp01 | Janelia Research Center | JRC:SS00727 | |
| genetic reagent (*D. melanogaster*) | DNp01 | Janelia Research Center | JRC:SS02299 | |
| genetic reagent (*D. melanogaster*) | Dsk- | Bloomington *Drosophila* Stock Center | BDSC:84497 | |
| genetic reagent (*D. melanogaster*) | elav-Gal4; UAS-Dcr2 | Bloomington *Drosophila* Stock Center | BDSC:25750 | |
| genetic reagent (*D. melanogaster*) | forS | Bloomington *Drosophila* Stock Center | BDSC:76120 | |
| genetic reagent (*D. melanogaster*) | fru-Gal4 | Bloomington *Drosophila* Stock Center | BDSC:30027 | |
| genetic reagent (*D. melanogaster*) | GH86-Gal4 | Bloomington *Drosophila* Stock Center | BDSC:36339 | |
| genetic reagent (*D. melanogaster*) | gl60j | Bloomington *Drosophila* Stock Center | BDSC:509 | |

*Continued*

| Reagent type (species) or resource | Designation | Source or reference | Identifiers | Additional information |
|---|---|---|---|---|
| genetic reagent (*D. melanogaster*) | GLSNP3375-Gal4 | Kyoto *Drosophila* Stock Center | KDSC:104479 | |
| genetic reagent (*D. melanogaster*) | His-RFP | Bloomington *Drosophila* Stock Center | BDSC:23651 | |
| genetic reagent (*D. melanogaster*) | Hug-Gal4 | Bloomington *Drosophila* Stock Center | BDSC:58769 | |
| genetic reagent (*D. melanogaster*) | iav-Gal4 | Bloomington *Drosophila* Stock Center | BDSC:52273 | |
| genetic reagent (*D. melanogaster*) | Ilp1-Gal4 | Bloomington *Drosophila* Stock Center | BDSC:66005 | |
| genetic reagent (*D. melanogaster*) | Ilp2-Gal4 | Bloomington *Drosophila* Stock Center | BDSC:37516 | |
| genetic reagent (*D. melanogaster*) | Ilp3-Gal4 | Bloomington *Drosophila* Stock Center | BDSC:52660 | |
| genetic reagent (*D. melanogaster*) | Ilp5-Gal4 | Bloomington *Drosophila* Stock Center | BDSC:66008 | |
| genetic reagent (*D. melanogaster*) | JO-ACE-Gal4 | Kyoto *Drosophila* Stock Center | KDSC:113902 | |
| genetic reagent (*D. melanogaster*) | JO-CE-Gal4 | Kyoto *Drosophila* Stock Center | KDSC:113878 | |
| genetic reagent (*D. melanogaster*) | JO15-Gal4 | Bloomington *Drosophila* Stock Center | BDSC:6753 | |
| genetic reagent (*D. melanogaster*) | Kurs58-Gal4 | Bloomington *Drosophila* Stock Center | BDSC:80985 | |
| genetic reagent (*D. melanogaster*) | MB010B-Gal4 | Janelia Research Center | JRC:MB010B | |
| genetic reagent (*D. melanogaster*) | Mmp2NP0509-Gal4 | Kyoto *Drosophila* Stock Center | KDSC:103625 | |
| genetic reagent (*D. melanogaster*) | nan-Gal4 | Bloomington *Drosophila* Stock Center | BDSC:24903 | |
| genetic reagent (*D. melanogaster*) | nan36a | Kristin Scott (*Kim et al., 2003*) | | |
| genetic reagent (*D. melanogaster*) | NPF- | Bloomington *Drosophila* Stock Center | BDSC:84549 | |
| genetic reagent (*D. melanogaster*) | Oamb- | Bloomington *Drosophila* Stock Center | BDSC:22758 | |
| genetic reagent (*D. melanogaster*) | OctBeta1R- | Bloomington *Drosophila* Stock Center | BDSC:18589 | |
| genetic reagent (*D. melanogaster*) | Octbeta2R- | Bloomington *Drosophila* Stock Center | BDSC:18896 | |
| genetic reagent (*D. melanogaster*) | OctBeta3R- | Bloomington *Drosophila* Stock Center | BDSC:24819 | |
| genetic reagent (*D. melanogaster*) | Pdf- | Bloomington *Drosophila* Stock Center | BDSC:84561 | |
| genetic reagent (*D. melanogaster*) | Pdf-Gal4 | Bloomington *Drosophila* Stock Center | BDSC:6899 | |
| genetic reagent (*D. melanogaster*) | Pdf-Gal80, cry24-Gal4 | Bloomington *Drosophila* Stock Center | BDSC:80940 | |

*Continued on next page*

*Continued*

| Reagent type (species) or resource | Designation | Source or reference | Identifiers | Additional information |
|---|---|---|---|---|
| genetic reagent (*D. melanogaster*) | Pdf01 | Bloomington *Drosophila* Stock Center | BDSC:26654 | |
| genetic reagent (*D. melanogaster*) | PdfR- | Bloomington *Drosophila* Stock Center | BDSC:84705 | |
| genetic reagent (*D. melanogaster*) | PdfR-; DH31R- | Derived from BDSC:84705 & BDSC:84491 | | |
| genetic reagent (*D. melanogaster*) | PdfR-Gal4 | Bloomington *Drosophila* Stock Center | BDSC:68215 | |
| genetic reagent (*D. melanogaster*) | PdfR5304 | Bloomington *Drosophila* Stock Center | BDSC:33068 | |
| genetic reagent (*D. melanogaster*) | per-Gal4 | Bloomington *Drosophila* Stock Center | BDSC:7127 | |
| genetic reagent (*D. melanogaster*) | per01 | Bloomington *Drosophila* Stock Center | BDSC:80928 | |
| genetic reagent (*D. melanogaster*) | per30 | Bloomington *Drosophila* Stock Center | BDSC:63136 | |
| genetic reagent (*D. melanogaster*) | perS | Bloomington *Drosophila* Stock Center | BDSC:80919 | |
| genetic reagent (*D. melanogaster*) | ple-Gal4 | Bloomington *Drosophila* Stock Center | BDSC:8848 | |
| genetic reagent (*D. melanogaster*) | Procc04750 | Bloomington *Drosophila* Stock Center | BDSC:11587 | |
| genetic reagent (*D. melanogaster*) | ProcMI06590 | Bloomington *Drosophila* Stock Center | BDSC:42407 | |
| genetic reagent (*D. melanogaster*) | ProcRMB00909 | Bloomington *Drosophila* Stock Center | BDSC:22930 | |
| genetic reagent (*D. melanogaster*) | R10F08-Gal4 | Bloomington *Drosophila* Stock Center | BDSC:48441 | |
| genetic reagent (*D. melanogaster*) | R10H10-Gal4 | Bloomington *Drosophila* Stock Center | BDSC:48445 | |
| genetic reagent (*D. melanogaster*) | R11B09-Gal4 | Bloomington *Drosophila* Stock Center | BDSC:48288 | |
| genetic reagent (*D. melanogaster*) | R11C01-Gal4 | Bloomington *Drosophila* Stock Center | BDSC:49240 | |
| genetic reagent (*D. melanogaster*) | R14F05-Gal4 | Bloomington *Drosophila* Stock Center | BDSC:49257 | |
| genetic reagent (*D. melanogaster*) | R16C05-Gal4 | Bloomington *Drosophila* Stock Center | BDSC:48718 | |
| genetic reagent (*D. melanogaster*) | R18H11-Gal4 | Bloomington *Drosophila* Stock Center | BDSC:48832 | |
| genetic reagent (*D. melanogaster*) | R19B09-Gal4 | Bloomington *Drosophila* Stock Center | BDSC:48840 | |
| genetic reagent (*D. melanogaster*) | R19G10-Gal4 | Bloomington *Drosophila* Stock Center | BDSC:47887 | |
| genetic reagent (*D. melanogaster*) | R20A02-Gal4 | Bloomington *Drosophila* Stock Center | BDSC:48870 | |
| genetic reagent (*D. melanogaster*) | R20E05-Gal4 | Bloomington *Drosophila* Stock Center | BDSC:48898 | |

*Continued*

| Reagent type (species) or resource | Designation | Source or reference | Identifiers | Additional information |
|---|---|---|---|---|
| genetic reagent (*D. melanogaster*) | R21H04-Gal4 | Bloomington *Drosophila* Stock Center | BDSC:48958 | |
| genetic reagent (*D. melanogaster*) | R23E10-Gal4 | Bloomington *Drosophila* Stock Center | BDSC:49032 | |
| genetic reagent (*D. melanogaster*) | R25G04-Gal4 | Bloomington *Drosophila* Stock Center | BDSC:49136 | |
| genetic reagent (*D. melanogaster*) | R26D11-Gal4 | Bloomington *Drosophila* Stock Center | BDSC:49323 | |
| genetic reagent (*D. melanogaster*) | R27A05-Gal4 | Bloomington *Drosophila* Stock Center | BDSC:49208 | |
| genetic reagent (*D. melanogaster*) | R30G08-Gal4 | Bloomington *Drosophila* Stock Center | BDSC:48101 | |
| genetic reagent (*D. melanogaster*) | R32G08-Gal4 | Bloomington *Drosophila* Stock Center | BDSC:49729 | |
| genetic reagent (*D. melanogaster*) | R32H03-Gal4 | Bloomington *Drosophila* Stock Center | BDSC:49733 | |
| genetic reagent (*D. melanogaster*) | R34C05-Gal4 | Bloomington *Drosophila* Stock Center | BDSC:49778 | |
| genetic reagent (*D. melanogaster*) | R43D05-Gal4 | Bloomington *Drosophila* Stock Center | BDSC:41259 | |
| genetic reagent (*D. melanogaster*) | R44B02-Gal4 | Bloomington *Drosophila* Stock Center | BDSC:50199 | |
| genetic reagent (*D. melanogaster*) | R45B03-Gal4 | Bloomington *Drosophila* Stock Center | BDSC:50221 | |
| genetic reagent (*D. melanogaster*) | R46E11-Gal4 | Bloomington *Drosophila* Stock Center | BDSC:50272 | |
| genetic reagent (*D. melanogaster*) | R47A08-Gal4 | Bloomington *Drosophila* Stock Center | BDSC:50288 | |
| genetic reagent (*D. melanogaster*) | R50C11-Gal4 | Bloomington *Drosophila* Stock Center | BDSC:38742 | |
| genetic reagent (*D. melanogaster*) | R50H05-Gal4 | Bloomington *Drosophila* Stock Center | BDSC:38764 | |
| genetic reagent (*D. melanogaster*) | R51H05-Gal4 | Bloomington *Drosophila* Stock Center | BDSC:41275 | |
| genetic reagent (*D. melanogaster*) | R54D11-Gal4 | Bloomington *Drosophila* Stock Center | BDSC:41279 | |
| genetic reagent (*D. melanogaster*) | R57C10-Gal4 | Bloomington *Drosophila* Stock Center | BDSC:39171 | |
| genetic reagent (*D. melanogaster*) | R57F07-Gal4 | Bloomington *Drosophila* Stock Center | BDSC:46389 | |
| genetic reagent (*D. melanogaster*) | R61G12-Gal4 | Bloomington *Drosophila* Stock Center | BDSC:41286 | |
| genetic reagent (*D. melanogaster*) | R64C04-Gal4 | Bloomington *Drosophila* Stock Center | BDSC:39296 | |
| genetic reagent (*D. melanogaster*) | R64C10-Gal4 | Bloomington *Drosophila* Stock Center | BDSC:39301 | |
| genetic reagent (*D. melanogaster*) | R65C07-Gal4 | Bloomington *Drosophila* Stock Center | BDSC:39344 | |

*Continued*

| Reagent type (species) or resource | Designation | Source or reference | Identifiers | Additional information |
|---|---|---|---|---|
| genetic reagent (*D. melanogaster*) | R65C11-Gal4 | Bloomington *Drosophila* Stock Center | BDSC:39347 | |
| genetic reagent (*D. melanogaster*) | R66B05-Gal4 | Bloomington *Drosophila* Stock Center | BDSC:39389 | |
| genetic reagent (*D. melanogaster*) | R70F10-Gal4 | Bloomington *Drosophila* Stock Center | BDSC:39545 | |
| genetic reagent (*D. melanogaster*) | R70G01-Gal4 | Bloomington *Drosophila* Stock Center | BDSC:39546 | |
| genetic reagent (*D. melanogaster*) | R78G02-Gal4 | Bloomington *Drosophila* Stock Center | BDSC:40010 | |
| genetic reagent (*D. melanogaster*) | R85A11-Gal4 | Bloomington *Drosophila* Stock Center | BDSC:40415 | |
| genetic reagent (*D. melanogaster*) | R86H08-Gal4 | Bloomington *Drosophila* Stock Center | BDSC:40471 | |
| genetic reagent (*D. melanogaster*) | R91A01-Gal4 | Bloomington *Drosophila* Stock Center | BDSC:40569 | |
| genetic reagent (*D. melanogaster*) | R95E11-Gal4 | Bloomington *Drosophila* Stock Center | BDSC:40711 | |
| genetic reagent (*D. melanogaster*) | RNAi-acj6 | Bloomington *Drosophila* Stock Center | BDSC:29335 | |
| genetic reagent (*D. melanogaster*) | RNAi-Akh | Bloomington *Drosophila* Stock Center | BDSC:27031 | |
| genetic reagent (*D. melanogaster*) | RNAi-Cry | Bloomington *Drosophila* Stock Center | BDSC:51033 | |
| genetic reagent (*D. melanogaster*) | RNAi-Crz | Bloomington *Drosophila* Stock Center | BDSC:25999 | |
| genetic reagent (*D. melanogaster*) | RNAi-Crz | Bloomington *Drosophila* Stock Center | BDSC:26017 | |
| genetic reagent (*D. melanogaster*) | RNAi-CrzR | Bloomington *Drosophila* Stock Center | BDSC:42751 | |
| genetic reagent (*D. melanogaster*) | RNAi-DAT | Bloomington *Drosophila* Stock Center | BDSC:31256 | |
| genetic reagent (*D. melanogaster*) | RNAi-DAT | Bloomington *Drosophila* Stock Center | BDSC:50619 | |
| genetic reagent (*D. melanogaster*) | RNAi-DDC | Bloomington *Drosophila* Stock Center | BDSC:27030 | |
| genetic reagent (*D. melanogaster*) | RNAi-DDC | Bloomington *Drosophila* Stock Center | BDSC:51462 | |
| genetic reagent (*D. melanogaster*) | RNAi-Dh31 | Bloomington *Drosophila* Stock Center | BDSC:41957 | |
| genetic reagent (*D. melanogaster*) | RNAi-Dh44 | Bloomington *Drosophila* Stock Center | BDSC:25804 | |
| genetic reagent (*D. melanogaster*) | RNAi-for | Bloomington *Drosophila* Stock Center | BDSC:21592 | |
| genetic reagent (*D. melanogaster*) | RNAi-for | Bloomington *Drosophila* Stock Center | BDSC:31698 | |
| genetic reagent (*D. melanogaster*) | RNAi-Lk | Bloomington *Drosophila* Stock Center | BDSC:25936 | |

*Continued*

| Reagent type (species) or resource | Designation | Source or reference | Identifiers | Additional information |
|---|---|---|---|---|
| genetic reagent (*D. melanogaster*) | RNAi-LkR | Bloomington *Drosophila* Stock Center | BDSC:25836 | |
| genetic reagent (*D. melanogaster*) | RNAi-Nplp2 | Bloomington *Drosophila* Stock Center | BDSC:53967 | |
| genetic reagent (*D. melanogaster*) | RNAi-Nplp2 | Bloomington *Drosophila* Stock Center | BDSC:54041 | |
| genetic reagent (*D. melanogaster*) | RNAi-Oamb | Bloomington *Drosophila* Stock Center | BDSC:31171 | |
| genetic reagent (*D. melanogaster*) | RNAi-Oamb | Bloomington *Drosophila* Stock Center | BDSC:31233 | |
| genetic reagent (*D. melanogaster*) | RNAi-Oct-Tyr | Bloomington *Drosophila* Stock Center | BDSC:28332 | |
| genetic reagent (*D. melanogaster*) | RNAi-OctAlpha2R | Bloomington *Drosophila* Stock Center | BDSC:50678 | |
| genetic reagent (*D. melanogaster*) | RNAi-OctBeta1R | Bloomington *Drosophila* Stock Center | BDSC:31106 | |
| genetic reagent (*D. melanogaster*) | RNAi-OctBeta1R | Bloomington *Drosophila* Stock Center | BDSC:31107 | |
| genetic reagent (*D. melanogaster*) | RNAi-OctBeta1R | Bloomington *Drosophila* Stock Center | BDSC:50701 | |
| genetic reagent (*D. melanogaster*) | RNAi-OctBeta1R | Bloomington *Drosophila* Stock Center | BDSC:58179 | |
| genetic reagent (*D. melanogaster*) | RNAi-OctBeta2R | Bloomington *Drosophila* Stock Center | BDSC:34673 | |
| genetic reagent (*D. melanogaster*) | RNAi-OctBeta2R | Bloomington *Drosophila* Stock Center | BDSC:50580 | |
| genetic reagent (*D. melanogaster*) | RNAi-OctBeta3R | Bloomington *Drosophila* Stock Center | BDSC:31108 | |
| genetic reagent (*D. melanogaster*) | RNAi-Pdf | Bloomington *Drosophila* Stock Center | BDSC:25802 | |
| genetic reagent (*D. melanogaster*) | RNAi-ple | Bloomington *Drosophila* Stock Center | BDSC:25796 | |
| genetic reagent (*D. melanogaster*) | RNAi-ple | Bloomington *Drosophila* Stock Center | BDSC:65875 | |
| genetic reagent (*D. melanogaster*) | RNAi-ple | Bloomington *Drosophila* Stock Center | BDSC:76062 | |
| genetic reagent (*D. melanogaster*) | RNAi-ple | Bloomington *Drosophila* Stock Center | BDSC:76069 | |
| genetic reagent (*D. melanogaster*) | RNAi-ppk25 | Bloomington *Drosophila* Stock Center | BDSC:27088 | |
| genetic reagent (*D. melanogaster*) | RNAi-ProcR | Bloomington *Drosophila* Stock Center | BDSC:29414 | |
| genetic reagent (*D. melanogaster*) | RNAi-ProcR | Bloomington *Drosophila* Stock Center | BDSC:29570 | |
| genetic reagent (*D. melanogaster*) | RNAi-ptp69D | Bloomington *Drosophila* Stock Center | BDSC:29462 | |
| genetic reagent (*D. melanogaster*) | RNAi-ShakB | Bloomington *Drosophila* Stock Center | BDSC:27292 | |

*Continued on next page*

*Continued*

| Reagent type (species) or resource | Designation | Source or reference | Identifiers | Additional information |
|---|---|---|---|---|
| genetic reagent (*D. melanogaster*) | RNAi-SifA | Bloomington *Drosophila* Stock Center | BDSC:29428 | |
| genetic reagent (*D. melanogaster*) | RNAi-SifA | Bloomington *Drosophila* Stock Center | BDSC:60484 | |
| genetic reagent (*D. melanogaster*) | RNAi-Tbh | Bloomington *Drosophila* Stock Center | BDSC:27667 | |
| genetic reagent (*D. melanogaster*) | RNAi-Tbh | Bloomington *Drosophila* Stock Center | BDSC:67968 | |
| genetic reagent (*D. melanogaster*) | RNAi-Tdc2 | Bloomington *Drosophila* Stock Center | BDSC:25871 | |
| genetic reagent (*D. melanogaster*) | RNAi-Tk | Bloomington *Drosophila* Stock Center | BDSC:25800 | |
| genetic reagent (*D. melanogaster*) | RNAi-TkR86C | Bloomington *Drosophila* Stock Center | BDSC:31884 | |
| genetic reagent (*D. melanogaster*) | RNAi-TkR99D | Bloomington *Drosophila* Stock Center | BDSC:27513 | |
| genetic reagent (*D. melanogaster*) | RNAi-trh | Bloomington *Drosophila* Stock Center | BDSC:25842 | |
| genetic reagent (*D. melanogaster*) | RNAi-tutl | Bloomington *Drosophila* Stock Center | BDSC:54850 | |
| genetic reagent (*D. melanogaster*) | RNAi-TyrR | Bloomington *Drosophila* Stock Center | BDSC:25857 | |
| genetic reagent (*D. melanogaster*) | RNAi-TyrR | Bloomington *Drosophila* Stock Center | BDSC:57296 | |
| genetic reagent (*D. melanogaster*) | RNAi-TyrRII | Bloomington *Drosophila* Stock Center | BDSC:27670 | |
| genetic reagent (*D. melanogaster*) | RNAi-TyrRII | Bloomington *Drosophila* Stock Center | BDSC:64964 | |
| genetic reagent (*D. melanogaster*) | ry506 | Bloomington *Drosophila* Stock Center | BDSC:225 | |
| genetic reagent (*D. melanogaster*) | RyaR- | Bloomington *Drosophila* Stock Center | BDSC:84571 | |
| genetic reagent (*D. melanogaster*) | shakB-Gal4 | Bloomington *Drosophila* Stock Center | BDSC:51633 | |
| genetic reagent (*D. melanogaster*) | SifA-Gal4 | Bloomington *Drosophila* Stock Center | BDSC:84690 | |
| genetic reagent (*D. melanogaster*) | sNPF- | Bloomington *Drosophila* Stock Center | BDSC:84574 | |
| genetic reagent (*D. melanogaster*) | SS00078-Gal4 | Janelia Research Center | JRC:SS00078 | |
| genetic reagent (*D. melanogaster*) | SS00090-Gal4 | Janelia Research Center | JRC:SS00090 | |
| genetic reagent (*D. melanogaster*) | SS00097-Gal4 | Janelia Research Center | JRC:SS00097 | |
| genetic reagent (*D. melanogaster*) | SS00117-Gal4 | Janelia Research Center | JRC:SS00117 | |
| genetic reagent (*D. melanogaster*) | SS01566-Gal4 | Janelia Research Center | JRC:SS01566 | |

*Continued on next page*

*Continued*

| Reagent type (species) or resource | Designation | Source or reference | Identifiers | Additional information |
|---|---|---|---|---|
| genetic reagent (*D. melanogaster*) | SS02214-Gal4 | Janelia Research Center | JRC:SS02214 | |
| genetic reagent (*D. melanogaster*) | SS02216-Gal4 | Janelia Research Center | JRC:SS02216 | |
| genetic reagent (*D. melanogaster*) | SS02255-Gal4 | Janelia Research Center | JRC:SS02255 | |
| genetic reagent (*D. melanogaster*) | SS02391-Gal4 | Janelia Research Center | JRC:SS02391 | |
| genetic reagent (*D. melanogaster*) | SS27853-Gal4 | Janelia Research Center | JRC:SS27853 | |
| genetic reagent (*D. melanogaster*) | SS50464-Gal4 | Janelia Research Center | JRC:SS50464 | |
| genetic reagent (*D. melanogaster*) | SS52578-Gal4 | Janelia Research Center | JRC:SS52578 | |
| genetic reagent (*D. melanogaster*) | Tbh- | Bloomington *Drosophila* Stock Center | BDSC:56660 | |
| genetic reagent (*D. melanogaster*) | Tdc-Gal4 | Bloomington *Drosophila* Stock Center | BDSC:9313 | |
| genetic reagent (*D. melanogaster*) | tim-Gal4 | Bloomington *Drosophila* Stock Center | BDSC:80941 | |
| genetic reagent (*D. melanogaster*) | Trh- | Bloomington *Drosophila* Stock Center | BDSC:10531 | |
| genetic reagent (*D. melanogaster*) | Trh-Gal4 | Bloomington *Drosophila* Stock Center | BDSC:38388 | |
| genetic reagent (*D. melanogaster*) | Trh-Gal4 | Bloomington *Drosophila* Stock Center | BDSC:38389 | |
| genetic reagent (*D. melanogaster*) | tutl-Gal4 | Bloomington *Drosophila* Stock Center | BDSC:63344 | |
| genetic reagent (*D. melanogaster*) | tutl-Gal4/CyO;Cha-Gal80 | Derived from BDSC:63344 and *Cha-Gal80* | | |
| genetic reagent (*D. melanogaster*) | tutl1/CyO | Kendal Broadie (***Bodily et al., 2001***) | | |
| genetic reagent (*D. melanogaster*) | TyrR- | Bloomington *Drosophila* Stock Center | BDSC:27797 | |
| genetic reagent (*D. melanogaster*) | TyrRII- | Bloomington *Drosophila* Stock Center | BDSC:23837 | |
| genetic reagent (*D. melanogaster*) | UAS-CsChrimson | Bloomington *Drosophila* Stock Center; ***Klapoetke et al., 2014*** | BDSC:55135 | |
| genetic reagent (*D. melanogaster*) | UAS-DTI | Bloomington *Drosophila* Stock Center | BDSC:25039 | |
| genetic reagent (*D. melanogaster*) | UAS-eGFP-Kir2.1.FRT.mCherry | David Anderson; ***Watanabe et al., 2017*** | | |
| genetic reagent (*D. melanogaster*) | UAS-hid | Bloomington *Drosophila* Stock Center | BDSC:65403 | |
| genetic reagent (*D. melanogaster*) | UAS-Kir2.1 | Jess Kanwal; ***Baines et al., 2001*** | | |
| genetic reagent (*D. melanogaster*) | UAS-mcd8GFP | Bloomington *Drosophila* Stock Center | BDSC:32185 | |

*Continued on next page*

*Continued*

| Reagent type (species) or resource | Designation | Source or reference | Identifiers | Additional information |
|---|---|---|---|---|
| genetic reagent (*D. melanogaster*) | UAS-mCherry.FRT.eGFP-Kir2.1 | David Anderson; *Watanabe et al., 2017* | | |
| genetic reagent (*D. melanogaster*) | UAS-NiPP1 | Bloomington *Drosophila* Stock Center | BDSC:23711 | |
| genetic reagent (*D. melanogaster*) | UAS-PdfRg/CyO; UAS-Cas9/TM6B | Matthias Schlichting; *Schlichting et al., 2019* | | |
| genetic reagent (*D. melanogaster*) | UAS-syt-eGFP, DenMark | Bloomington *Drosophila* Stock Center | BDSC:33064 | |
| genetic reagent (*D. melanogaster*) | UAS-TNT-C | Bloomington *Drosophila* Stock Center | BDSC:28996 | |
| genetic reagent (*D. melanogaster*) | UAS-TNT-E | Bloomington *Drosophila* Stock Center | BDSC:28837 | |
| genetic reagent (*D. melanogaster*) | UAS-TNT-G | Bloomington *Drosophila* Stock Center | BDSC:28838 | |
| genetic reagent (*D. melanogaster*) | UAS-TrpA1 | Bloomington *Drosophila* Stock Center | BDSC:26263 | |
| genetic reagent (*D. melanogaster*) | VT002215-Gal4 | Janelia Research Center | JRC:VT002215 | |
| genetic reagent (*D. melanogaster*) | VTDh44-Gal4/TM3, Sb | VT039046 (via Daniel Cavanaugh) | | |
| genetic reagent (*D. melanogaster*) | w; Aug21-Gal4, UAS-GFP/CyO | Derived from *C(1)Dxyfv(X^X)/Y; Aug21-Gal4, UAS-GFP/CyO* | | |
| other | Acetone | Sigma | Cat#: 179124 | Vehicle for Methoprene and Precocene I |
| other | Fluvastatin | Sigma | Cat: PHR1620 | Mevalonate synthesis pathway inhibitor |
| other | Hoechst 33342 | Thermo Fisher | Cat#: H-3570 | IF(1:1000) |
| other | Methoprene | Sigma | Cat#: 33375 | JH analog |
| other | Precocene I | Sigma | Cat#: 195855 | CA inhibitor |
| other | Pyriproxyfen | Sigma | Cat#: 34174 | JH analog |
| other | RhoB | Sigma | Cat#: R6626 | (1.44 mg/mL) |
| peptide, recombinant protein | Phalloidin | Thermo Fisher | Cat#: A-12380 | IF(1:400) |
| software, algorithm | MARGO | *Werkhoven et al., 2019* | | |
| strain (Entomophthora muscae) | Entomophthora muscae | *Elya et al., 2018* | ARSEF #13514 | |

## Fly stocks and husbandry

All fly stocks were maintained in vials on cornmeal-dextrose media (11% dextrose, 3% cornmeal, 2.3% yeast, 0.64% agar, 0.125% tegosept [w/v]) at 21 °C and ~40% humidity in Percival incubators under 12 hr light and 12 hr dark lighting conditions and kept free of mites. All fly stocks used for experiments are listed in Key Resources Table, designations for screened lines are given in *Supplementary file 1*, and full genotype information by figure panel is given in *Supplementary file 2*. Imaging and metabolomic data are from female flies and behavior data come from mixed-sex populations, unless otherwise specified in the text.

## *E. muscae* husbandry

A continuous *in vivo* culture of *E. muscae* 'Berkeley' (referred to herein as *E. muscae*; USDA ARSEF#13514) isolated from wild Drosophilids (*Elya et al., 2018*) was maintained in Canton-S flies

cleared of *Wolbachia* bacteria following the protocol described in *Elya et al., 2018* and summarized as follows. Canton-S flies were reared in bottles containing cornmeal-dextrose media (see Fly stocks and husbandry) at 21 °C and ~40% humidity under 12 hr light and 12 hr dark lighting conditions. *E. muscae*-killed flies were collected daily between ZT15 and ZT18 using $CO_2$ anesthesia. To infect new Canton-S flies, 30 fresh cadavers were embedded head first in the lid of a 60 mm Petri dish filled with a minimal medium (autoclaved 5% sucrose, 1.5% agar prepared in milliQ-purified deionized water, aka '5AS'). Approximately 330 mg of 0–5 day-old Canton-S flies were transferred to a small embryo collection cage (Genesee #59–100, San Diego, CA) which was topped with the dish containing the cadavers. The cage was placed mesh-side down on a grate propped up on the sides (to permit airflow into the cage) within an insectrearing enclosure (Bugdorm #4F3030, InsectaBio, Riverside, CA) and incubated at 21 °C, ~40% humidity on a 12:12 L:D cycle. After 24 hr, the cage was inverted and placed food-side down directly on the bottom of the insect enclosure. After 48 hr, the cadaver dish was removed from the cage and replaced with a new dish of 5AS without cadavers. Starting at 96 hr, the collection cage was checked daily for up to four days between ZT15 and ZT18 for *E. muscae*-killed flies. These were collected using $CO_2$ anesthesia and used to infect additional flies for experiments as described below.

## Summit behavior box design and fabrication

The summit assay box was designed in Adobe Illustrator in the style of other high throughput behavioral assays used by our lab (See *Werkhoven et al., 2021*; https://github.com/de-Bivort-Lab/dblab-schematics). Nine behavior boxes were assembled from laser-cut acrylic and extruded aluminum railing (80/20 LLC). Each box consists of a ⅛" black acrylic base supporting an edge-lit dual-channel white (5300 K) and infrared (850 nm) light LED board (KNEMA, Anyang City, South Korea), three ⅛" black acrylic sides, a ¼" black hinged door and a ⅛" black ceiling upon which is mounted a digital camera (ELP #USB130W01MT-FV, Shenzhen, China) equipped with an 87° C Wratten infrared longpass filter (B&H Video #KO87C33O, New York City, New York). The summit arenas sit on a ⅛" clear acrylic board held 6–7 cm above the illuminator by fasteners in the aluminum rail supports. 850 nm infrared illumination (invisible to flies) is used for tracking and white illumination (visible to flies) provides 12 hr light:dark circadian cues. Intensity of infrared and white light was independently controlled by pulsewidth modulation via a Teensy (v3.2, PJRC, Sherwood, OR) microcontroller mounted to a custom printed circuit board (PCB) (*Werkhoven et al., 2019*). Each box's camera and PCB connect to a dedicated Lenovo mini-tower PC running Windows 10 and Matlab v.2018b equipped with MARGO v.1.03, Matlab-based software optimized to track many objects simultaneously, to record centroid positions for each of the assayed flies (*Werkhoven et al., 2019*). A complete list of parts and instructions for fabricating a summing box can be found at https://github.com/de-Bivort-Lab/dblab-schematics/tree/master/Summit_Assay copy archived at *de Bivort Lab, 2023*.

## Summiting behavior arena designs

Several different arena variants were used in the summiting assay tracking boxes. All arenas were fabricated in arrays in acrylic trays that fit snugly into the assay boxes. Each arena includes a small hole at one end through which a fly can be aspirated and subsequently sealed using a small cotton ball. Arenas were 3.2 mm tall, allowing flies to walk freely and raise their wings, the final manipulation by *E. muscae*.

An early prototype summiting assay was angled at 90°, but we found that even with a sandpaper-roughened walking surface, dying flies struggled to maintain their grip on the vertical surface. This was manifested in two ways: (1) flies exhibited sudden, rapid downward movement in their behavioral traces consistent with falls and (2) *E. muscae*-killed flies were predominantly found at the bottom of the well at the end of the experiment. This was subsequently confirmed by reviewing videos taken from these experiments. To remedy this, we reduced the incline to 30°, which is sufficient for flies to respond behaviorally to the direction of gravity (M. Reiser, personal communication). This eliminated obvious falling bouts and yielded a wide range of final positions ranging from the bottom to the top of the arena.

## Standard arena (e.g. Figure 1F)

Standard arenas measured 6.5 cm long by 0.5 cm wide by 0.32 cm tall and housed a single fly. Arenas were constructed in rows of 32 from three layers of ⅛" laser-cut acrylic consisting of a clear base

manually roughened with 120 grit sandpaper, black walls, and a clear top. The layers were held together with 8–32 screws and nuts. A 3 mm loading hole in the lid at one end of the arena permitted the loading of an anesthetized fly with a paintbrush. This entry hole was sealed with a piece of dental cotton after the fly was loaded. A minimal medium, 5AS, was provided at the opposite end of the chamber. The end of the chamber with food was sealed with two layers of Parafilm to slow the desiccation of the food. Fully prepared (i.e. with food at the bottom and the loading hole sealed), the long axis of the arena had ~5 cm of open space. Each tray had four rows of arenas, for a total of 128 arenas per tray. Laser-cutting designs for the standard arenas are available at https://github.com/de-Bivort-Lab/dblab-schematics/tree/master/Summit_Assay (*de Bivort Lab, 2023*).

### Starvation arena (e.g. Figure 1—figure supplement 2A)

Starvation arenas were constructed as standard arenas, substituting 1.5% agar (no sucrose) for 5AS media.

### Desiccation arena (e.g. Figure 1—figure supplement 2B)

Desiccation arenas were constructed as standard arenas, except each arena was 6 cm tall (~5.7 cm effective height) and lacked food and any opening at the bottom for the introduction of food.

### Two-choice arena (e.g., Figure 1—figure supplement 2F)

Two choice arenas consisted of a five-layer acrylic sandwich secured with 8–32 fasteners: a bottom layer consisted of a ⅛″ clear base texturized with 120 grit sandpaper. The next two layers each consisted of 1/16″ black walls dividing the row into 32 chambers. These layers were rotated 180° with respect to each other, leaving gaps in the floor and ceiling at opposite ends of the arena that could be filled with media. Thus, the total height of the arena, except at the ends, was 1/8″. Each chamber was 4.6 cm long and contained 5AS at one end, and 1.5% agar at the other. The lid layer consisted of ⅛″ clear acrylic. Flies were loaded quickly into the arenas and the lid was placed before the flies could wake up. Each tray had four rows of arenas, for a total of 128 arenas per tray.

### Tall arena (e.g. Figure 1—figure supplement 2I)

Tall arenas were constructed in the same fashion as standard arenas but measured 13 cm high instead of 6.5 cm. Two rows of 30 tall arenas each filled each tray. Food was pipetted into the middle of each arena and allowed to cool before the arenas were inclined. Flies were loaded through a loading hole at one end of the arena. The hole was plugged with cotton, for an effective length of ~12.8 cm.

### Summiting behavior experiments with *E. muscae* exposed flies

All summiting experiments with *E. muscae*-exposed flies were run as follows (unless otherwise indicated): flies were exposed to *E. muscae* by first embedding eight sporulating Canton-S cadavers in a 2.3 cm-diameter disc of ~3.5 mm thick 5AS that was transferred with 6″ forceps into the bottom of an empty wide-mouth *Drosophila* vial (Genesee #32–118). A ruler was used to mark 1.5 cm above the top of the disc. 0–5-day-old flies of the experimental genotype were anesthetized with $CO_2$, and 35 (~half male, ~half female) were transferred into the vial. The vial was capped with a Droso-Plug (Genesee #59–201) which was pushed down into the vial until the bottom was level with the 1.5 cm mark. For each experimental tray, three vials of flies were prepared in this way to expose a total of 105 flies; one additional vial of 35 flies was prepared identically but omitted cadavers as a non-exposed control. Together, these four vials were sufficient to fill a tray of 128 arenas. All prepared vials were incubated in a humid chamber (a small tupperware lined with deionized water-wetted paper towels) at 21 °C on a 12:12 L:D cycle. After 24 hr, the vials were removed from the humid chamber, and the Droso-plugs were pulled to the top of the vial to reduce fly crowding.

After 48–72 hr in the incubator, flies were loaded into the arenas using $CO_2$ anesthesia. Flies loaded into arenas during scotophase (the dark period of their 12:12 L:D circadian cycle) were shielded from ambient light in a foil-lined cardboard box. To begin behavioral experiments, arena trays were placed in the summit assay box and flies were tracked starting between ZT17 and ZT20. Tracking proceeded until ZT13 the next day (day 4). If many flies remained alive, tracking continued until ZT13 the following day. Some experiments, particularly in periods of COVID-restricted lab access, ran unattended until ZT13 on day 6 or 7. This variation in the timing of the end of the experiment had no effect on our

measured outcomes, since all behavioral data were analyzed with respect to times of fly death, and any tracking data after death were ignored.

Tracking data were collected at 3 Hz using the circadian experiment template (https://github.com/de-Bivort-Lab/margo/tree/master/examples/Circadian; *Werkhoven, 2018*) in MARGO v1.03 (*Werkhoven et al., 2019*; https://github.com/de-Bivort-Lab/margo) with the following settings: white light intensity 50%, infrared between 70–100%, adjusted to provide the best contrast for tracking, tracking threshold = 18, minimum area = 10, min trace duration = 6. Default settings were used for other configuration parameters. After tracking concluded, flies were manually scored as either alive (coded as survival = 1 and outcome = 0), dead with evidence of E. muscae sporulation (survival = 0, outcome = 1), or dead with no E. muscae sporulation (survival = 0, outcome = 0). These annotations were saved in a metadata file accompanying each MARGO output file and used in downstream analyses.

## Summit behavior data analysis

For each tray of flies (N≤128), we generated an experiment metadata table that incorporated the manually-scored survival outcome described above as well as fly genotype, sex, and fungal exposure status (exposed or non-exposed). Experiment metadata along with tracking data were input into a Matlab-based analysis pipeline that proceeded through the following steps: (1) automatic denoising, (2) manual time of death calling, (3) behavioral trajectory alignment to time of death, (4) SM calculation, (5) effect size estimation. See http://lab.debivort.org/zombie-summiting/.

The automatic denoising algorithm scanned speed throughout the experiment and flagged any ROIs that exhibited more than 20 instances per day of experimental time greater than ~40 mm/s. This threshold was chosen based on the examination of individual ROI speed traces as a value that would only be exceeded with noise. The bulk of noisy behavioral recordings arose when the flies' position was erroneously tracked as moving along the long edges of the arenas. Denoising was achieved by reducing the horizontal width of the arena region-of-interest (ROI) and recalculating centroid trajectory until speed violations fell below the threshold or the ROI was trimmed to nine pixels, at which point its data was discarded.

Time of death was called manually for every cadaver (N=~23,500) by CE throughout this study by checking time-aligned plots of y position and speed. Time of death was estimated as the time the fly was last observed to exhibit walking behavior. Extremely slow changes in y-position and tracking jitter around a particular y-position were not considered to be walking behavior. These definitions were initially validated by comparing paired behavioral video and tracking data. ROIs were flagged if sparse tracking occurred or residual noise was so great that the time of death couldn't be reasonably determined. These ROIs were dropped in subsequent analysis. For the gene and Gal4 screen (*Figure 2B and C*), the scoring of time-of-death was not blind to the fly genotype; for all subsequent experiments, times of death were scored blind to the experimental group. Time of death was stored as a frame number in the experimental metadata file.

Denoised tracking data and experimental metadata with time-of-death calls were input into a script that performed the following tasks: (1) determined the earliest start time for all experiments and aligned all data relative to this timepoint. This was necessary as experiments were not all started at precisely the same time (e.g. one experiment may start at 5:08 pm, another at 5:24 pm); (2) categorized each fly-trajectory as either a zombie (cadaver), survivor (alive), or unexposed control (uninfected), based on experimental metadata; (3) randomly assigned a 'time of death' for survivor and control flies from the pool of observed times of death within cadavers for that genotype, to make data between groups more comparable; (4) align all fly behavioral (y position and speed) trajectories relative to their time of death; (5) output a variable containing aligned and original vectors of data by category (zombie, survivor, unexposed) for a given genotype.

To calculate the summit metric (SM) for each cadaver, we first determined the period of summiting. The beginning of summiting was defined as 2.5 hr before death. The speed trajectory was smoothed with a 1 hr sliding window average and the end of summiting was defined as the earliest moment when the smoothed speed dropped to the same level as the start of summiting. The speed trajectory was baseline corrected by subtracting the smoothed speed at the onset of summiting, and the area under the resulting curve during the period of summiting divided by the duration of summiting (end of summiting – the start of summiting) was taken as the value of SM. Thus, SM has units of distance/time and is a measure of speed.

## Statistical tests

Summing effect size estimate distributions were calculated by bootstrapping flies, separately in experimental and control groups, calculating the manipulation effect size as (mean(Experimental SM) – mean(Control SM))/mean(Control SM), over 1,00 resamplings. Distributions were plotted as kernel density estimates. Two-tailed unpaired t-tests were used to assess the significance of differences between SM in experimental and control groups. All reported p-values are nominal. Confidence intervals on time-varying data were calculated by bootstrapping individual flies over 1000 replicates and shading the original mean values and +/−1 standard deviation of the bootstrapped means.

## Thermogenetic activation of DN1p and PI-CA

Unexposed flies (up to 8 days post eclosion) were loaded into standard summing arenas (5AS food placed at y position = 0, 30° incline) and were tracked starting at ~ZT17 in a temperature-controlled room initially held at 21 °C, below the activation temperature of TrpA1 (*Hamada et al., 2008*). At ZT5:30 the following day, the temperature setpoint of the environmental room was increased to 28 °C. The room took approximately 30 min to reach the setpoint temperature. Temperature in the room was monitored via a Bluetooth Thermometer (Govee #H5075). At ZT7:30 (2 hr after the initial setpoint change), the setpoint was returned to 21 °C. Flies were tracked until ~ZT13, for a total tracking time of 20 hr. Temperature measurements taken concurrently with behavioral tracking were used to generate the heatmap strips in *Figure 2I and J*, etc.

## Optogenetic activation of PI-CA

Young (up to 3 days post eclosion), unexposed UAS-CsChrimson/+; R19G10-Gal4/+flies were placed in narrow (24.8 mm diameter) foil-wrapped vials, in which either 10 µL of 100 mM all-trans-retinal (ATR; Sigma #R2500) in ethanol, a required cofactor for CsChrimson, or 10 µL of 70% ethanol had been applied to the surface of the food. Flies in both groups were transferred to freshly-applied ATR/ethanol vials every 2 days. After 8 days, flies were tracked in individual, circular 28 mm diameter arenas (*Werkhoven et al., 2021*) using MARGO under IR illumination. For *Figure 2K*, *Figure 2—figure supplement 2G*, flies were tracked for 30 min. After 15 min of tracking in darkness, constant red light (3.15 µW/mm$^2$) was projected onto the behavioral arenas using an overhead-mounted modified DLP projector (*Werkhoven et al., 2019*). For *Figure 2—figure supplement 2E and F*, the red light was delivered in 5 ms pulses at 5 Hz for 30 seconds using the same projector under the control of the MATLAB PsychToolBox package (http://psychtoolbox.org/). Each 30 s pulsed red light trial was followed by 65 s of darkness (the projector light path was manually blocked with black acrylic during these periods), for 38 trials, totaling 1 hr of tracking.

## Immunohistochemistry

Tissues (brains, ventral nerve cords, and/or anterior foreguts with retrocerebral complexes) were dissected in 1 x PBS from female flies and stained generally following the Janelia FlyLight protocol (*Janelia FlyLight Team, 2015*) as follows. Fixation, incubation, and washing all took place under gentle orbital shaking. Tissues were fixed in 2% paraformaldehyde for 55 min at room temperature in 2 mL Protein LoBind tubes (Eppendorf #022431064, Enfield, CT). Fixative was removed and tissues were washed 4x10 min with 1.5 mL PBS with 0.5% Triton X-100 (PBT). Tissues were then blocked for 1.5 hr at room temperature in 200 µL of PBT with 5% normal goat serum (NGS) before adding primary antibodies prepared at the indicated dilutions in PBT with 5% NGS (Key Resources). Tissues were incubated with primary antibodies for up to 4 hr at room temperature then placed at 4 °C for at least 36 hr and no more than 108 hr. Primary antibody solution was removed and samples were washed at room temperature at 3x30 min in 1.5 mL PBT. Tissues were then incubated in 200 µL of PBT containing 5% NGS and secondary antibodies (Key Resources) for 2-4 hr at room temperature before moving to 4 °C for approximately 60 hr. Secondary antibody solution was removed and tissues were washed 3x 30 min in PBT. Samples were then mounted in a drop of Vectashield (Vector Laboratories #H-1200–10, Newark, CA) placed within one or more 3-ring binder reinforcer stickers, which served as a coverslip bridge. Slides were sealed with nail polish and stored in the dark at 4 °C until imaging on an LSM 700 confocal microscope (Zeiss, Oberkochen, Germany) in the Harvard Center for Biological Imaging.

## Genetic ablation of CA

CA of adult flies was completely or partially ablated following the methods of *Bilen et al., 2013* and *Yamamoto et al., 2013*, respectively. For complete ablation (*Figure 3C and D*, *Figure 3—figure supplement 1D*), virgin females of genotype *Aug21-Gal4,UAS-GFP/CyO* were crossed to males of genotype *UAS-DTI/CyO; tub-Gal80^ts^/TM6B* and reared at 21 °C until progeny reached third wandering instar. At this point, progeny were either transferred to 29 °C until eclosion or kept at 21 °C. Progeny of the genotype *Aug21-Gal4,UAS-GFP/UAS-DTI; tub-Gal80^ts^/+* were then exposed to *E. muscae* and run in the summit behavior assay. In separate experiments to assess ablation efficiency, experimental and control female flies (N=5) were dissected and examined using a compound epifluorescence microscope (80i, Nikon, Melville, NY).

For partial CA ablation (*Figure 3—figure supplement 1B and C*), virgin females of genotype *C(1)Dxyfv(X^X)/Y;Aug21-Gal4, UAS-GFP/CyO* were crossed to *UAS-NiPP1* males at 29 °C. Experimental flies (*C(1)Dxyfv(X^X)/Y; Aug21-Gal4,UAS-GFP/+;UAS-NiPP1/+*) and sibling controls (*C(1)Dxyfv(X^X-)/Y;Aug21-Gal4, UAS-GFP/+;TM6C/+*) were exposed to *E. muscae* and run in the summit behavior assay. To assess ablation efficiency, experimental and control female flies (N≥7) were subjected to immunohistochemistry using anti-GFP and anti-nc82 primary antibodies and imaged on an LSM 700 confocal microscope (Zeiss).

## Pharmacological perturbation of CA

Precocene I (Sigma #195855) and methoprene (Sigma #33375) were diluted in acetone (Sigma #179124) and applied topically to the ventral abdomen of $CO_2$-anesthetized flies that had been exposed to *E. muscae* (72 hr prior) or mock unexposed controls. 0.2 µL of the compounds were applied per fly using a 10 µL Hamilton syringe (Hamilton #80075, Reno, NV) with a repeater attachment (Hamilton #83700). Acetone-only flies served as a vehicle control. To avoid compounds cross-contaminating flies, anesthetized flies were placed on top of two layers of fresh filter paper and handled with a reagent-dedicated paint brush as soon as they had been dosed with the desired compound. The syringe was thoroughly flushed with acetone between compounds.

Solutions of pyriproxyfen (Sigma #34174, dissolved in ethanol) and fluvastatin (Sigma #PHR1620, dissolved in ultrapure water) were individually pipetted onto the media in standard summit arenas prepared with 5AS in 5 µL volumes using a 250 µL Hamilton syringe (Hamilton #81101) and repeater attachment. Five µL of either ethanol or water were applied to a second set of arena media to serve as vehicle controls for pyriproxyfen and fluvastatin, respectively. Arenas were then parafilm-sealed and stored at 4 °C overnight. The following day, chambers were allowed to warm to room temperature before introducing flies for summit behavior assays.

## Real-time summiting classifier

A ground truth dataset was pooled from 14 experiments comprising 1306 mixed-sex Canton-S flies exposed to *E. muscae* (961 survivors and 345 zombies). These data were processed into 61-dimensional feature vectors, each representing an individual fly's behavior up to a particular time of observation. The variables in the feature vector were as follows:

- Feature 1: the time of observation since the start of the experiment (in hours).
- Features 2–11: historical y position values at 10 frames logarithmically spaced between the start of the experiment and 10 min prior to the time of observation. frames near the start of the experiment are chosen more sparsely than more recent frames. See *Figure 4B*.
- Features 12–21: historical fly speed, at the same logarithmically-sampled frames as described above.
- Features 22–41: recent y position at frames uniformly spaced between the time of observation and 10 min prior.
- Features 42–61: recent fly speed, at the same uniformly-sampled frames as described above.

Two hundred feature vectors were generated for each fly by selecting 200 random times of observation uniformly distributed across the experiment. Thus, the dataset might independently include a feature vector for fly A at ZT13:30 as well as fly at ZT8:00. This yielded a total of 261,200 vectors.

Each feature vector was paired with one of four summiting labels (never-summit, pre-summiting, during summiting, or post-summiting). The resultant dataset of 61-dimensional feature vectors and summiting status labels was then randomly subdivided: 75% were used to train a random-forest

classifier, and the remaining 25% were withheld as a validation set to evaluate classifier performance. We varied the random forest parameters until satisfactory classifier performance was achieved. At this point, the classifier was tested on a novel experimental dataset generated from a single summiting behavior experiment to assess performance.

In experiments utilizing the classifier in real-time, a fly was called as summiting as soon as the predicted during-summiting label probability exceeded the predicted non-infected probability for three consecutive prediction frames (a span of 8 min). For experiments requiring paired non-summiting control flies for each flagged summiting fly, five non-summiting candidates were chosen by picking the flies with the highest 'non-summiting' score, constructed by multiplying the following four factors:

- the average never-summit label probability over the duration of the experiment
- 1 - the maximum predicted during-summiting probability
- whether the fly was moving at least 10% of all frames in the experiment so far
- the current speed percentile

These factors were chosen heuristically to boost active flies showing few signs of summiting.

## Brain and CA morphology during summiting

Female summiting flies were identified in real-time using the random forest classifier, then quickly collected from the summiting assay using a vacuum-connected aspirator and anesthetized with $CO_2$ before being placed on ice. These flies were harvested no earlier than ZT12 on the fourth or fifth day following *E. muscae* exposure. Tissues were dissected and kept ice cold until they were mounted in Vectashield to monitor endogenous fluorescence (in the case of Aug21 >GFP flies) or subjected to fixation and subsequent immunohistochemistry (HisRFP and R19G10>mcd8 GFP flies).

Corpora allata of Aug21-GFP summiting females were dissected by gently separating the head from the thorax to expose the esophagus and proventriculus. The foregut was severed posterior to the proventriculus and the tissue was mounted in a drop of Vectashield deposited in the middle of three stacked 3-hole reinforcer stickers on a #1 22 × 22 mm coverslip with the back of the head (posterior side) down. The coverslip was then mounted on an untreated glass slide by gently lowering the slide onto the coverslip until adhesion. The slide was then inverted and imaged at 10 x magnification on an upright epifluorescent compound microscope (Nikon 80i) using a constant exposure across samples (300 ms).

Fungal nuclei (HisRFP brains: Hoechst positive, HisRFP negative) or neuropil holes (R19G10>mcd8 GFP brains: oval voids) were manually counted in three brain-wide z-stacks (2 µm z-step) of HisRFP brains using FIJI (*Schindelin et al., 2012*). All fungal nuclei were counted in each plane. A comparison of the fraction of nuclei using the manual 'raw' method (counting every nucleus across every plane) to an estimate of the actual number of nuclei (via computational collapsing of nuclei counts if their centers are within 2 µm in x and y dimensions and 10 µm in z) showed both methods gave comparable estimates of the distribution of fungal nuclei across brain regions (*Figure 5—figure supplement 1C*, D). Therefore, raw counts were used. Pars intercerebralis cell bodies (R19G10>mcd8 GFP brains) were counted in Zen Blue (Zeiss). Each cell body was counted only once, since for this analysis we were investigating the total number of these cells, not their distribution.

## Whole body morphology during the end of life

*E. muscae*-exposed Canton-S flies were manually staged at five distinct end-of-life stages and subjected to paraffin embedding, histology, and microscopy in Michael Eisen's lab at UC Berkeley. Briefly, flies were transferred at 72 hr after exposure to *E. muscae* to individual 500 µL Eppendorf tubes prepared with 100 µL of permissive medium and a ventilation hole poked in the lid with an 18 gauge needle. Flies were manually monitored from ZT8 to ZT13 and immediately immersed in fixative when the following behaviors were first observed: (1) cessation of flight (fly appears to walk normally but does not fly when provoked by the experimenter; corresponds to mid or late summiting), (2) cessation of walking (fly continues to stand upright with proboscis retracted but no longer initiates sustained walking behavior in response to provocation), (3) proboscis extension (proboscis is extended but wings remain horizontal), (4) mid-wing raise (proboscis is extended and wings are approximately half-raised), (5) full-wing raise (proboscis is extended and wings have stopped raising). Paraffin-embedded flies were sliced into eight-micron sections and stained with safranin and fast green to visualize interior structures (*Elya and Martinez, 2017*). Two flies were sectioned for each stage, one sliced sagittally

and the other coronally, and imaged on a Zeiss Axio Scan.Z1 Slide Scanner at the Molecular Imaging Center at UC Berkeley.

## Blood-brain barrier integrity

Canton-S flies were exposed to *E. muscae* or housed under mock exposure conditions as previously described. At ZT14 on day four following exposure, ~50 exposed female flies exhibiting extensive abdominal fungal growth with very white and opaque abdomens ('creamy-bellied'), ~50 exposed female flies of normal appearance, and ~50 unexposed controls were injected in the mesopleuron with a cocktail of rhodamine B (1.44 mg/mL, Sigma #R6626) and 10 KDa dextran conjugated to Cascade Blue (20 mg/mL, ThermoFisher #D1976) using a pulled glass capillary needle mounted in a brass needle holder (Tritech Research #MINJ-4, Los Angeles, CA) connected to a 20 mL syringe. The dye cocktail was injected until the anterior abdomen was visibly colored, but not with so much as to completely fill the body cavity and lead to proboscis extension. The volume of injected dye was approximately 75 nL per fly. Injected flies were transferred to foil-wrapped vials containing 5AS to recover. Foil-wrapped vials were placed in an opaque box to further minimize light exposure. After 4 hr, flies were anesthetized with $CO_2$, and their eye fluorescence was scored by an experimenter blind to experimental treatment. Prior to assessing eye fluorescence, flies were screened for rhodamine B fluorescence in the whole body. Flies with weak whole-body fluorescence were excluded from scoring as they were not loaded with enough dye. Flies were considered 'bright-eyed' if there was fluorescence across the entire eye and 'dark-eyed' if fluorescence was only apparent at the pseudopupil. Eye fluorescence was used to infer that RhoB was in the brain (*Mayer et al., 2009*).

## Metabolomics of summiting flies

In two separate experiments, hemolymph was extracted from summiting, non-summiting, and unexposed female flies. In the first experiment, summiting and non-summiting flies were identified manually. This was achieved by releasing *E. muscae*-exposed flies at ~ZT17 of the third or fourth day following exposure into a large insect-rearing cage (Bugdorm #BD4F3030) and continuous visual monitoring of flies from ZT8:30 until ZT11:30 the following day for signs of infection (creamy belly and lack of flight upon provocation). Flies that did not fly and/or right themselves after being provoked by the experimenter were designated summiting and collected. For each summiting fly collected, one exposed fly that did respond to provocation (non-summiting) and one unexposed fly (kept in a separate enclosure, unexposed) were collected simultaneously. All flies were retrieved from their enclosures using mouth aspiration, then stored on ice in Eppendorf tubes until a total of 20 flies had been collected. This was repeated to obtain duplicate pools of 20 flies for each infection status (summiting, non-summiting, and unexposed).

For the second experiment, summiting and non-summiting flies were identified in real-time using the random forest classifier. *E. muscae*-exposed females were loaded into standard summit arenas on the third or fourth evening following *E. muscae* exposure and tracked until ZT13 of the following day. Summiting and non-summiting flies were flagged in pairs automatically and the experimenter was alerted by email. Flies were promptly collected using a vacuum-assisted aspirator then briefly anesthetized with $CO_2$ and placed in 1.7 mL Eppendorf tubes on ice until twenty individuals were collected per treatment. An unexposed control fly was collected simultaneously with every summiting/non-summiting pair. Triplicate pools of 20 flies were collected for each infection status.

Hemolymph was extracted from a pool of 20 flies by piercing the mesopleuron of each with a 0.2 Minutien pin (Fine Science Tools #26002–20, Foster City, CA) mounted on a nickel-plated pin holder (Fine Science Tools #26018–17) under $CO_2$ anesthesia (*Musselman, 2013*). Pierced flies were transferred to a 500 µL microcentrifuge tube pierced at the bottom with a 29 ½ gauge needle nested in a 1.7 mL Eppendorf tube. Tubes were centrifuged at room temperature for 10 min at 2700 g to collect a droplet of hemolymph. Hemolymph was stored on ice until all samples had been extracted. Samples for metabolomic analysis were 1 µL of hemolymph added to 2 µL of 1 x PBS.

Metabolite detection and putative compound identification were performed by the Harvard Center for Mass Spectrometry. Hemolymph samples were brought to a final volume of 20 µL with the addition of acetonitrile, to precipitate proteins. Following centrifugation, 5 µL of supernatant was separated on a SeqQuant Zic-pHILIC 5 µm column (Millipore #150460, Temecula, CA). For each experiment, solvent mixtures comprising 20 mM ammonium carbonate, 0.1% ammonium hydroxide in water (solvent A),

and 97% acetonitrile in water (solvent B) flowed for ~50 min at 40 °C. For the manually-staged experiment, the following solvent mixtures flowed at 0.2 mL per minute: 100% B (20 min), 40% B, 60% A (10 min), 100% A (5 min), 100% A (5 min), 100% B (10 min). For the classifier-staged experiment, the following solvent mixtures flowed at 0.15 mL per min: 99% B (17 min), 40% B+ 60 % A (10 min), 100% A (5 min), 100% A (4 min), 99% B (11 min). For the manually-staged experiment, separated compounds were fragmented using electrospray ionization (ESI+) and detected using a Thermo Fisher Q-exactive mass spectrometer under each positive and negative polarity (Resolution: 70,000, AGC target: 3e6, mz range: 66.7–1000). For the classifier-staged experiment, separated compounds were fragmented using heated electrospray ionization (HESI+) and detected using a ThermoFisher Orbitrap ID-X mass spectrometer under each positive and negative polarity (Resolution: 500,000, AGC target: 1e5, mz range: 65–1000). The variations in flow rate and ionization protocol were unlikely to substantially affect the compounds we were able to detect between the experiments.

MS-MS was performed twice (once each for the manual and classifier-staged experiments) on mixed pools (5 µL of each of the three samples per experiment) using AcquireX DeepScan in each positive and negative mode and 2-level depth. All data were normalized (median centering) to compensate for biomass differences and analyzed with Compound Discoverer v. 3.1 (Thermo Fisher). Molecular formulae were predicted from measured mass and isotopic pattern fit. Abundance values were determined for every peak observed within the MS-MS experimental pool for every sample. All chromatograms were manually checked to distinguish likely real signal from noise, with compounds typically considered absent from a sample if intensity counts were <1e3. Putative compound identities were manually assigned from high-confidence database matches (MZcloud, MZvault, HCMS locally-curated mass list) based on accurate mass and MS-MS spectra. Compounds were considered to be observed in both experiments (manually-staged and classifier-staged) if their molecular weights were within 5 ppm. All MS data are available in *Supplementary file 3*.

## Hemolymph transfusion

Three and four days prior to the transfusion experiment, mixed-sex Canton-S flies were exposed to *E. muscae* in cages as described above. One day prior to the transfusion experiment, flies destined to receive hemolymph (either unexposed Canton-S males or 72 hr exposed Canton-S females; *Figure 6C and D*) were transferred into individual housing consisting of PCR tubes containing ~100 µL of 5AS and with two holes poked in the cap using an 18 gauge needle to provide airflow. Donor flies (females exposed ~72 or ~96 hr prior) was loaded into standard summiting arenas. Donor tracking began at ~ZT17 and the summiting classifier was launched.

The next day, two experimenters, A and B for the purposes of this explanation, implemented the transfusion experiment from ZT8 until ZT12 or until 32 pairs of recipient flies had been transfused. Experimenter A collected donor flies; experimenter B performed the transfusions. Each transfusion began when a fly was flagged by the classifier (see *Real-time summiting classifier*) and Experimenter A was alerted via email. Experimenter A inspected behavioral traces to confirm the accuracy of the summiting classification and selected one of five identified non-summiters to serve as a time-matched control. Experimenter A then collected these flies from arenas via vacuum-assisted aspiration, anesthetized them with $CO_2$, and placed them in adjacent wells of a 96-well plate on ice. Fly placement was randomized (i.e. sometimes the summiting fly was placed first, sometimes the non-summiting fly) and recorded before the plate was passed to Experimenter B. Thus, Experimenter B was blind to fly summiting status. Experimenter B then used a pulled capillary needle to remove ~50 nL of hemolymph from the first donor through the mesopleuron and injected this material into the mesopleuron of a cold-anesthetized recipient. The needle was rinsed thoroughly in molecular-grade water between transfusions. Immediately after transfusion, recipient flies were transferred to standard summiting arenas that were already in place in an imaging box and being tracked by MARGO. Tracking continued for no less than 3 hr after the final fly had been transfused. Used donor flies were transferred into individual housing and monitored for the next 48 hr for death by *E. muscae*.

Behavioral data were processed blind by Experimenter B. The time of recovery from anesthesia (i.e. resumption of locomotion) was manually determined for each fly based on its behavioral trace. Flies that did not recover or showed very little total movement were discarded from subsequent analysis. Fly summiting status was then revealed by Experimenter A to determine the average distance traveled vs time for each treatment group. After the data had been curated in this blinded manner,

Experimenter A revealed the behavior calls for each donor to Experimenter B. Experimenter B used this information as well as donor outcome to determine the average distance traveled for each treatment group. Donors that were identified as summiting but failed to sporulate on the day of the experiment were interpreted as misclassified and their corresponding recipients were dropped from the analysis.

## Acknowledgements

We thank Ryan Maloney and David Zimmerman for their helpful comments on the manuscript. We are indebted to Ed Soucy and Brett Graham of Harvard University's Center for Brain Science's magnificent Neuroengineering Core for their help in designing and fabricating the summiting behavior assay and to Charles Vidoudez of the Harvard Center for Mass Spectrometry for generating and curating hemolymph metabolomic data. We are also grateful to many folks for generously sharing reagents: Rochele Yamamoto (*Aug21-Gal4, UAS-GFP* flies), David Anderson (*UAS-mCherry.FRT.eGFP-Kir2.1* and *UAS-eGFP-Kir2.1.FRT.mCherry* flies), Daniel Cavanaugh (*VTDh44-Gal4*, *Clk856-Gal4*, and pan-DN1p-*Gal4* flies), Matthias Schlichting (*UAS-Cas9, UAS-PdfR-sgRNA* flies), Toshihiro Kitamoto (*Cha-Gal80* flies), Jess Kanwal (*UAS-Kir2.1* flies), Kristin Scott (*nan*[36a] and *tub-Gal80*[ts] flies), Kendal Broadie (*tutl*[1] flies), and Ryusuke Niwa (JHAMT primary antibody). We also would like to thank all of the core facilities that enabled this work: the Harvard Center for Mass Spectrometry (LC-MS), the Harvard Center for Biological Imaging (confocal imaging), UC Berkeley's Biological Imaging Facility (whole fly microtomy and histology), UC Berkeley's Molecular Imaging Center (whole fly histology imaging), the Kyoto *Drosophila* Stock Center (fly lines), Janelia Research Campus (fly lines), Bloomington *Drosophila* Stock Center (many, many fly lines). Finally, we are grateful to all of the hard-working individuals (i.e. *E. muscae*teers) who have helped keep the fungus thriving in the lab: Benno Rodemann, Fosca Bechthold, Aundrea Kroger, Nicole Pittoors, Ryan Maloney, Stanislav Lazopulo, Haesung Jee, Dylan Roy, and Noah Rodman.

## Additional information

### Funding

| Funder | Grant reference number | Author |
|---|---|---|
| Howard Hughes Medical Institute | GT11087 | Carolyn Elya |
| Alfred P. Sloan Foundation | Research Fellowship | Benjamin de Bivort |
| Esther A. and Joseph Klingenstein Fund | Klingenstein-Simons Fellowship Award | Benjamin de Bivort |
| Richard and Susan Smith Family Foundation | Odyssey Award | Benjamin de Bivort |
| Harvard/MIT | Basic Neuroscience Grant | Benjamin de Bivort |
| National Science Foundation | IOS-1557913 | Benjamin de Bivort |
| National Institute of Neurological Disorders and Stroke | 1R01NS121874-01 | Benjamin de Bivort |
| NSF-Simons Center for Mathematical and Statistical Analysis of Biology | 1764269 | Danylo Lavrentovich |
| Harvard Mind Brain and Behavior Initiative | Postdoctoral Fellow Award | Carolyn Elya |
| Harvard Quantitative Biology Initiative | | Danylo Lavrentovich |

| Funder | Grant reference number | Author |
|---|---|---|

The funders had no role in study design, data collection and interpretation, or the decision to submit the work for publication.

## Author contributions

Carolyn Elya, Conceptualization, Resources, Data curation, Software, Formal analysis, Supervision, Funding acquisition, Validation, Investigation, Visualization, Methodology, Writing – original draft, Project administration, Writing – review and editing; Danylo Lavrentovich, Resources, Data curation, Software, Formal analysis, Funding acquisition, Validation, Investigation, Visualization, Methodology, Writing – original draft, Writing – review and editing; Emily Lee, Cassandra Pasadyn, Jasper Duval, Investigation, Writing – review and editing; Maya Basak, Valerie Saykina, Investigation; Benjamin de Bivort, Conceptualization, Supervision, Funding acquisition, Visualization, Methodology, Writing – original draft, Project administration, Writing – review and editing

## Author ORCIDs

Carolyn Elya  http://orcid.org/0000-0002-9634-0303
Danylo Lavrentovich  http://orcid.org/0000-0002-8432-9596
Benjamin de Bivort  http://orcid.org/0000-0001-6165-7696

## Decision letter and Author response

Decision letter https://doi.org/10.7554/eLife.85410.sa1
Author response https://doi.org/10.7554/eLife.85410.sa2

## Additional files

### Supplementary files

• Supplementary file 1. Fly strains tested in summit inactivation screen. Genotypes are abbreviated for lines deposited at stock centers, for clarity. Stock centers are as follows: BDSC = Bloomington *Drosophila* Stock Center; KDSC = Kyoto *Drosophila* Stock Center; JRC = Janelia Research Campus. Functional/morphological annotations for the summit screen are abbreviated as follows: AM = AMMC; Ar = arousal; Ci = circadian; CX = central complex; Gr = gravitaxis; SO = subesophageal ganglion; MB = mushroom body; NM = neuromodulator & neurotransmitter; NP = neuropeptide; PI = pars intercerebralis.

• Supplementary file 2. Genotypes of flies in manuscript figures. Genotypes for deposited lines are abbreviated for clarity (i.e. interrupted alleles are designated [-], most y and w alleles have been omitted). Stock centers are as follows: BDSC = Bloomington *Drosophila* Stock Center; KDSC = Kyoto Drosophila Stock Center; JRC = Janelia Research Campus.

• Supplementary file 3. Metabolomics data for both manual and classifier-assisted experiments. Sheets with prefix 'All_' contain all of the data from the indicated metabolomics experiment. Sheet with prefix 'Overlap_' contains data for compounds that were observed in both experiments (compounds whose molecular weight corresponded within five parts per million [ppm]). Column headers ending in (M) indicate results from the manually-staged experiment; (C) from the classifier-staged experiments. Annotations for columns A, B, C, K, and N are as follows (with values of 1 or TRUE indicating the condition is met, 0 or FALSE that the condition is not met): 'Significant' = significantly different between NS and S across both experiments; 'Infected-specific' = compound only present in NS and S, but not C, samples, per manual chromatogram inspection; 'Summiting-specific' = compound only present in S, but not NS or C, samples, per manual chromatogram inspections; 'Same tentative formula?' = predicted formulae for given compound match exactly across experiments; 'Same sign?' = log2 fold change of compound abundance between S and NS samples is consistently positive or negative across experiments. Abbreviations: S = summiting, NS = exposed but non-summiting, C = unexposed control.

• MDAR checklist

### Data availability

Data supporting these results and the analysis code are available at http://lab.debivort.org/zombie-summiting/ and https://doi.org/10.5281/zenodo.7464925. All raw behavioral tracking (centroid versus time) data are available via Harvard Dataverse at https://doi.org/10.7910/DVN/LTMCFR.

The following datasets were generated:

| Author(s) | Year | Dataset title | Dataset URL | Database and Identifier |
|---|---|---|---|---|
| Elya C, Lavrentovich D, Lee E, Pasadyn C, Duval J, Basak M, Saykina V, de Bivort B | 2022 | Supporting data for Neural mechanisms of parasite-induced summiting behavior in 'Zombie' *Drosophila* | https://doi.org/10.5281/zenodo.7464925 | Zenodo, 10.5281/zenodo.7464925 |
| Elya C, Lavrentovich D, Lee E, Pasadyn C, Duval J, Basak M, Saykina V, de Bivort B | 2023 | Centroid tracking data for summiting flies | https://doi.org/10.7910/DVN/LTMCFR | Harvard Dataverse, 10.7910/DVN/LTMCFR |

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
