## [Editor Report]

The phenomenon of summit disease, where complex animal behaviours are controlled by single-celled parasites, captivates biologists and non-scientists alike. In this valuable study, the authors use a laboratory model (*Drosophila melanogaster* infected with Entomophthora muscae) for this disease to provide compelling evidence for the neuroanatomical and physiological underpinnings of summit disease. This is an excellent example of how seemingly intractable questions in behavioural ecology can be effectively addressed in laboratory settings using decades of work in creating 'models' for biology.

---

## [Decision Letter]

**Decision letter after peer review:**

Thank you for submitting your article "Neural mechanisms of parasite-induced summiting behavior in "zombie" *Drosophila*" for consideration by *eLife*. Your article has been reviewed by 3 peer reviewers, including Sonia Sen as the Reviewing Editor and Reviewer #1, and the evaluation has been overseen by K VijayRaghavan as the Senior Editor.

Essential revisions:

Circadian control of the behaviour:

The authors identify a few circadian neurons in their screen and show that the behaviour is circadian-locked. But to make the claim that the behaviour is under circadian control they would need to change photoperiod or knock down clock genes in DN1P and demonstrate that this alters behaviour. Could the authors please do either of these experiments? The findings in this paper would be interesting even if the results show that despite being circadian-locked, the behaviour is not under circadian control, the text can be altered to reflect this.

The writing:

There are a few things the authors could fix and I am listing them below.

1. Better descriptions of the screens, their analyses of the screens, and the metabolomics in the main text.

2. Closure on the candidates they did not follow up, who might be secreting them, and whether JH was on their list.

3. Aside from this, there are specific edits that have been recommended in the detailed reviews provided below that the authors would need to pay attention to.

*Reviewer #1 (Recommendations for the authors):*

I really enjoyed reading this manuscript and appreciate how well the authors have used the fly model to study an otherwise intractable, but extremely interesting, biology and behaviour. There are a few points I wanted to make, most of which will likely only be addressed through textual edits.

– The manuscript consists of two parts – the timeline of the fungal infection with the metabalomics, and the neuromechanistic framework of the behaviour. The manner in which the story is currently presented leaves a few loose ends and the reader with the impression that these are two somewhat disjointed stories. I am sure that this can be addressed through some deft writing : For example, would it be possible, to begin with, the timeline of infection and metabolomics and end with the circuitry? This would also tie in with their preferred mechanistic model. I recognise the possible issues with this approach (not knowing the PI-CA axis, for example), but it might be worth considering ending the story on firmer ground, albeit with a few loose ends (see below).

– The authors identify two factors – Dh31 and CNMa – as responsible for summiting behaviour. This was not followed up, but neither was it explicitly set aside for the reader. I also didn't completely understand who secretes them (DPN1?) and it looks like there were other factors too, which were not addressed in the text. Were any of the JH pathway genes in their screen?

– The model the authors favour is one where the release of metabolites from the fungus might induce the PI-CA axis to induce the release of JH, which might stimulate the locomotory burst prior to summiting. Aspects of this have not been worked out. For example, based on this, one might expect an increase in activity of the DPN1>PI>PC neurons upon infection (possibly addressable by CaMPARI or a similar tool). One might also expect an increase in JH at the time of summiting. The link between the fungal infection timeline and the metabolite build-up was also not clear to me. Neither was the implication of the location of the fungal infection to the pathology (do the secreted molecules have anything to do with this?). I am not suggesting that these experiments be done (unless the authors think them feasible), but that the authors revisit their writing to explicitly address these loose ends.

*Reviewer #2 (Recommendations for the authors):*

If there was some evidence of change in circadian phase by manipulating the photoperiod or by using the knockdown of circadian genes, it would have been more helpful to imagine circadian control. The best evidence at this time is the fact that the event anticipates lights off by a few hours which suggests that it is not a startle response to some environmental factor. However, one cannot rule out with the current experiments that the lights-on which happened 12 hours ago had set in motion a series of steps that eventually culminated in the locomotion followed by death around ZT 10-11.

Ln 72-73 Reference de Bekker 2015 is incorrect, it should be de Bekker 2022.

Ln 85 – 90 While the facts that in all the examples cited, the behaviour happens at a specific time of day, it is not automatic that they are under circadian control – merely that there is a daily rhythmicity which suggests circadian control. Hence I would re-word this statement.

Ln 270 – That DN1p receives inputs from sLNv was not shown by Kaneko & Hall. In fact, we have only indirect evidence and no proof that sLNv receives synaptic contact from sLNv – (recent connectomics data also suggest a total of 3 contacts, Shafer et al. 2022).

Ln 278 “DN1ps send their axons medially, with many projections terminating at or near the π (Kaneko and Hall, 2000).” This is not correct, DN1ps were not even categorised as such. Please refer to more recent literature (10.3389/fphys.2022.886432; Chatterjee et al., 2018; Guo et al., 2018; Lamaze et al., 2018).

Figure 2 – S1 D What do the two N values depicted represent? Bold and grey font?

Ln 496 – Is this surprising that the BBB is disrupted, considering how heavily the nervous system has been infiltrated by the fungus as shown in previous figure 5? It's not clear whether the question is about the temporal sequence of events. As it is presented now, it appears that BBB being disrupted is obvious. The supplementary data suggests that its disruption increases over time.

Ln 509: include the method (LC-MS) for metabolomics in the text.

Authors may wish to clarify how they arrive at the current model of an enhanced break in the barrier at -2.5 hrs while the fungus is already detected in the SMP much before that (almost 48 hrs). There is something wrong with how the legend describes a dashed outline (Ln572-3).

*Reviewer #3 (Recommendations for the authors):*

The authors write "(Following the convention of Wolff and Rubin, 2018, the letters before the dash indicate the postsynaptic compartment, the letters after the presynaptic compartment)."

Is this reversed? Should the letters before the dash indicate the presynaptic compartment?

What about the effects of the fungus on the thoracic and abdominal ganglions?

If injecting infected hemolymph is sufficient to trigger summiting behaviors, is the permeability of the BBB also affected by the infected hemolymph? Does this experiment indicate that the BBB effects are unimportant?

---

## [Author Response]

Essential revisions:Circadian control of the behaviour:The authors identify a few circadian neurons in their screen and show that the behaviour is circadian-locked. But to make the claim that the behaviour is under circadian control they would need to change photoperiod or knock down clock genes in DN1P and demonstrate that this alters behaviour. Could the authors please do either of these experiments? The findings in this paper would be interesting even if the results show that despite being circadian-locked, the behaviour is not under circadian control, the text can be altered to reflect this.

We apologize for a lack of clarity on our part: we did not aim to address the underpinnings of the timing of summiting behavior in this study and did not mean to suggest that the timing of summiting behavior is explained by the DN1p > PI-CA > CA circuit. Rather, we interpret the data presented here as evidence that the host’s circadian network is exploited by the fungus to mediate the characteristic burst of pre-death locomotor activity that we refer to as summiting. We have added the following paragraph in the discussion (see Host circadian and pars intercerebralis neurons mediate summiting) to clarify this point:

“Our data indicate that the host circadian network is involved in mediating the increased locomotor activity that we now understand to define summiting. However, our data do not speak to how the timing of this behavior is determined in the zombie-fly system. That is, we have yet to address the mechanisms underlying the temporal gating of summiting and death. Our observation that *E. muscae*-infected fruit flies continue to die at specific times of day in the absence of proximal lighting cues (Figure 1-S1) suggests that the timing of death is under circadian control and aligns with previous work in *E. muscae*-infected house flies (Krasnoff et al., 1995). Given that molecular clocks are prevalent across the tree of life, it is likely that two clocks (one in the fly, one in *E. muscae*) are present in this system. Additional work is needed to determine if the host clock is required for the timing of death under free-running conditions and to assess if *E. muscae* can keep time.”

This paragraph references new supplementary data that we’ve added showing that infected flies continue to die in a gated fashion under free-running conditions, addressing the reviewer’s suggestion of experiments that alter photoperiod to demonstrate that this behavior is under circadian control (Figure 1-S1). We have accordingly added the following sentence in the Results section at the end of A novel assay to measure summiting behavior:

“As expected, wild-type flies killed by E. muscae tended to die in the evening (mean death time = ZT9:50 Figure 1E), but there was variability in the timing of death. 90% of all deaths occurred between ZT7 and ZT12. *E. muscae*-exposed flies continued to die at specific times of day even in complete darkness (Figure 1-S1), suggesting that the timing of death is under circadian control.

The writing:There are a few things the authors could fix and I am listing them below.1. Better descriptions of the screens, their analyses of the screens, and the metabolomics in the main text.

Thank you for raising this concern. We have improved the description of our summiting screen (see first paragraph under Summiting behavior requires host circadian and neurosecretory pathways) with the following bolded edits.

“With the understanding that a burst of activity shortly before death is the signature of summiting in this assay, we performed a screen to identify circuit and genetic components mediating summiting in the host fly. We adopted a candidate approach, but cast a wide net for neurons and genes involved in neuromodulation or previously implicated in arousal and gravitaxis (Figure 2A-C, Table 1). To disrupt neurons, we drove the expression of tetanus toxin (TNT-E; a vesicle release blocker; Keller et al., 2002) using 103 different (Gal4 drivers Table 1). The effect size of each of these perturbations on summiting behavior was estimated relative to a common heterozygous control (*UAS-TNT-E*/+), and confidence intervals on each effect size was calculated by bootstrapping (Figure 2B). Similarly, we screened 101 lines targeting candidate genes, either by pan-neuronally reducing their expression via RNAi (i.e., driving CNS-wide expression of short hairpin RNAs targeting the desired gene) or testing mutant alleles (Table 1). Again, effect sizes were estimated by comparing each line’s summiting metric to common control genotypes, for pan-neuronal RNAi, the heterozygous pan-neuronal driver (*R57C10-Gal4*/+); for mutants, wild-type (CantonS) control (Figure 2C). Genotype details and our rationales for including each line in the screen are given in Tables 1 and 2. In both the circuit and genetic screens we observed a range of effects on summiting from extreme impairment of the behavior (effect size -1) to rare amplification of summiting (effect size > 0). Most perturbations had effects that were not statistically distinguishable from zero.“

We have also elaborated on our analysis strategy for our screen data. The following are our revisions to the second paragraph under Summiting behavior requires host circadian and neurosecretory pathways:

“Our manipulations targeted low-level biological elements (single genes and sparse neuronal expression patterns, as well as some broad expression patterns). To determine what higher level systems might be *E. muscae*’s target, we looked for enrichment of large effect sizes in the genes (or circuit elements) involved in the same higher level functions (or brain regions). We binned the behavioral data for each reagent type (i.e., neurons or genes) into quintiles according to effect size, looked at annotation frequencies across these bins, and noted annotations that occurred in a given quintile more frequently than expected by chance (Figure 3D, E). We found that neurons in the antennal mechanosensory and motor center (AMMC), subesophageal ganglion (SOG), circadian system, and pars intercerebralis (PI) were overrepresented in the quintile of most negative effect size (Figure 3D). Underscoring the potential importance of the PI, we observed that many of the neurons of large effect in the AMMC, SOG, and circadian system also innervated the π (Figure 3D – pink overlay). In a similar analysis for our genetic manipulations, we saw a clear enrichment for genes expressed in circadian cells (Figure 2E). Thus, our screen pointed conspicuously toward roles for the π and the circadian network in summiting behavior. “

We’ve also added more details of the metabolomics experiment in the main text (second paragraph under Evidence for the metabolic induction of summiting behavior)>

“We next used LC-MS metabolomics to compare the molecular composition of hemolymph in summiting flies to that of exposed, non-summiting flies. We performed this experiment twice: once staging animals by hand based on flightlessness, which occurs during mid to late summiting (Figure 1B), and a second time using our automated classifier. For each experiment, we collected 1 μL samples of hemolymph bled from a pool of 20 mated females for each of three conditions: (1) healthy (unexposed flies), (2) exposed, non-summiting, and (3) summiting. Triplicate samples were analyzed when the classifier was employed (Figure 6-S2B) and duplicate samples were analyzed in the manual experiment (Figure 6-S2C). We found that 168 compounds were detected in both of these experiments (Figure 6B, Figure 6-S2A-C),…”

2. Closure on the candidates they did not follow up, who might be secreting them, and whether JH was on their list.

We assume that the reviewer is asking for close on screened lines that had significant effects on summiting, but were not the focus of follow-up experiments. We have added additional descriptions of these hits in Figure 2F, and clarification as to who is secreting what:

“With these high level systems implicated as targets of fungal manipulation, we returned to a granular analysis to determine what specific circuit elements in circadian cells and the π best recapitulated the high level effects. We measured the summiting response of an individually tailored genetic control for each circadian gene and π or circadian circuit element (rather than screen-wide controls), and recalculated the effect size of each perturbation (Figure 2F,G). With respect to the circadian experiments, eleven mutants (Figure 2F) and four Gal4 lines (Figure 2G) showed impaired summiting compared to matched genetic background and/or sibling controls. Three different mutants of *Clock* (*Clk*), a gene expressed in all clock cells, showed greatly reduced summiting behavior (62%-104%, 3.4e-28 < p < 7e-8). The cryptochrome gene (*cry*) encodes a blue light sensor expressed by a subset of circadian neurons that synchronizes the molecular oscillator with environmental lighting cues (Emery et al., 2000; Benito et al., 2008; Yoshii et al., 2008). A *cry* mutant and a pan-neuronal RNAi knockdown of *cry* both showed reduced summiting (32%, p = 0.018; 45%, p = 0.00097, respectively).

We noticed that several of our hits affected a subtype of clock neurons, the group 1 posterior dorsal neurons (DN1ps). DN1ps are a heterogeneous population of neurons numbering approximately 15 cells per brain hemisphere (Ma et al., 2021). About half of DN1ps express *cry* (Yoshii et al., 2008). Silencing neurons with two drivers that label many, but not all, of the DN1ps (Clk4.1 and R18H11; Zhang et al., 2010; Kunst et al., 2014) via TNT-E expression reduced summiting by 24-25% (p = 0.005, 0.019; Figure 2G, Figure 2-S1B,C). However, silencing the entire population of DN1p neurons by driving the inward-rectifying potassium channel Kir2.1 (Baines et al., 2001) with a pan-DN1p driver had no apparent effect (Figure 2-S1D) as did silencing neurons labeled by an additional driver previously reported to be expressed in DN1ps (R51H05; Kunst et al., 2014). Silencing a sparser population of DN1ps (Clk4.5) with TNT-E led to an increase in summiting (Figure 2G). Genetic disruption of two signaling molecules expressed by DN1ps, Diuretic Hormone 31 (*Dh31*) and the neuropeptide CNMamide (*CNMa*), reduced summiting by 59 to 72% (3e-16 < p < 0.025; Figure 2F). However, flies mutant for the receptors that recognize these molecules (*Dh31R* and *PdfR* for *Dh31*; *CNMaR* for *CNMa*) did not show significantly impaired summiting (0.054<p<0.3), though *Dh31R* came close with a 33% impairment at p=0.054. Taken together, these results implicate DN1ps as mediating fungal manipulation while also revealing fine-scale complexity, as activity in some DN1ps, but not others, is required for full summiting.

DN1p activity is influenced by a class of pacemaker neurons called small ventrolateral neurons (sLNvs) (Zhang et al., 2010) that express the neuropeptide Pigment-dispersing factor (Pdf; Helfrich-Förster and Homberg, 1993; Renn et al., 1999). While one *Pdf* mutant (*Pdf^01^*) exhibited a large, significant reduction in summiting (67%; p = 1.8e-16; Figure 2F), we saw no effect with another mutant whose *Pdf* locus was completely replaced (*Pdf^-^*). We also did not observe a significant decrease in summiting in Pdf receptor (PdfR) mutants (0.3 < p < 0.38). Disrupting sLNVs by expressing TNT-E, channel Kir2.1, or pro-apoptotic protein hid (Grether et al., 1995) also had no effect on summiting (Figure 2-S1D, E). This suggests that the main population of clock neurons upstream of DN1ps is irrelevant for summiting.”

Genes involved in JH metabolism were not part of the initial screen, which we now mention in the discussion (see second paragraph of PI-CA neurons induce summiting via their connection to the corpora allata):

“We showed that summiting was reduced in *E. muscae*-infected flies with ablated CA (Figure 3C) or when treated with the JH synthesis inhibitor precocene (Figure 3E). However, we did not observe exacerbated summiting behavior in animals that had been treated with the juvenile hormone analog (JHA) methoprene (Figure 3-S2H) or a restoration of summiting behavior when animals received JHAs in addition to precocene (Figure 3-S2I). JH manipulations were not part of our initial screen, becoming a focus after the discovery of the role of the PI-CA neurons.”

We also now clarify in the discussion that we didn’t detect JH in our metabolomics experiment (see fourth paragraph of PI-CA neurons induce summiting via their connection to the corpora allata):

“Thus, another possibility is that the fungus is metabolizing the JHAs before they have a behavioral effect. We did not detect JH in any of our metabolomic experiments, however this was expected given that we used extraction and separation methods appropriate for polar, not hydrophobic, compounds. Future work leveraging targeted, high-sensitivity chemical detection of hydrophobic compounds is needed to verify that JH titers are indeed elevated during the transient summiting window.”

Additional clarifying edits are described in the detailed responses to reviewer comments below.

3. Aside from this, there are specific edits that have been recommended in the detailed reviews provided below that the authors would need to pay attention to.

Thank you for your comment. We agree that there are many loose ends at this point, though we like to think of them as “future research opportunities.” We are hoping that the manuscript is nevertheless satisfying for addressing many aspects of the mechanisms of summiting.

We originally decided to organize the manuscript with the metabolomics at the end because this section culminates in an important, open question: what is the identity of the effector in the hemolymph? We thought that placing such an open question at the end of the manuscript was more appropriate than in the middle of the manuscript, as it segues naturally to the discussion of future work. We acknowledge that our narratively abruptly transitioned from results on fly neural circuits to fungal inputs. We have modified the text to ease this transition:

“Having identified a neurohormonal circuit that is required in the fly host for summiting, we next sought to investigate how the fungus gains access to this target and manipulates it to induce summiting. We reasoned that there may be physiological and anatomical differences between summiting and non-summiting flies that reflect causal mechanisms on the fungal side.”

Reviewer #1 (Recommendations for the authors):I really enjoyed reading this manuscript and appreciate how well the authors have used the fly model to study an otherwise intractable, but extremely interesting, biology and behaviour. There are a few points I wanted to make, most of which will likely only be addressed through textual edits.– The manuscript consists of two parts – the timeline of the fungal infection with the metabalomics, and the neuromechanistic framework of the behaviour. The manner in which the story is currently presented leaves a few loose ends and the reader with the impression that these are two somewhat disjointed stories. I am sure that this can be addressed through some deft writing : For example, would it be possible, to begin with, the timeline of infection and metabolomics and end with the circuitry? This would also tie in with their preferred mechanistic model. I recognise the possible issues with this approach (not knowing the PI-CA axis, for example), but it might be worth considering ending the story on firmer ground, albeit with a few loose ends (see below).

Thanks for highlighting this gap. The sentence “Genetic disruption of two signaling molecules expressed by DN1ps, Diuretic Hormone 31 (*Dh31*) and the neuropeptide CNMamide (*CNMa*)” now clarifies that DN1ps secrete these compounds.

We have expanded the Results section to address additional reagents shown in Figure 2F and contextualize them within the circadian clock network (receptors for Dh31, Pdf and CNMa), as described in our response to the high-level editorial suggestions above. These edits are repeated here, for the reviewer’s convenience:

“With these high level systems implicated as targets of fungal manipulation, we returned to a granular analysis to determine what specific circuit elements in circadian cells and the π best recapitulated the high level effects. We measured the summiting response of an individually tailored genetic control for each circadian gene and π or circadian circuit element (rather than screen-wide controls), and recalculated the effect size of each perturbation (Figure 2F,G). With respect to the circadian experiments, eleven mutants (Figure 2F) and four Gal4 lines (Figure 2G) showed impaired summiting compared to matched genetic background and/or sibling controls. Three different mutants of *Clock* (*Clk*), a gene expressed in all clock cells, showed greatly reduced summiting behavior (62%-104%, 3.4e-28 < p < 7e-8). The cryptochrome gene (*cry*) encodes a blue light sensor expressed by a subset of circadian neurons that synchronizes the molecular oscillator with environmental lighting cues (Emery et al., 2000; Benito et al., 2008; Yoshii et al., 2008). A *cry* mutant and a pan-neuronal RNAi knockdown of *cry* both showed reduced summiting (32%, p = 0.018; 45%, p = 0.00097, respectively).

We noticed that several of our hits affected a subtype of clock neurons, the group 1 posterior dorsal neurons (DN1ps). DN1ps are a heterogeneous population of neurons numbering approximately 15 cells per brain hemisphere (Ma et al., 2021). About half of DN1ps express *cry* (Yoshii et al., 2008). Silencing neurons with two drivers that label many, but not all, of the DN1ps (Clk4.1 and R18H11; Zhang et al., 2010; Kunst et al., 2014) via TNT-E expression reduced summiting by 24-25% (p = 0.005, 0.019; Figure 2G, Figure 2-S1B,C). However, silencing the entire population of DN1p neurons by driving the inward-rectifying potassium channel Kir2.1 (Baines et al., 2001) with a pan-DN1p driver had no apparent effect (Figure 2-S1D) as did silencing neurons labeled by an additional driver previously reported to be expressed in DN1ps (R51H05; Kunst et al., 2014). Silencing a sparser population of DN1ps (Clk4.5) with TNT-E led to an increase in summiting (Figure 2G). Genetic disruption of two signaling molecules expressed by DN1ps, Diuretic Hormone 31 (*Dh31*) and the neuropeptide CNMamide (*CNMa*), reduced summiting by 59 to 72% (3e-16 < p < 0.025; Figure 2F). However, flies mutant for the receptors that recognize these molecules (*Dh31R* and *PdfR* for *Dh31*; *CNMaR* for *CNMa*) did not show significantly impaired summiting (0.054<p<0.3), though *Dh31R* came close with a 33% impairment at p=0.054. Taken together, these results implicate DN1ps as mediating fungal manipulation while also revealing fine-scale complexity, as activity in some DN1ps, but not others, is required for full summiting.

DN1p activity is influenced by a class of pacemaker neurons called small ventrolateral neurons (sLNvs) (Zhang et al., 2010) that express the neuropeptide Pigment-dispersing factor (Pdf; Helfrich-Förster and Homberg, 1993; Renn et al., 1999). While one *Pdf* mutant (*Pdf^01^*) exhibited a large, significant reduction in summiting (67%; p = 1.8e-16; Figure 2F), we saw no effect with another mutant whose *Pdf* locus was completely replaced (*Pdf^-^*). We also did not observe a significant decrease in summiting in Pdf receptor (*PdfR*) mutants (0.3 < p < 0.38). Disrupting sLNVs by expressing TNT-E, channel Kir2.1, or pro-apoptotic protein hid (Grether et al., 1995) also had no effect on summiting (Figure 2-S1D, E). This suggests that the main population of clock neurons upstream of DN1ps is irrelevant for summiting.”

JH pathway genes were not included in our initial screen. As stated in response to general comments, this is now clearly stated in the discussion. The new wording is repeated here, for the reviewer’s convenience.

“We showed that summiting was reduced in *E. muscae*-infected flies with ablated CA (Figure 3C) or when treated with the JH synthesis inhibitor precocene (Figure 3E). However, we did not observe exacerbated summiting behavior in animals that had been treated with the juvenile hormone analog (JHA) methoprene (Figure 3-S2H) or a restoration of summiting behavior when animals received JHAs in addition to precocene (Figure 3-S2I). JH manipulations were not part of our initial screen, becoming a focus after the discovery of the role of the PI-CA neurons.”

– The authors identify two factors – Dh31 and CNMa – as responsible for summiting behaviour. This was not followed up, but neither was it explicitly set aside for the reader. I also didn't completely understand who secretes them (DPN1?) and it looks like there were other factors too, which were not addressed in the text. Were any of the JH pathway genes in their screen?

Thanks for highlighting this gap. The sentence “Genetic disruption of two signaling molecules expressed by DN1ps, Diuretic Hormone 31 (*Dh31*) and the neuropeptide CNMamide (*CNMa*)” now clarifies that DN1ps secrete these compounds.

We have expanded the Results section to address additional reagents shown in Figure 2F and contextualize them within the circadian clock network (receptors for Dh31, Pdf and CNMa), as described in our response to the high-level editorial suggestions above. These edits are repeated here, for the reviewer’s convenience:

“With these high level systems implicated as targets of fungal manipulation, we returned to a granular analysis to determine what specific circuit elements in circadian cells and the π best recapitulated the high level effects. We measured the summiting response of an individually tailored genetic control for each circadian gene and π or circadian circuit element (rather than screen-wide controls), and recalculated the effect size of each perturbation (Figure 2F,G). With respect to the circadian experiments, eleven mutants (Figure 2F) and four Gal4 lines (Figure 2G) showed impaired summiting compared to matched genetic background and/or sibling controls. Three different mutants of *Clock* (*Clk*), a gene expressed in all clock cells, showed greatly reduced summiting behavior (62%-104%, 3.4e-28 < p < 7e-8). The cryptochrome gene (*cry*) encodes a blue light sensor expressed by a subset of circadian neurons that synchronizes the molecular oscillator with environmental lighting cues (Emery et al., 2000; Benito et al., 2008; Yoshii et al., 2008). A *cry* mutant and a pan-neuronal RNAi knockdown of *cry* both showed reduced summiting (32%, p = 0.018; 45%, p = 0.00097, respectively).

We noticed that several of our hits affected a subtype of clock neurons, the group 1 posterior dorsal neurons (DN1ps). DN1ps are a heterogeneous population of neurons numbering approximately 15 cells per brain hemisphere (Ma et al., 2021). About half of DN1ps express *cry* (Yoshii et al., 2008). Silencing neurons with two drivers that label many, but not all, of the DN1ps (Clk4.1 and R18H11; Zhang et al., 2010; Kunst et al., 2014) via TNT-E expression reduced summiting by 24-25% (p = 0.005, 0.019; Figure 2G, Figure 2-S1B,C). However, silencing the entire population of DN1p neurons by driving the inward-rectifying potassium channel Kir2.1 (Baines et al., 2001) with a pan-DN1p driver had no apparent effect (Figure 2-S1D) as did silencing neurons labeled by an additional driver previously reported to be expressed in DN1ps (R51H05; Kunst et al., 2014). Silencing a sparser population of DN1ps (Clk4.5) with TNT-E led to an increase in summiting (Figure 2G). Genetic disruption of two signaling molecules expressed by DN1ps, Diuretic Hormone 31 (*Dh31*) and the neuropeptide CNMamide (*CNMa*), reduced summiting by 59 to 72% (3e-16 < p < 0.025; Figure 2F). However, flies mutant for the receptors that recognize these molecules (*Dh31R* and *PdfR* for *Dh31*; *CNMaR* for *CNMa*) did not show significantly impaired summiting (0.054<p<0.3), though *Dh31R* came close with a 33% impairment at p=0.054. Taken together, these results implicate DN1ps as mediating fungal manipulation while also revealing fine-scale complexity, as activity in some DN1ps, but not others, is required for full summiting.

DN1p activity is influenced by a class of pacemaker neurons called small ventrolateral neurons (sLNvs) (Zhang et al., 2010) that express the neuropeptide Pigment-dispersing factor (Pdf; Helfrich-Förster and Homberg, 1993; Renn et al., 1999). While one *Pdf* mutant (*Pdf^01^*) exhibited a large, significant reduction in summiting (67%; p = 1.8e-16; Figure 2F), we saw no effect with another mutant whose *Pdf* locus was completely replaced (*Pdf^-^*). We also did not observe a significant decrease in summiting in Pdf receptor (*PdfR*) mutants (0.3 < p < 0.38). Disrupting sLNVs by expressing TNT-E, channel Kir2.1, or pro-apoptotic protein hid (Grether et al., 1995) also had no effect on summiting (Figure 2-S1D, E). This suggests that the main population of clock neurons upstream of DN1ps is irrelevant for summiting.”

JH pathway genes were not included in our initial screen. As stated in response to general comments, this is now clearly stated in the discussion. The new wording is repeated here, for the reviewer’s convenience.

“We showed that summiting was reduced in *E. muscae*-infected flies with ablated CA (Figure 3C) or when treated with the JH synthesis inhibitor precocene (Figure 3E). However, we did not observe exacerbated summiting behavior in animals that had been treated with the juvenile hormone analog (JHA) methoprene (Figure 3-S2H) or a restoration of summiting behavior when animals received JHAs in addition to precocene (Figure 3-S2I). JH manipulations were not part of our initial screen, becoming a focus after the discovery of the role of the PI-CA neurons.”

– The model the authors favour is one where the release of metabolites from the fungus might induce the PI-CA axis to induce the release of JH, which might stimulate the locomotory burst prior to summiting. Aspects of this have not been worked out. For example, based on this, one might expect an increase in activity of the DPN1>PI>PC neurons upon infection (possibly addressable by CaMPARI or a similar tool). One might also expect an increase in JH at the time of summiting. The link between the fungal infection timeline and the metabolite build-up was also not clear to me. Neither was the implication of the location of the fungal infection to the pathology (do the secreted molecules have anything to do with this?). I am not suggesting that these experiments be done (unless the authors think them feasible), but that the authors revisit their writing to explicitly address these loose ends.

We agree that there are many details of our proposed framework that have yet to be worked out. The scope of the mechanism that we proposed is very broad and we don’t think all details can be satisfactorily addressed in a single paper. That said, we have made various edits to better resolve these loose ends in the main text. For example, we made these changes:

Penultimate paragraph of Discussion: PI-CA neurons induce summiting via their connection to the corpora allata

“We showed that summiting was reduced in *E. muscae*-infected flies with ablated CA (Figure 3C) or when treated with the JH synthesis inhibitor precocene (Figure 3E). However, we did not observe exacerbated summiting behavior in animals that had been treated with the juvenile hormone analog (JHA) methoprene (Figure 3-S2H) or a restoration of summiting behavior when animals received JHAs in addition to precocene (Figure 3-S2I). JH manipulations were not part of our initial screen, becoming a focus after the discovery of the role of the PI-CA neurons. We suspect that summiting is driven by an acute spike in JH starting ~2.5 hours before death, and our JHA experiments did not have this timing: methoprene was delivered in a single burst 20 hours prior to summiting and pyriproxyfen was administered chronically via the food. Secondly, we have strong reason to believe that whatever we applied to the fly was also making its way to the fungus (recall that healthy flies treated with both fluvastatin and methoprene were fine, but that this treatment was lethal for exposed flies (Figure 3-S2D)). Thus, another possibility is that the fungus is metabolizing the JHAs before they have a behavioral effect. We did not detect JH in any of our metabolomic experiments, however this was expected given that we used extraction and separation methods appropriate for polar, not hydrophobic, compounds. Future work leveraging targeted, high-sensitivity chemical detection of hydrophobic compounds is needed to verify that JH titers are indeed elevated during the transient summiting window.”

First paragraph of Discussion: Host circadian and pars intercerebralis neurons mediate summiting

“We leveraged our high throughput assay to screen for fly circuit elements mediating summiting and found evidence for the involvement of circadian and neurosecretory systems (Figure 2A-E). We identified two specific neuronal populations important for summiting: DN1p circadian neurons labeled by *Clk4.1-Gal4* (Figure 2F) and a small population of PI-CA neurons labeled by *R19G10-Gal4* (Figure 2G). Silencing these neurons significantly reduced summiting and ectopically activating them induced a summiting-like burst of locomotor activity (Figure 2I-K). These neurons are likely part of the same circuit; the projection of DN1ps to the π has been confirmed both anatomically (Cavanaugh, et al., 2014) and functionally (Barber et al., 2021). Future work to visualize PI-CA and DN1p activity during summiting is needed to verify this assertion.”

First paragraph of Discussion: Morphological correlates of summiting

“The distribution of *E. muscae* across neuropils, which is consistent across animals (Figure 5C)*,* is interesting both for where fungal cells are and are not found. Fungal cells are noticeably absent from the central complex, a pre-motor center (Bender et al., 2010; Strausfeld, 1976) that may be involved in coordinating walking during summiting. Though morphological examination suggested that fungal cells are displacing (Figure 5-S1B), rather than consuming, nervous tissue, more work is needed to determine if neurons are damaged or dying as a result of adjacent fungal cells. In addition, it remains unclear what role, if any, the pattern of fungal brain occupancy plays in the mechanism of summiting or if the fungal cells in the brain play a distinct role in behavior manipulation compared to those in the body cavity. Additional work is needed to address these questions.”

Last paragraph of Discussion: Physiological correlates of summiting

“To test whether hemolymph-circulating factors in summiting animals can cause an increase in locomotion, we transfused hemolymph from classifier-flagged summiting flies into fungus-exposed and non-exposed recipients (Figure 6C,D). In both of these experiments, recipient flies exhibited a significant increase in locomotion over ~1.5 hours post-transfusion. The effect size was modest (40% increase in total distance traveled in that interval), but this was not surprising as (1) we could only extract and transfer very small quantities (MacMillan et al., 2014) of hemolymph between animals and (2) this small quantity was diluted throughout the whole recipient fly’s body. Overall, this experiment provides direct evidence that one or more factors in the hemolymph of summiting flies cause summiting. The identity of these factors and their precise timing and origin of production (fungal or fly) remain mysteries that we hope to address in future studies.”

Looking at neural activity in π cells during summiting is a great idea, but will require improvement of existing techniques in order to be viable. We attempted to measure Ca++ levels in DN1ps and PI-CA neurons using 2-photon imaging during application of hemolymph from summiting flies to the saline bath. These experiments were not successful for technical reasons, predominantly difficulties constructing a flow chamber to direct a tiny volume of summiting hemolymph to perfuse a living fly brain without getting lost in the bulk volume of saline, while at the same maintaining oxygenation of tissues. We hope to resolve these challenges and do this experiment in the future. We likewise attempted R19G10>Campari experiments, but did not detect a signal in positive control flies that expressed Campari under pan-neuronal-Gal4 and were exposed to apple cider vinegar. At that point, we set these experiments aside, concerned that the UV needed to activate Campari might interfere with the fungus, whose effects are light- and heat-sensitive. Like the GCaMP experiments, we feel these technical challenges are worth revisiting in the future.

We attempted to measure JH levels to test our hypothesis, but were unable to reliably detect JH in summiting hemolymph. We hypothesize that circulating JH titers could be exceptionally low due to a few possibilities: (1) acute starvation (production of JH is regulated by nutritional state, e.g., Clifton & Noriega, 2012, http://dx.doi.org/10.1016/j.jinsphys.2012.05.005), (2) degradation by *E. muscae* (we have observed that compounds we’ve tried to deliver to the fly via injection/feeding have wound up affecting the fungus – fluvastatin, dextran-labeled beads). We speculate that the local release of JH from the intact CA is sufficient to drive behavior. However, given the quantities of material involved, we haven’t yet been able to do that experiment.

We don’t yet understand the kinetics of metabolomic changes. We have only sampled time-matched non-summiting and summiting animals. We also do not yet understand whether the localization of fungal cells in the brain is important for driving summiting, just that a specific pattern of occupancy correlates with summiting.

Reviewer #2 (Recommendations for the authors):If there was some evidence of change in circadian phase by manipulating the photoperiod or by using the knockdown of circadian genes, it would have been more helpful to imagine circadian control. The best evidence at this time is the fact that the event anticipates lights off by a few hours which suggests that it is not a startle response to some environmental factor. However, one cannot rule out with the current experiments that the lights-on which happened 12 hours ago had set in motion a series of steps that eventually culminated in the locomotion followed by death around ZT 10-11.

We are similarly intrigued by the mechanism of timing of summit behavior. We regret not having clearly stated this in our manuscript. As mentioned above, we have added a paragraph in Host circadian and pars intercerebralis neurons mediate summiting to clarify this point.

“Our data indicate that the host circadian network is involved in mediating the increased locomotor activity that we now understand to define summiting. However, our data do not speak to how the timing of this behavior is determined in the zombie-fly system. That is, we have yet to address the mechanisms underlying the temporal gating of summiting and death. Our observation that *E. muscae*-infected fruit flies continue to die at specific times of day in the absence of proximal lighting cues (Figure 1-S1) suggests that the timing of death is under circadian control and aligns with previous work in *E. muscae*-infected house flies (Krasnoff et al., 1995). Given that molecular clocks are prevalent across the tree of life, it is likely that two clocks (one in the fly, one in *E. muscae*) are present in this system. Additional work is needed to determine if the host clock is required for the timing of death under free-running conditions and to assess if *E. muscae* can keep time.”

Ln 72-73 Reference de Bekker 2015 is incorrect, it should be de Bekker 2022

We have double-checked the cited reference and believe it to be correct. The text with this citation states: “work in Ophiocordyceps suggests that the parasitic fungus may use enterotoxins and small secreted proteins to mediate end-of-life “zombie” behaviors (Beckerson et al., 2022; de Bekker et al., 2015; Will et al., 2020).” The pertinent result from the abstract of de Bekker et al., 2015 is: “we found genes up-regulated during manipulation that putatively encode for proteins with reported effects on behavioral outputs, proteins involved in various neuropathologies and proteins involved in the biosynthesis of secondary metabolites such as alkaloids.”

Ln 85 – 90 While the facts that in all the examples cited, the behaviour happens at a specific time of day, it is not automatic that they are under circadian control – merely that there is a daily rhythmicity which suggests circadian control. Hence I would re-word this statement.

Thank you for the correction. We have modified the text accordingly:

*“E. muscae* kills flies at a specific time of day: flies die around sunset and exhibit their final bout of locomotion between 0-5 hours prior to lights off (Elya et al., 2018; Krasnoff et al., 1995). Time-of-day specificity is a common feature of fungal-induced summit disease:”

Ln 270 – That DN1p receives inputs from sLNv was not shown by Kaneko & Hall. In fact, we have only indirect evidence and no proof that sLNv receives synaptic contact from sLNv – (recent connectomics data also suggest a total of 3 contacts, Shafer et al. 2022).

We apologize for citing the wrong reference for this observation. Indeed, Shafer et al., 2022 reports only sparse synaptic connectivity between sLNvs and DN1ps (Figure 1C of that paper). But it is important to note that roughly half of the DN1p neurons had not yet been annotated in the hemibrain. Cavanaugh et al., 2014 demonstrated sLNv-DN1p adjacency by GRASP, and this signal may come from DN1ps not annotated in the hembrain. Nevertheless, the reviewer is correct that the evidence for direct synaptic contact between sLNvs and DN1ps is thin. That said, it is relevant that Pdf can act at a distance via diffusion (i.e.,, non-synaptically). We have edited the text to be more precise in our wording:

“DN1p activity is influenced by a class of pacemaker neurons called small ventrolateral neurons (sLNvs) (Zhang et al., 2010) that express the neuropeptide Pigment-dispersing factor (Pdf; Helfrich-Förster and Homberg, 1993; Renn et al., 1999).”

Ln 278 "DN1ps send their axons medially, with many projections terminating at or near the π (Kaneko and Hall, 2000)." This is not correct, DN1ps were not even categorised as such. Please refer to more recent literature (10.3389/fphys.2022.886432; Chatterjee et al., 2018; Guo et al., 2018; Lamaze et al., 2018).

Thank you for your correction. We have changed our wording here to reflect that DN1p neurons send fibers (rather than axons, specifically) to the midline and, per the Hemibrain and Reinhard et al., 2022, have presynaptic sites at or near the PI:

“DN1ps send some processes medially, with presynaptic sites occurring at or near the π (Reinhard et al., 2022; Chatterjee et al., 2018). ”

The pertinent text from Reinhard et al., 2022 is: “Previous studies … revealed putative presynaptic output sites of the DN_1p_s in the SMP close to the pars intercerebralis and in the AOTU … We used the connectome data of the hemibrain (Scheffer et al., 2020) to reveal the input and output sites of the DN_1p_s and could confirm the results of Chatterjee et al. (2018) …”

Figure 2 – S1 D What do the two N values depicted represent? Bold and grey font?

These represent the sample size of experimental (black) and control (gray) animals, respectively. We have added this information to the figure caption.

Ln 496 – Is this surprising that the BBB is disrupted, considering how heavily the nervous system has been infiltrated by the fungus as shown in previous figure 5? It's not clear whether the question is about the temporal sequence of events. As it is presented now, it appears that BBB being disrupted is obvious. The supplementary data suggests that its disruption increases over time.

Thank you for your comment. Our view is that it was not a foregone conclusion that the BBB would show substantial disruption because it was unclear whether each cell in the brain arrived by invasion from the hemolymph or by division of cells already occupying the brain. Moreover, we did not know how the fungal cells entered the CNS — invasion via a transcellular mechanism could preclude any damage to the BBB. It is also possible that fungal cells could pass through the BBB and the BBB then reseals tightly afterward, resulting in minimal loss of integrity.

We agree that the possibility of BBB disruption seemed very likely *a priori*, but we could not be certain that this was the case until we did the experiment. We speculate that the increase in BBB disruption over time is a consequence of more cumulative damage to the BBB by invading fungal cells and additional stress placed on infected animals.

Ln 509: include the method (LC-MS) for metabolomics in the text.

Good catch. We have updated the text to specify that by metabolomics we specifically mean LC-MS.

“We next used LC-MS metabolomics to compare the molecular composition of hemolymph in summiting flies to that of exposed, non-summiting flies. “

Authors may wish to clarify how they arrive at the current model of an enhanced break in the barrier at -2.5 hrs while the fungus is already detected in the SMP much before that (almost 48 hrs). There is something wrong with how the legend describes a dashed outline (Ln572-3).

Our model (Figure 7) includes a gradual increase in BBB permeability starting at least 48h before summiting. We had depicted this in Figure 7 with increasing the visibility of the dashed line between steps 1, 2 and 3. We described this depiction in the caption as follows: “The dashed outline of the brain becomes more prominent between steps 1 and 3 to reflect an increase in BBB permeability over these timepoints.” Since this may not be sufficiently clear, we have added a “permeable BBB” label to step 2 of this Figure and we have updated the figure so now the gaps in the BBB increase in size between Steps 1 and Step 3. We described this depiction in the caption as follows: “The dashed outline of the brain becomes more prominent between steps 1 and 3 to reflect an increase in BBB permeability over these timepoints.”

Reviewer #3 (Recommendations for the authors):The authors write "(Following the convention of Wolff and Rubin, 2018, the letters before the dash indicate the postsynaptic compartment, the letters after the presynaptic compartment)."Is this reversed? Should the letters before the dash indicate the presynaptic compartment?

The letters before the dash indicate predominantly dendritic compartments, which contain postsynaptic sites. After the dash are any axonic (presynaptic) compartments, thus the letters suggest the flow of information. The pertinent passage from Wolff and Rubin 2018 is: “(… 5) Apparent input (spiny morphology) neuropils are listed first followed by neuropils with mixed input and output followed last by apparent output neuropils (boutons). (6) Input, mixed, and output neuropils are separated with hyphens.”

What about the effects of the fungus on the thoracic and abdominal ganglions?

This is a great question. We haven’t yet looked, but anticipate doing so in an upcoming study.

If injecting infected hemolymph is sufficient to trigger summiting behaviors, is the permeability of the BBB also affected by the infected hemolymph? Does this experiment indicate that the BBB effects are unimportant?

Our interpretation is that we are only seeing a partial recapitulation of summiting behavior in transfused recipients as the difference between distance traveled/speed of summiting versus non-summiting recipients is much smaller than between zombies and survivors. We think that this may be for several reasons, including limited BBB permeability in recipient flies, and also that the effector molecules in the hemolymph are greatly diluted when transfused.

We did see a larger difference in distance traveled when summiting hemolymph was transfused into exposed recipients, compared to unexposed recipients. This difference, however, is likely not statistically significant and we did not formally test it. Additionally, as we state in the text, any potential difference here is confounded by a change in the sex of recipient flies.

One hypothesis in which BBB permeabilization could be essential is the following. A full summiting effect may only be seen when a critical number of fungal cells are present in sufficient proximity to target host neurons so that their secreted molecular effectors are at a high enough local concentration to induce a strong effect in the host. We hope to more comprehensively address the importance of BBB integrity in our future work.